# Differentially Private Range Subgraph Counting

Xian Chen [* 1]  Ruobing Bai [* 1]  Pan Peng [1]

## Abstract

Subgraph counting is a fundamental problem in graph analysis. Motivated by practical scenarios where graph analytics are performed on subgraphs induced by selected vertices – rather than on the entire graph – and by growing privacy concerns, we initiate the study of *differentially private range subgraph counting (DPRSC)*. The goal is to privately count occurrences of a fixed pattern graph within induced subgraphs defined by multi-dimensional attribute ranges. Unlike classical point counting, subgraph counting is inherently nonlinear and exhibits high sensitivity: a single edge modification can affect many subgraph occurrences. We present the first efficient algorithms for DPRSC with small additive error. Our approach introduces a subgraph projection that reduces DPRSC to weighted orthogonal range counting, enabling the use of range trees and local sensitivity estimation to achieve accurate private query answering. We complement our algorithms with matching lower bounds, obtained by reducing reconstruction attacks to DPRSC and leveraging discrepancy theory. In particular, we show that any differentially private algorithm for DPRSC must incur additive error exponential in the dimension. Empirical evaluations demonstrate that our algorithms significantly outperform baseline methods in accuracy and runtime while maintaining strong privacy guarantees.

## 1. Introduction

*Subgraph counting* is essential for understanding the properties of a data graph and has been extensively studied (Chiba & Nishizeki, 1985; Alon et al., 1995; Björklund et al., 2014; Curticapean et al., 2017; Assadi et al., 2019; Fichtenberger et al., 2020). Given a *host* graph $G = (V, E)$ and a *pattern*

graph $H$, a subgraph of $G$ that is isomorphic to $H$ is called an *occurrence* of $H$. The goal is to compute the number of such occurrences. Subgraph counts serve as key graph statistics: for instance, triangles and $k$-stars are central to computing the clustering coefficient, which are widely used to evaluate social and recommendation networks, while counting four-cycles is particularly informative for capturing clustering behavior in bipartite graphs such as online dating and mentor-student networks.

In many applications, beyond counting subgraphs in the entire graph, we are often interested in counting subgraphs within *specific induced subgraphs* determined by vertex attributes. For instance, in patient or social networks, analysts may wish to count patterns among individuals within certain age ranges or geographic regions. In financial networks, counting transaction patterns among entities with similar risk profiles or locations can help identify fraudulent activities or assess systemic risks. More generally, vertices often carry multidimensional attributes, and practitioners seek to issue repeated subgraph-counting queries restricted to vertices whose attributes fall within specified ranges.

Motivated by these scenarios, we introduce the *Range Subgraph Counting* problem that combines classical subgraph counting with multidimensional range queries over vertex attributes.

**Definition 1.1** (Range Subgraph Counting (RSC))**.** Let $G = (V, E)$ be an $n$-vertex undirected graph, where each vertex $v \in V$ is associated with an attribute vector $\mathbf{a}(v) \in \mathbb{R}^d$. For a query range $q = [\ell_1, r_1] \times \cdots \times [\ell_d, r_d]$, define

$$V_q = \{v \in V \mid \ell_i \leq a_i(v) \leq r_i, i \in [d]\}.$$

Let $G_q = G[V_q]$ denote the subgraph of $G$ induced by $V_q$.

Let $H$ be a fixed connected pattern graph with $O(1)$ vertices and let $Q$ be a set of query ranges. The goal is to return, for each query $q \in Q$, the number of occurrences of $H$ in $G_q$.

Note that vertex attributes may represent age, time or location, depending on the practical context. Additionally, an occurrence is only counted if all its vertices are contained within $V_q$. Prior work by Deng et al. (2023b) studied the 1-dimensional version of this problem, focusing on optimizing the trade-off between space and query time.

**Differential privacy perspective** In this work, we study the

[*]Equal contribution [1]School of Computer Science and Technology, University of Science and Technology of China, Hefei, China. Correspondence to: Pan Peng <ppeng@ustc.edu.cn>.

*Proceedings of the 43^{rd} International Conference on Machine Learning*, Seoul, South Korea. PMLR 306, 2026. Copyright 2026 by the author(s).

range subgraph counting problem from the perspective of *differential privacy (DP)*. DP enables meaningful statistical analysis while ensuring that the presence or absence of any single individual's data has a limited impact on the output (Dwork et al., 2006). We focus on *edge DP*. Two graphs $G$ and $G'$ with the same node set $V$ are said to be *neighboring*, denoted $G \sim G'$, if they differ by exactly one edge. (A stronger notion, known as node DP, has also been studied; see, e.g., (Blocki et al., 2013; Chen & Zhou, 2013; Kasiviswanathan et al., 2013).) Throughout this work, vertex attributes are assumed to be public.

**Definition 1.2** (Edge DP (Nissim et al., 2007))**.** Let $\varepsilon > 0$ and $\delta \in [0, 1)$. A randomized algorithm $\mathcal{A}$ is $(\varepsilon, \delta)$-*DP* if for all measurable events $S$ in the output space of $\mathcal{A}$ and all neighboring graph $G \sim G'$,

$$\Pr[\mathcal{A}(G) \in S] \le e^\varepsilon \Pr[\mathcal{A}(G') \in S] + \delta.$$

When $\delta = 0$, we say that $\mathcal{A}$ satisfies *pure DP* (or $\varepsilon$-DP); when $0 < \delta < 1$, it satisfies *approximate* DP.

Let $f_H(G)$ denote the number of occurrences of $H$ in $G$. A fundamental property of DP implies that any edge-DP algorithm for subgraph counting must incur additive error proportional to the *local sensitivity* $\mathrm{LS}_{f_H}(G)$ with constant probability (Lindell, 2017). Classical DP mechanisms instead depend on the *global sensitivity* $\mathrm{GS}_{f_H}$, which can be significantly larger. Various alternatives to global sensitivity – often via instance-dependent sensitivity bounds – have been proposed, e.g., smooth sensitivity, ladder functions, and private higher-order local sensitivity (Nissim et al., 2007; Zhang et al., 2015; Nguyen et al., 2023). While DP subgraph counting on the entire graph has been studied extensively, private algorithms for *range* subgraph counting have remained unexplored.

The challenge in range subgraph counting is twofold. First, subgraph counts already exhibit high sensitivity, even without range restrictions. Second, range queries require answering subgraph counts over many induced subgraphs $G_q$, leading to increased algorithmic and privacy complexity that depends on the size of the query set $Q$. A naive approach – privately answering each query independently by adding Laplace noise – results in prohibitive error due to sensitivity scaling with $|V_q|$ and privacy loss accumulating via composition. When $|Q| = \Theta(n^{2d})$, the resulting error can overwhelm the true signal, even for simple patterns such as triangles. Moreover, range subgraph counting is inherently *nonlinear*, preventing the direct application of DP techniques developed for linear query workloads.

## 1.1. Our contribution

Before stating our main results, we note that for any algorithm producing *consistent* outputs on identical induced subgraphs $G[V_q]$, we may safely assume that the query set $Q$ satisfies[1] $|Q| = O\big(\min(n^{2d}, 2^n)\big)$.

**Upper bound** We give the first efficient algorithm for differentially private range subgraph counting (DPRSC) that achieves small additive error.

**Theorem 1.3** (Approximate DPRSC)**.** *For any $\varepsilon > 0$ and $\delta \in (0, 1)$, there exists a $(\varepsilon, \delta)$-DP algorithm that, given a graph $G = (V, E, \mathbf{a})$ with $d$-dimensional vertex attributes, a fixed pattern graph $H$, and a query set $Q$, outputs noisy answers $\widetilde{f}_H(G_q)$ which satisfy*

$$\max_{q \in Q} \left| f_H(G_q) - \widetilde{f}_H(G_q) \right|$$
$$= O\left( \frac{\widetilde{\mathrm{HS}}_{f_H}(G) \cdot \sqrt{(\varepsilon + \log(1/\delta)) \log(n|Q|)} \cdot \log^{2d} n}{\varepsilon} \right)$$

*with probability at least $1 - \frac{1}{n}$, where $\widetilde{\mathrm{HS}}_{f_H}$ denotes the output in Algorithm 4. Here, the hidden constants are of the form $c^{O(d)}$ for some universal constant $c > 1$.*

In the above, the quantity $\widetilde{\mathrm{HS}}_{f_H}(G)$ can be viewed as an approximation of $\mathrm{LS}_{f_H}(G)$ with an implicit dependence on $\mathrm{poly}(\log(1/\delta)/\varepsilon)$ (see (Nguyen et al., 2023)). In real-world graphs, which are typically sparse, $\widetilde{\mathrm{HS}}_{f_H}(G)$ is often significantly smaller than $\mathrm{GS}_{f_H}$. For instance, when $H$ is a triangle, if $\varepsilon$ and $\delta$ are constants, we have $\widetilde{\mathrm{HS}}_{f_H}(G) = \mathrm{LS}_{f_H}(G) + O(1) \le d_{\max}(G) + O(1) \ll \mathrm{GS}_{f_H} = n - 2$ with constant probability, where $d_{\max}(G)$ represents the maximum degree of graph $G$ (see Section 3.2 and Appendices C.2 and C.3 for more discussions on $\widetilde{\mathrm{HS}}_{f_H}(G)$ and $\mathrm{GS}_{f_H}$). Note that the number of triangles in many real-world graphs is significantly larger than $n$. Consequently, the additive error of our algorithm is substantially smaller than the true triangle count in such graphs.

For pure DP, which provides a stronger privacy guarantee, we also obtain a corresponding additive error bound of $O(\frac{\mathrm{GS}_{f_H} \cdot \sqrt{\log(n|Q|)} \cdot \log^{3d} n}{\varepsilon})$ (see Theorem 3.3 for details).

We also provide the complexity analysis of the above algorithms in Appendix E.3. In particular, these algorithms run in polynomial time for $d = O(\log n / \log \log n)$. Furthermore, when each dimension of the attributes takes fewer than $n$ distinct values, our algorithms achieve improved additive error as well as better time and space complexity (see Appendix G.4 for details). In the special case where all vertices share the same attribute (so that the only non-trivial query corresponds to counting subgraphs in the entire graph $G$), our bounds are consistent with the results of Nguyen et al. (2023) for DP subgraph counting.

---

[1]Note that the VC-dimension of $Q$ is at most $2d$. By Sauer's Lemma, the number of distinct induced subgraphs that $Q$ can generate is at most $\sum_{i=0}^{2d} \binom{n}{i} = O(\min(n^{2d}, 2^n))$ (Shalev-Shwartz & Ben-David, 2014).

**Lower bound** We also establish a lower bound for the DPRSC problem as the dimension $d$ varies.

**Theorem 1.4** (Lower Bound of DPRSC). *For any pattern graph $H$, let $\mathcal{A}$ be an $(\varepsilon, \delta)$-DP algorithm with constants $\varepsilon > 0$ and $\delta \in [0, \frac{1}{2})$ for the range subgraph counting problem with additive error $\eta = \max_{q \in Q} |f_H(G_q) - \widetilde{f}_H(G_q)|$ with a sufficiently large constant success probability. Assume that $|Q| \geq n^c$ for any sufficiently small constant $c > 0$. Then there exist infinitely many $n$-vertex graphs $G = (V, E, \mathbf{a})$ with $d$-dimensional vertex attributes and a corresponding query set $Q$ such that*

- *if $d = O(1)$, then $\eta = \Omega(\log^{d-1} n \cdot \widetilde{\mathrm{HS}}_{f_H}(G))$;*
- *if $d = O(\log n)$, then $\eta = 2^{\Omega(d)} \cdot \widetilde{\mathrm{HS}}_{f_H}(G)$;*
- *if $d = \Omega(\log n)$, then $\eta = n^{\Omega(1)} \cdot \widetilde{\mathrm{HS}}_{f_H}(G)$.*

The above theorem follows directly as a corollary of the more general lower bound in Theorem 4.1, which additionally captures the dependence on $|Q|$. In particular, for the hard instances underlying Theorem 1.4, the quantity $\widetilde{\mathrm{HS}}_{f_H}(G)$ is, with high constant probability, close to $\mathrm{GS}_{f_H}$. In Appendix F.4, we further provide a lower bound $\Omega(\log n \cdot \widetilde{\mathrm{HS}}_{f_H}(G))$ for the case $d = 1$, under the additional assumption that $\delta = n^{-\Omega(1)}$.

The above results demonstrate the inevitable exponential dependence of the additive error on the dimension $d$ for range subgraph counting when $d = O(\log n)$. In particular, when $d$ is constant, the base of the corresponding exponential dependence is $\log n$, which is consistent with our upper bounds, up to constant factors in the exponent.

**Experiments** We experimentally test our DP algorithms for RSC on real network datasets in Section 5.

**Other extensions** We also explore better upper and lower bounds of DPRSC in higher dimensions $d$ in Appendix H.

We present a simple polynomial-time $\varepsilon$-DP algorithm with an additive error $O(n\sqrt{\log(n|Q|)}\mathrm{GS}_{f_H})$ for any $d$ and constant $\varepsilon$ based on randomized response (see Theorem H.6). The algorithm achieves better additive error and time complexity than Theorem 1.3 and Theorem 3.3 for $d = \Omega(\log n / \log \log n)$. We further extend an approach in Eden et al. (2023) to obtain an instance-dependent error for any $d$ and $H$ (see Theorem H.7). For example, when $H$ is a triangle and $\varepsilon$ is a constant, our algorithm has an additive error $O(\log^3(n|Q|)n^{\frac{3}{2}} + \log(n|Q|)\sqrt{f_{C_4}(G)})$, where $C_4$ represents a 4-cycle. For many real-world graphs, the additive error is significantly better than the aforementioned randomized response method.

Moreover, when $d$ is sufficiently large, we extend the technique in Eliáš et al. (2020) to prove that for any pattern graph $H$ and any $(\varepsilon, \delta)$-DP algorithm with constant $\varepsilon$ and $\delta$, there exist infinitely many $n$-vertex graphs that incur an additive error of $\Omega(n\mathrm{GS}_{f_H})$ with constant probability (see Theorem H.16 for details). Since in Theorem 1.4, the exponent of $n$ multiplied in front of $\widetilde{\mathrm{HS}}_{f_H}(G)$ is always less than 1, this result provides a stronger lower bound than Theorem 1.4 when $d$ is sufficiently large.

## 1.2. Technical overview

A natural approach is to reduce range subgraph counting to a DP range point counting problem by embedding vertices into a Euclidean space, and then applying existing DP algorithms for orthogonal range counting (see, e.g., Dwork et al. (2015)). However, unlike point counting, subgraph counting is inherently nonlinear: the sum of occurrences of a pattern graph in two graphs is not necessarily equal to the number of occurrences in their union, and a single edge change can affect many subgraphs (e.g., one edge may participate in $\Theta(n)$ triangles), leading to high sensitivity.

To address these challenges, we introduce a subgraph projection technique that maps each occurrence of the pattern graph $H$ to a point in $\mathbb{R}^{2d}$ based on vertex attributes, converting range subgraph counting into estimating weighted point counts over axis-aligned rectangles. The weight of each point corresponds to the number of occurrences associated with the underlying vertex tuple. A single edge can change the weights of all points projected by the subgraphs it is associated with. We organize these points using a range tree (Bentley & Saxe, 1978; Dwork et al., 2015) and add Laplace noise to node weights, allowing each query to be answered by aggregating a small number of tree nodes. This approach effectively leverages query correlations and substantially reduces noise. Finally, to further improve accuracy, we adapt the local sensitivity estimation technique of Nguyen et al. (2023) and incorporate it into the range tree, yielding smaller additive error.

To complement our algorithms, we establish lower bounds on the additive error for differentially private range subgraph counting via a reduction from the reconstruction attacks (RA) problem, inspired by the framework of Muthukrishnan & Nikolov (2012). Given an RA instance $\mathbf{x}$, together with a point set $P \subseteq \mathbb{R}^d$ and a query set $Q$ with non-empty common intersection, we construct a $\Theta(n)$-vertex graph $G(\mathbf{x})$ whose private edges encode the entries of $\mathbf{x}$. Any DP algorithm for RSC on $(G(\mathbf{x}), Q)$ can then be transformed into a DP algorithm for RA.

The lower bound follows from two key ingredients: (i) known lower bounds for DP reconstruction attacks, and (ii) the existence of point sets $P$ and query families $Q$ whose incidence matrices have large discrepancy (Lemma 2.10). Our main technical contribution shows that the discrepancy of a matrix associated with the RSC instance $(G(\mathbf{x}), Q)$ is larger by a factor of $\Theta(\mathrm{GS}_{f_H})$ (Lemma 4.3). The proof relies on an approximate reduction and a careful classification

of subgraphs according to the number of private edges they contain. Combining this discrepancy bound with standard connections between discrepancy and differential privacy completes the lower bound argument.

Due to space constraint, we defer the discussion of other related work to Appendix B.

# 2. Preliminaries

Let $G = (V, E, \mathbf{a})$ be an undirected graph with node set $V$ of size $|V| = n$, edge set $E$ of size $|E| = m$, and vertex attribute vector $\mathbf{a} : V \to \mathbb{R}^d$. $H = (V_H, E_H)$ is a pattern graph with $|V_H| = h$ and $|E_H| = m_H$. A subgraph of $G$ isomorphic to $H$ is called an *occurrence* of $H$. We use $f_H(G)$ to represent the number of occurrences of $H$ in $G$. Let $K_n$ denote the complete graph on $n$ vertices. For $\mathbf{x} \in \mathbb{R}^k$, we denote $\|\mathbf{x}\|_1 = \sum_{i \in [k]} |\mathbf{x}_i|$ and $\|\mathbf{x}\|_\infty = \max_{i \in [k]} |\mathbf{x}_i|$.

**Differential privacy** The global sensitivity and local sensitivity of a function are defined as follows.

**Definition 2.1** (Local Sensitivity (Nissim et al., 2007)). For any function $f : \mathcal{X} \to \mathbb{R}^k$ defined over a domain space $\mathcal{X}$ and any $G \in \mathcal{X}$, the *local sensitivity* of the function $f$ at input $G$ is defined as $\mathrm{LS}_f(G) = \max_{G':G \sim G'} \|f(G) - f(G')\|_1$.

**Definition 2.2** (Global Sensitivity (Dwork et al., 2006)). For any function $f : \mathcal{X} \to \mathbb{R}^k$ defined over a domain space $\mathcal{X}$, the *global sensitivity* of the function $f$ is defined as $\mathrm{GS}_f = \max_{G \sim G'} \|f(G) - f(G')\|_1 = \max_G \mathrm{LS}_f(G)$.

We will make use of the following composition theorems of differential privacy.

**Proposition 2.3** (Basic Composition Theorem (Dwork et al., 2006)). *For any $\varepsilon, \delta > 0$, the composition of $k$ $(\varepsilon, \delta)$-differentially private algorithms is $(k\varepsilon, k\delta)$-differentially private.*

**Proposition 2.4** (Advanced Composition Theorem (Kairouz et al., 2015)). *For any $\varepsilon, \delta, \delta' > 0$, the composition of $k$ $(\varepsilon, \delta)$-differentially private algorithms is $(k\varepsilon^2/2 + \varepsilon\sqrt{2k \ln(1/\delta')}, k\delta + \delta')$-differentially private.*

**Laplace distribution and Laplace mechanism** We now introduce the definitions of Laplace distribution and Laplace mechanism.

**Definition 2.5** (Laplace Distribution). We say a zero-mean random variable $X$ follows the *Laplace distribution* with parameter b if the probability density function of $X$ follows $\mathrm{Lap}(b) = \frac{1}{2b} e^{-\frac{|x|}{b}}$.

**Fact 2.6.** *If $Y \sim \mathrm{Lap}(b)$, then $\Pr[|Y| > tb] \leq e^{-t}$.*

**Definition 2.7** (Laplace Mechanism (Dwork et al., 2006)). For any function $f : \mathcal{X} \to \mathbb{R}^k$, the *Laplace mechanism*

on input $x \in \mathcal{X}$ samples $\mathcal{Y}_1, \ldots, \mathcal{Y}_k$ independently from $\mathrm{Lap}(\frac{\mathrm{GS}_f}{\varepsilon})$ and outputs $M(x) = f(x) + (\mathcal{Y}_1, \ldots, \mathcal{Y}_k)$. The Laplace mechanism is $\varepsilon$-DP.

## 2.1. Discrepancy and the point-query incidence matrix

**Definition 2.8.** Given a point set $P \subseteq \mathbb{R}^d$ and a query set $Q$ defined by $m$ axis-parallel boxes $B_1, B_2, \ldots, B_m \subseteq \mathbb{R}^d$, let $\mathbf{A} \in \mathbb{R}^{m \times n}$ be an incidence matrix of the collection of the set $\{B_j \cap P, j \in [m]\}$ (i.e. a matrix whose rows are the indicator vectors of $\{B_j \cap P, j \in [m]\}$).

**Definition 2.9.** For a matrix $\mathbf{M} \in \mathbb{R}^{m \times n}$ and a set $C \subseteq \{0, \pm 1\}^n$, the discrepancy of the matrix $\mathbf{M}$ is defined as $\mathrm{disc}_C(\mathbf{M}) = \min_{\chi \in C} \|\mathbf{M}\chi\|_\infty$.

For any $\alpha = \Omega(1)$, let $C_\alpha$ be the set of all vectors $\chi \in \{0, \pm 1\}^n$ satisfying $\|\chi\|_1 \geq \alpha n$. We have following result, whose proof idea follows from Chazelle & Lvov (2001); Muthukrishnan & Nikolov (2012); Matoušek & Nikolov (2015) and we defer the proof to Appendix D.1.

**Lemma 2.10.** *For any $m$ large enough, there exists a set of $n = \Theta(m)$ points $P \subseteq \mathbb{R}^d$ and a query set $Q$ defined by $m$ axis-parallel boxes $B_1, B_2, \ldots, B_m \subseteq \mathbb{R}^d$, such that these boxes have a non-empty common intersection and the following holds. Let $\mathbf{A}$ denote the incidence matrix of the collection of set $\{B_j \cap P, j \in [m]\}$. For any $\alpha = \Omega(1)$,*

- *if $d = O(1)$, then $\mathrm{disc}_{C_\alpha}(\mathbf{A}) = \Omega(\log^{d-1} n)$;*
- *if $d = O(\log n)$, then $\mathrm{disc}_{C_\alpha}(\mathbf{A}) = 2^{\Omega(d)}$;*
- *if $d = \Omega(\log n)$, then $\mathrm{disc}_{C_\alpha}(\mathbf{A}) = n^{\Omega(1)}$.*

# 3. The Upper Bound

Our approximate DP algorithm in Theorem 1.3 is based on a pure DP algorithm, which we introduce below.

## 3.1. The pure DP algorithm

The high level description of the algorithm is as follows: (1) Find all occurrences of subgraph $H$ in $G$ and map each to a *rank tuple* (see below) in $[n]^{2d}$. Project each occurrence into its rank tuple to obtain a weight vector $\mathbf{w}$ of length $n^{2d}$ (see Algorithm 1). (2) Treat the rank tuples weighted points in $[n]^{2d}$ and build a differentially private $2d$D range tree over them (see Algorithm 2). (3) Answer each query by traversing the range tree and aggregating the noisy weights of the corresponding nodes (see Algorithm 3).

Now we describe our algorithm in more detail.

**Subgraph counting projection** Algorithm 1 takes as input an $n$-vertex graph $G = (V, E, \mathbf{a})$. For each $i \in [d]$, let $A_i(V, \mathbf{a})$ be the set of the $i$-th attributes for all vertices in $V$. Let $s_i(v) \in [n]$ represent the rank of $\mathbf{a}_i(v)$ in $A_i(V, \mathbf{a})$ for some vertex $v$, when the $i$-th attributes in $A_i(V, \mathbf{a})$ are sorted in ascending order. Note that we have $s_i(u) = s_i(v)$

**Algorithm 1** PROJ $(G = (V, E, \mathbf{a}), H)$    ▷ Subgraph Counting Projection

1: **Input:** An $n$-vertex graph $G = (V, E, \mathbf{a})$.
2: For each $i \in [d]$, sort the $i$-th attributes in $A_i(V, \mathbf{a})$, and then define the rank function $s_i : V \to [n]$.
3: Initialize $w_{\mathbf{v}} = 0$, for any $\mathbf{v} \in [n]^{2d}$.
4: **for all** occurrences of subgraph $H$ in $G$ **do**
5:      Compute $w_{(s_i(u_i), s_i(v_i))_{i=1}^d} = w_{(s_i(u_i), s_i(v_i))_{i=1}^d} + 1$, where $u_i$ (resp. $v_i$) be the vertex in this occurrence with the smallest (resp. largest) rank in dimension $i$.
6: **return** $\mathbf{w}$.

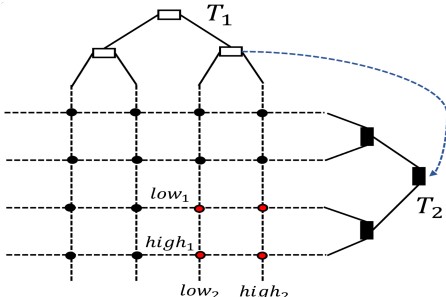

*Figure 1.* Schematic diagram of the $2d$D range tree $T_1$ for $d = 1$. The tree construction process recursively divides $[n]^2$ into two equal parts, where each tree node in $T_1$ is associated with a 1D range tree (e.g., $T_2$), and each tree node in a 1D range tree stores the corresponding weight sum (see Appendix C.1 for more details).

if two vertices $u$ and $v$ have the same $i$-th attribute. Now we present the definition of the rank tuple of an occurrence.

**Definition 3.1** (Rank Tuple). The rank tuple of an occurrence $H' = (V(H'), E(H'))$ is defined as a $2d$-tuple $(s_i(u_i), s_i(v_i))_{i=1}^d$ satisfying $u_i = \arg\min_{v \in V(H')} s_i(v)$ and $v_i = \arg\max_{v \in V(H')} s_i(v)$ for all $i \in [d]$.

For each occurrence of a subgraph $H$ in $G$, it updates a weight vector $\mathbf{w}$ at the position corresponding to its rank tuple. Each position in $\mathbf{w}$ records the number of occurrences that share a specific rank tuple.

**DP range tree construction** Algorithm 2 takes as input the project vector $\mathbf{w}$ from Algorithm 1. Then the algorithm treats each rank tuple $\mathbf{v}$ in $[n]^{2d}$ as a point with weight $\mathbf{w_v}$ and utilizes a $2d$D range tree (see Appendix C.1) to preprocess these points.

The schematic diagram of the $2d$D range tree, with $d = 1$ as an example, is shown in Figure 1. In the $2d$D range tree, each leaf node of every 1D range tree corresponds to a point in $[n]^{2d}$, and its weight is set to the weight of the corresponding point. The weight of each internal node of every 1D range tree is set to the sum of the weights of its children. We add Laplace noise to the weights of nodes in all 1D range trees to ensure privacy.

**Algorithm 2** TREECONST$(\mathbf{w}, \varepsilon')$    ▷ DP Range Tree Construction

1: **Input:** Projection vector $\mathbf{w}$, and privacy parameter $\varepsilon'$.
2: Construct a $2d$D range tree $T_1$ according to Definition C.5 using a set of points $\{(\mathbf{v}, w_{\mathbf{v}})\}_{\mathbf{v} \in [n]^{2d}}$.
3: Create a noisy version $\widetilde{T}_1$ by adding Laplace noise to the weights of nodes in all 1D range trees within $T_1$. Specifically, update the weight as weight $=$ weight $+$ Lap$(1/\varepsilon')$.
4: **return** $\widetilde{T}_1$.

**Algorithm 3** PDP_RSC$(G, H, Q, \varepsilon)$    ▷ Pure DP Range Subgraph Counting

1: **Input:** An $n$-vertex graph $G = (V, E, \mathbf{a})$, a pattern graph $H$, a query set $Q$, and privacy parameter $\varepsilon$.
2: Set $\mathrm{GS}_{f_H} = f_H(K_n) - f_H(K_n - e)$.
3: Set $\mathbf{w} = \mathrm{PROJ}(G, H)$ and $J_Q = \emptyset$.
4: Set $\widetilde{T}_1 = \mathrm{TREECONST}(\mathbf{w}, \frac{\varepsilon}{\mathrm{GS}_{f_H} \cdot (\lceil \log n \rceil + 1)^{2d}})$.
5: **for** each query $q \in Q$ **do**
6:      Determine $\prod_{i=1}^d [\ell_i, r_i]$ according to Definition 3.2.
7:      Add the query result of $\widetilde{T}_1$ (see Definition C.6) using the range $\prod_{i=1}^d ([\ell_i, n] \times [1, r_i])$ to $J_Q$.
8: **return** $J_Q$.

**Query procedure** For any query $q = \prod_{i=1}^d [\ell_i, r_i] \in Q$, we first apply the discretization described below to obtain a new query $q'$. Note that the ranges $q$ and $q'$ correspond to the same induced subgraph. For simplicity, we will use $q = \prod_{i=1}^d [\ell_i, r_i]$ to refer to the discretized query.

**Definition 3.2** (Discretization). Let $q = \prod_{i=1}^d [\ell_i, r_i]$ be a query. For each $[l_i, r_i]$, we associate it with two vertices $v_{\ell_i}$ and $v_{r_i}$, where $v_{\ell_i} = \arg\min_{v \in V, \mathbf{a}_i(v) \geq \ell_i} \mathbf{a}_i(v)$ and $v_{r_i} = \arg\max_{v \in V, \mathbf{a}_i(v) \leq r_i} \mathbf{a}_i(v)$. The discretized query $q'$ is defined as $\prod_{i=1}^d [s_i(v_{\ell_i}), s_i(v_{r_i})]$.

For a discretized query $q = \prod_{i=1}^d [\ell_i, r_i]$, we call the $2d$D range tree query with range $\prod_{i=1}^d ([\ell_i, n] \times [1, r_i])$, which ensures that all subgraphs exactly falling within the range are counted in the answer.

The complete algorithm is presented in Algorithm 3, with its analysis deferred to Appendix E.1. To improve time efficiency, the implementation of Algorithm 3 differs slightly from the pseudocode without affecting correctness (see Appendix E.3). As a result, we obtain the following theorem.

**Theorem 3.3** (Pure DPRSC). *For any $\varepsilon > 0$, there exists an $\varepsilon$-DP algorithm that, given a graph $G = (V, E, \mathbf{a})$ with $d$-dimensional vertex attributes, a fixed pattern graph $H$, and a query set $Q$, outputs noisy answers $\widetilde{f}_H(G_q)$ which satisfy $\max_{q \in Q} \left| f_H(G_q) - \widetilde{f}_H(G_q) \right| =$*

**Algorithm 4** ESTIMATEHS$(G, H, \varepsilon', \delta')$    ▷ Estimating private higher-order local sensitivity (Algorithm 5 in Nguyen et al. (2023))

1: **Input**: An $n$-vertex graph $G$, a pattern graph $H$ with $m_H$ edges, privacy parameters $\varepsilon', \delta'$.
2: Set $\widetilde{\mathrm{HS}}_{f_H}^{(m_H)} = f_H^{(m_H)}(G)$.
3: **for** $k = m_H - 1$ down to 1 **do**
4:   Set $\widetilde{\mathrm{HS}}_{f_H}^{(k)}(G) = f_H^{(k)}(G) + \widetilde{\mathrm{HS}}_{f_H}^{(k+1)}(G)\frac{\ln(1/\delta')}{\varepsilon'} +$ $\mathrm{Lap}(\widetilde{\mathrm{HS}}_{f_H}^{(k+1)}(G)/\varepsilon')$.
5: **return** $\widetilde{\mathrm{HS}}_{f_H}(G) = \widetilde{\mathrm{HS}}_{f_H}^{(1)}(G)$.

$O\left(\frac{\mathrm{GS}_{f_H} \cdot \sqrt{\log(n|Q|)} \cdot \log^{3d} n}{\varepsilon}\right)$, *with probability* $\geq 1 - \frac{1}{n}$.
*Here, the hidden constants are of the form $c^{O(d)}$ for some universal constant $c > 1$.*

### 3.2. The approximate DP algorithm

Now we present our approximate DP algorithm. We begin by introducing some notation. Let $S \subseteq V \times V$ be a set of vertex pairs. Let $f_H(G, S)$ denote the number of occurrences of a fixed pattern graph $H$ containing all edges in $S$ in the graph $(V(G), E(G) \cup S)$. We define $f_H^{(k)}(G) = \max_{|S|=k} f_H(G, S)$. Note that we have $f_H^{(1)}(G) = \mathrm{LS}_{f_H}(G)$ by definition.

The local sensitivity of $\mathbf{w}$ is estimated by iteratively adding noise to ensure privacy (see Algorithm 4). The approach is derived from Nguyen et al. (2023), whose method was originally designed to estimate subgraph counts but has been modified here to estimate the local sensitivity.

The complete approximate DP algorithm is presented in Algorithm 5. It closely resembles the pure DP one (i.e. Algorithm 3), with the key difference being in the parameter settings. The algorithm adds noise in the range tree based on the private estimation of local sensitivity rather than global sensitivity. The analysis of this algorithm (i.e., proof of Theorem 1.3) and the implementation details are deferred to Appendices E.2 and E.3, respectively.

## 4. The Lower Bound

In this section, we prove the following theorem which presents a lower bound for DPRSC.

**Theorem 4.1** (Lower Bound of DPRSC). *For any pattern graph $H$, let $\mathcal{A}$ be an $(\varepsilon, \delta)$-DP algorithm with constants $\varepsilon > 0$ and $\delta \in [0, \frac{1}{2})$ for the range subgraph counting problem with additive error $\eta = \max_{q \in Q} |f_H(G_q) - \widetilde{f}_H(G_q)|$ with a sufficiently large constant success probability.*

*Then there exist infinitely many $n$-vertex graphs $G =$*

**Algorithm 5** ADP_RSC $(G, H, Q, \varepsilon, \delta)$    ▷ Approximate DP Range Subgraph Counting

1: **Input:** An $n$-vertex graph $G = (V, E, \mathbf{a})$, a pattern graph $H$ with $m_H$ edges, a query set $Q$, and privacy parameters $\varepsilon, \delta$.
2: Set $\varepsilon'$ and $\delta'$ such that $\varepsilon = (m_H + 1)\varepsilon'$ and $\delta = \max(2e^{m_H \varepsilon'} + m_H e^{\varepsilon'} + 1, m_H e^{2\varepsilon'} + e^{\varepsilon'} + 2)\delta'$.
3: Set $\widetilde{\mathrm{HS}}_{f_H}(G) = $ ESTIMATEHS$(G, H, \varepsilon', \delta')$.
4: Set $\mathbf{w} = $ PROJ$(G, H)$ and $J_Q = \emptyset$.
5: Set $\delta'' = \min(e^{-\varepsilon'/8}, \delta')$ and $\varepsilon'' = \varepsilon'/(\widetilde{\mathrm{HS}}_{f_H}(G) \cdot (\lceil \log n \rceil + 1)^d \cdot 2\sqrt{2\ln(1/\delta'')})$.
6: Set $\widetilde{T}_1 = $ TREECONST $(\mathbf{w}, \varepsilon'')$.
7: **for** each query $q \in Q$ **do**
8:   Determine $\prod_{i=1}^{d}[\ell_i, r_i]$ according to Definition 3.2.
9:   Add the query result of $\widetilde{T}_1$ (see Definition C.6) using the range $\prod_{i=1}^{d}([\ell_i, n] \times [1, r_i])$ to $J_Q$.
10: **return** $J_Q$.

$(V, E, \mathbf{a})$ *with $d$-dimensional vertex attributes and a corresponding query set $Q$, such that*

- *if $d = O(1)$, then $\eta = \Omega(\log^{d-1} \bar{n} \cdot \mathrm{GS}_{f_H})$;*
- *if $d = O(\log \bar{n})$, then $\eta = 2^{\Omega(d)}\mathrm{GS}_{f_H}$;*
- *if $d = \Omega(\log \bar{n})$, then $\eta = \bar{n}^{\Omega(1)}\mathrm{GS}_{f_H}$;*

*where $\bar{n} = \min(n, |Q|)$ can be arbitrarily large.*

We prove Theorem 4.1 via a reduction from the reconstruction attacks (RA) problem to DPRSC, using techniques inspired by Muthukrishnan & Nikolov (2012). Moreover, the proof reveals that for the constructed hard instances, the approximated local sensitivity $\widetilde{\mathrm{HS}}_{f_H}(G)$ is, with high probability, within a constant factor of the global sensitivity $\mathrm{GS}_{f_H}$. This directly implies Theorem 1.4, whose full proof is deferred to Appendix F.3.

In Appendix F.4, we further provide a lower bound of $\eta = \Omega(\log \bar{n} \cdot \mathrm{GS}_{f_H})$ for the case $d = 1$, under the additional assumption that $\delta = n^{-\Omega(1)}$.

### 4.1. The reduction

We first introduce the reconstruction attacks problem and then reduce it to range subgraph counting.

**Definition 4.2** (Reconstruction Attacks (RA); (Dinur & Nissim, 2003)). Given a private vector $\mathbf{x} \in \{0, 1\}^N$, a reconstruction algorithm outputs a vector in $\{0, 1\}^N$. The goal is to design an $(\varepsilon, \delta)$-DP algorithm $\mathcal{B}$ minimizing $\|\mathcal{B}(\mathbf{x}) - \mathbf{x}\|_1$.

**From RA to RSC** Let $P \subseteq \mathbb{R}^d$ be a point set of size $n$ and let $Q$ be a collection of $m$ axis-parallel boxes $B_1, \ldots, B_m$ with a non-empty common intersection.

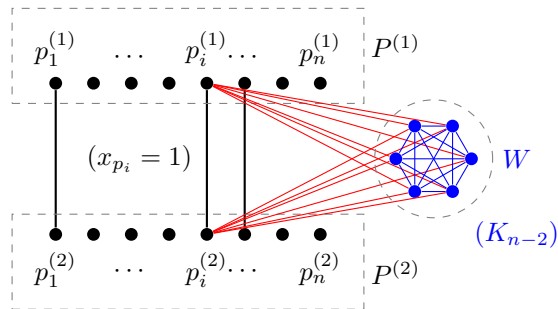

*Figure 2.* The construction of the graph $G(\mathbf{x})$. $P^{(1)}$ and $P^{(2)}$ contain $n$ vertices respectively, while $W$ contain $n-2$ vertices. We omit the edges from the set $P^{(1)} \cup P^{(2)}$ to $W$ except those incident to the vertices $p_i^{(1)}$ and $p_i^{(2)}$ (red edges). The vertices $p_i^{(1)}$ and $p_i^{(2)}$, together with those in $W$, induce a $n$-vertex clique.

**Graph construction** We construct a graph $G(\mathbf{x})$ as follows. Create two copies of $P$, denoted $P^{(1)}$ and $P^{(2)}$. For each $p \in P$, let $p^{(1)} \in P^{(1)}$ and $p^{(2)} \in P^{(2)}$ denote its copies. Let $W$ be a set of $n-2$ auxiliary vertices, all lying in the common intersection of the boxes in $Q$.

The vertex set of $G(\mathbf{x})$ is $V = P^{(1)} \cup P^{(2)} \cup W$, so that $|V| = 3n-2$ (see Figure 2). For each $p \in P$, we add an edge between $p^{(1)}$ and $p^{(2)}$ if and only if $x_p = 1$. In addition, every vertex in $P^{(1)} \cup P^{(2)}$ is connected to every vertex in $W$, and $W$ induces a clique.

The edges between $p^{(1)}$ and $p^{(2)}$ encode the private data $\mathbf{x}$, while all other edges are public. Consequently, two vectors $\mathbf{x}, \mathbf{x}'$ are neighboring if and only if the corresponding graphs $G(\mathbf{x})$ and $G(\mathbf{x}')$ are neighboring.

**Hard instance** We begin with a point set of size $\bar{n}$ and a query set $Q$ from Lemma 2.10. We then add $n - \bar{n}$ dummy points lying outside all boxes in $Q$ to form $P$. Using this point set $P$ and query set $Q$, we construct the graph $G(\mathbf{x})$ as described above and use $Q$ as the query set for the RSC problem. We denote the resulting instance by $(G(\mathbf{x}), Q)$, which completes the construction of the hard instance.

### 4.2. The analysis

To prove the lower bound using the constructed instance $(G(\mathbf{x}), Q)$, we associate it with a matrix $\mathbf{A}_H$ and show that this matrix has large discrepancy.

**The matrix** The matrix $\mathbf{A}_H$ is defined with respect to the complete graph $K_{3n-2}$ on vertex set $V = P^{(1)} \cup P^{(2)} \cup W$. It has $|Q|$ rows and $f_H(K_{3n-2})$ columns. Each row corresponds to a query $q \in Q$, i.e., the induced subgraph $K_{3n-2}[V_q]$, and each column corresponds to an occurrence $H'$ of the pattern $H$ in $K_{3n-2}$. The entries are defined as

$$(A_H)_{q,H'} = \begin{cases} 1, & \text{if } E(H') \subseteq E(K_{3n-2}[V_q]); \\ 0, & \text{otherwise.} \end{cases}$$

*Table 1.* Summary of real-world datasets used in our experiments (see (Newman, 2006; Cho et al., 2014; Rozemberczki et al., 2019)). Here $n$, $m$, $d_{\mathrm{avg}}$, and $d_{\mathrm{max}}$ denote the number of nodes, edges, average degree, and maximum degree, respectively.

| Dataset | $n$ | $m$ | $d_{\mathrm{avg}}$ | $d_{\mathrm{max}}$ |
|---|---|---|---|---|
| CA-Netscience | 379 | 914 | 4.8 | 34 |
| Wiki-Squirrel | 5,201 | 198,353 | 76.3 | 1,903 |
| WormNet-v3 | 16,347 | 762,822 | 93.3 | 1,272 |

Note that $\mathbf{A}_H$ is fixed and depends only on $K_{3n-2}$ and the query set $Q$, and is independent of the edge set of $G(\mathbf{x})$.

Let $\mathbf{x}_H \in \{0,1\}^{f_H(K_{3n-2})}$ be the indicator vector whose coordinates correspond to occurrences $H'$ of $H$ in $K_{3n-2}$, where $(\mathbf{x}_H)_{H'} = 1$ if all edges of $H'$ are present in $G(\mathbf{x})$, and 0 otherwise. Then for each query $q \in Q$, the quantity $(\mathbf{A}_H \mathbf{x}_H)_q$ equals the number of occurrences of $H$ in the induced subgraph $(G(\mathbf{x}))[V_q]$.

Let $C_{\alpha,H}$ denote the set of vectors $\chi = \mathbf{x}_H - \mathbf{x}'_H \in \{0, \pm 1\}^{f_H(K_{3n-2})}$ such that $\mathbf{x} - \mathbf{x}' \in C_\alpha$. The following lemma, proved in Appendix F.1, shows that $\mathbf{A}_H$ inherits large discrepancy from the point-incidence matrix.

**Lemma 4.3.** *Let $P \subseteq \mathbb{R}^d$ be a set of $n$ points, and let $Q$ be a collection of $m$ sets $B_1, \ldots, B_m \subseteq \mathbb{R}^d$ with a non-empty common intersection. Let $\mathbf{A}$ be the incidence matrix of $\{B_j \cap P : j \in [m]\}$. For any $\alpha = \Omega(1)$ and any pattern graph $H$, if $\mathrm{disc}_{C_\alpha}(\mathbf{A}) = \omega(1)$, then $\mathrm{disc}_{C_{\alpha,H}}(\mathbf{A}_H) = \Omega(\mathrm{GS}_{f_H} \cdot \mathrm{disc}_{C_\alpha}(\mathbf{A}))$, where $\mathrm{GS}_{f_H}$ denotes the global sensitivity of $f_H$ on $n$-vertex graphs.*

**Proof sketch of Theorem 4.1** Given the above reduction, we show that if there exists a DP algorithm for RSC whose additive error is asymptotically smaller than $\mathrm{disc}_{C_{\alpha,H}}(\mathbf{A}_H)$, then this algorithm can be used to solve RA with small error (Lemma F.2). This contradicts known lower bounds for RA (Lemma F.3), and thus yields the desired lower bound for DPRSC. We defer the full argument to Appendix F.2.

## 5. Experiments

Now we present our experimental results. All algorithms were implemented in Python[2], and run on an Intel(R) Xeon(R) Platinum 8562Y Processor @ 2.80 GHZ with 768 GB RAM. We evaluated our methods on three real-world datasets, after removing self-loops and duplicate edges from each graph. Basic statistics for these datasets are summarized in Table 1. The edges are treated as sensitive data.

**Baseline** There is no prior work on DP range subgraph counting. We construct two natural baselines for comparison. The first baseline **PDP_Comp** answers each query

---

[2]The corresponding codes and data are available at https://github.com/Airleave/DPRSC.

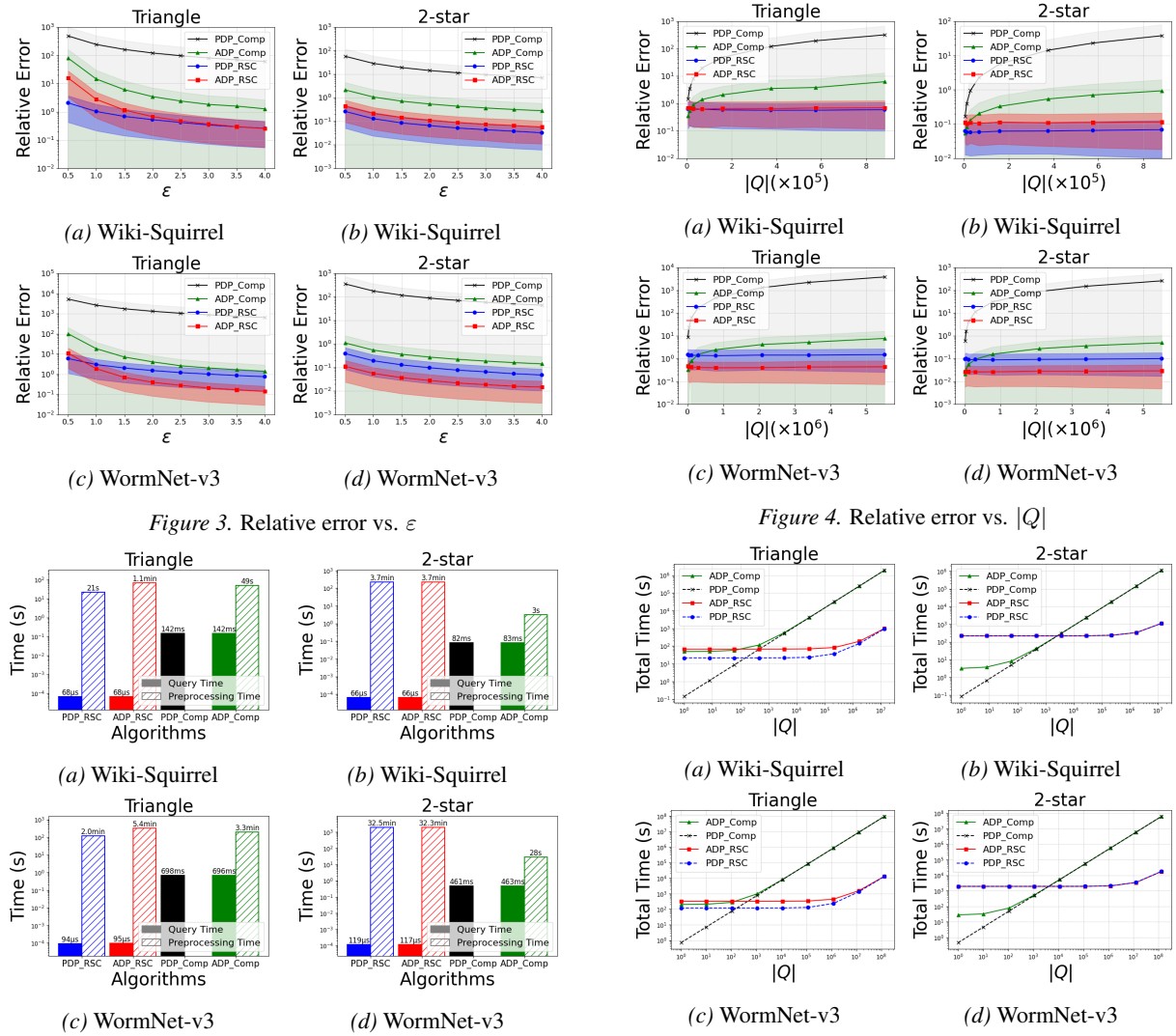

*Figure 3.* Relative error vs. $\varepsilon$

*Figure 4.* Relative error vs. $|Q|$

*Figure 5.* Query time and preprocessing time

*Figure 6.* Total time

independently by adding Laplace noise with scale $\frac{\mathrm{GS}_{f_H} \cdot |Q|}{\varepsilon}$, using the basic composition theorem (Proposition 2.3). The second baseline **ADP_Comp** improves upon this by applying Lemma E.5 and the advanced composition theorem (Proposition 2.4), adding Laplace noise with scale $\Theta\left(\frac{\widetilde{\mathrm{HS}}_{f_H}(G) \cdot \sqrt{|Q| \ln(1/\delta)}}{\varepsilon}\right)$. We denote our pure DP algorithm (Algorithm 3) and approximate DP algorithm (Algorithm 5) by **PDP_RSC** and **ADP_RSC**, respectively. The comparison of different algorithms is provided in Table 2.

**Setting and metric** All algorithms satisfy either $\varepsilon$-DP or $(\varepsilon, \delta)$-DP. We set $\varepsilon = 2.0, \delta = 10^{-5}$ (which satisfies $\delta < \frac{1}{n}$ for all datasets considered), $d = 1$ and $|Q| = \lceil n^{1.5} \rceil$ by default. Each query $q$ in $Q$ is generated randomly and corresponds to an induced vertex set $V_q$ with $|V_q| = \Theta(n)$. Experiments are conducted on three datasets, with $H$ being a triangle, 2-star, or edge. Results for edge and CA-Netscience

*Table 2.* The performance guarantees of DP algorithms for counting occurrences of $H$. We assume that $\varepsilon, \delta, d$ are constants and $|Q|$ is a polynomial in $n$. The notation $\widetilde{O}(\cdot)$ hides factors that are polylogarithmic in $n$. The additive errors hold with probability at least $1 - \frac{1}{n}$ and are given by Fact 2.6, Theorems 1.3 and 3.3.

| Algorithm | DP | Additive Error |
|---|---|---|
| PDP_Comp | $\varepsilon$ | $\widetilde{O}(\mathrm{GS}_{f_H} \cdot |Q|)$ |
| ADP_Comp | $(\varepsilon, \delta)$ | $\widetilde{O}(\widetilde{\mathrm{HS}}_{f_H}(G) \cdot \sqrt{|Q|})$ |
| **PDP_RSC** | $\varepsilon$ | $\widetilde{O}(\mathbf{GS}_{f_H})$ |
| **ADP_RSC** | $(\varepsilon, \delta)$ | $\widetilde{O}(\widetilde{\mathbf{HS}}_{f_H}(G))$ |

are deferred to Appendices G.1 and G.2, respectively. For each vertex in these networks, we independently sample attributes from a standard normal distribution as a conservative (worst-case) evaluation; the rationale for this choice

is provided in Appendix G.4. We also include additional experiments to complement the default setting: Results for $d > 1$, small-range queries and real attributes are provided in Appendices G.2 to G.4, respectively. We measure relative error as $\frac{|\widetilde{f}_H(G_q) - f_H(G_q)|}{\max(f_H(G_q), 0.001n)}$, following standard practice (Imola et al., 2021). Each algorithm is run at least 20 times; we report mean relative error and runtime, with standard-deviation bands shown in figures. We define the *query time* as the time required to answer a single query, the *total query time* as the sum of the query times over all queries, and the *total time* as the sum of the preprocessing time and the total query time.

**Relative error vs. $\varepsilon$** Figure 3 shows that our algorithms consistently outperform the baselines across all tested privacy parameter $\varepsilon$ (smaller values of $\varepsilon$ correspond to stronger privacy guarantees), achieving substantially lower mean error and variance. The approximate DP algorithm ADP_RSC generally performs best on sparse graphs, except for 2-stars on Wiki-Squirrel, where large $d_{\max}$ increases the local sensitivity and noise relative to PDP_RSC, since $\widetilde{\mathrm{HS}}_{f_H}(G) \approx 2d_{\max}$ for 2-stars (Appendix C.3). For triangles, ADP_RSC is more sensitive to small $\varepsilon$ since estimating $\widetilde{\mathrm{HS}}_{f_H}$ requires iteratively adding noise related to $\varepsilon$ and $\delta$.

**Relative error vs. $|Q|$** As shown in Figure 4, our algorithms are largely insensitive to $|Q|$, whereas baseline errors increase quickly with $|Q|$. A minor exception occurs for very small $|Q| < 2n$ in the ADP_Comp vs. ADP_RSC comparison, consistent with the theoretical bounds in Theorem 1.3. The performance gap grows as $|Q|$ increases.

**Runtime** For runtime evaluation, we assume that the queries in $Q$ are i.i.d. and uniformly sampled from all queries that produce distinct induced subgraphs. We evaluated the relationship between the total time and $|Q|$ in Figure 6. $|Q|$ ranges from 1 to $\Theta(n^2)$, which corresponds to the full range of $|Q|$ under our setting when $d = 1$. Since exhaustively evaluating all $\Theta(n^2)$ queries is computationally prohibitive, we sample a sufficiently large number of queries from the distribution defined above. We continue sampling until the average query time converges, as measured by a small relative standard error (below 5%), and use the average query time and preprocessing time (shown in Figure 5) to estimate the total time. Given the substantial differences in query time between our algorithms and the baselines, the resulting estimation error is negligible. As shown in Figure 5, our algorithms significantly outperform the baselines in query time, achieving lower latency by 3 to 4 orders of magnitude at the cost of higher preprocessing time. Note that PDP_comp requires no preprocessing. Here, the baselines compute exact counts of 2-stars using a straightforward implementation in $O(m)$ time, while exact triangle counting is carried out using the widely adopted algorithm from Chiba & Nishizeki (1985). Figure 6 shows that our algorithm

consistently outperforms the baselines in total time when $|Q| = \Omega(n)$, and the advantage becomes increasingly pronounced as $|Q|$ grows. Moreover, the advantage is more significant on larger datasets. Overall, when $Q = \Theta(n^2)$, the total time is dominated by the total query time, yielding a speedup of 3 to 4 orders of magnitude.

# 6. Conclusion

In this paper, we study the DPRSC problem and present the first efficient algorithms achieving small additive error. We complement these results with lower bounds showing an exponential dependence on the attribute dimension, nearly matching our upper bounds.

A limitation of our work is that our current algorithms do not address settings in which attribute information is sensitive and must be protected. Combining our techniques with existing methods (e.g., (Dwork et al., 2015)) may yield upper and lower bounds in this setting. However, achieving general algorithms with optimal utility guarantees would likely require new ideas beyond our current approach. In addition, a gap remains between our upper and lower bounds in certain parameter regimes, leaving room for improvement in both analysis and algorithm design. Another interesting direction is to extend the RSC problem to more challenging settings, such as dynamic graphs where edges or attributes evolve over time, and alternative privacy models including local differential privacy and node-level differential privacy.

## Acknowledgments

This work is supported in part by NSFC Grant 62272431 and Quantum Science and Technology - National Science and Technology Major Project (Grant No. 2021ZD0302901).

## Impact Statement

This paper presents work whose goal is to advance the field of Machine Learning and Algorithmic Privacy. There are many potential societal consequences of our work, none which we feel must be specifically highlighted here.

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

## A. Overview of the Appendix

The appendix is organized as follows:

## B. Other Related Work

**DP subgraph counting** The DP subgraph counting problem is a topic that has been extensively studied, primarily for the entire graph $G$. Nissim et al. (2007) improved the utility guarantees for triangle counting in differential privacy by incorporating instance-dependent noise. Karwa et al. (2011) extended the smooth sensitivity approach to $k$-stars and proposed methods for computing local sensitivity to perform $k$-triangle counting. Kasiviswanathan et al. (2013) introduced a triangle counting algorithm under the node-DP framework. Zhang et al. (2015) developed ladder functions for various subgraph counting tasks. Nguyen et al. (2023) focused on optimizing runtime by calculating approximate smooth sensitivity for graphs with certain properties, achieving both privacy and utility while reducing time complexity. Additionally, several studies have examined subgraph counting under the local DP model, such as Imola et al. (2021; 2022a;b); Eden et al. (2023), while Fichtenberger et al. (2021) studied DP subgraph counting in dynamic model.

**DP range queries** Muthukrishnan & Nikolov (2012) present algorithms for the halfspace range counting problem under differential privacy, achieving good approximate accuracy in terms of average squared error. Deng et al. (2023a) propose an algorithm for counting queries and bottleneck queries on shortest paths while ensuring differential privacy. A closer examination of their model reveals that they effectively address a range counting problem on a graph. A cut query on a graph is a specialized form of range counting, where the range space includes all possible cuts. The cut query problem is widely studied in the field of differential privacy, with significant research dedicated to it (Gupta et al., 2010; Blocki et al., 2012; Gupta et al., 2012; Arora & Upadhyay, 2019; Eliáš et al., 2020; Dalirrooyfard et al., 2024).

**DP lower bound** DP lower bounds are often motivated by reconstruction attacks (Dinur & Nissim, 2003), which show that answering too many queries accurately can leak sensitive information about the dataset. Building on this technique, Muthukrishnan & Nikolov (2012) explore the connections between discrepancy theory and DP lower bound. In particular, they establish the first lower bound on the average squared error for the private range counting problem. Subsequent improvements in discrepancy results by Matoušek & Nikolov (2015) further strengthen the lower bound. Furthermore, based on the connections, Eliáš et al. (2020) studied the lower bound for DP synthetic graph generation to approximate all cuts, while Peng & Xu (2025) extended this to triangle-motif cuts. Their approaches heavily rely on the randomness inherent in Erdős–Rényi graphs, which limits its applicability to range subgraph counting for moderate $d$.

## C. Supplementary Tools

We will make use of the following post-processing property of differential privacy.

**Proposition C.1** (Post-processing Property (Dwork et al., 2006)). *Let $M\colon \mathbb{R}^{d_1} \to \mathbb{R}^{d_2}$ be an $(\varepsilon, \delta)$-differentially private mechanism and let $h\colon \mathbb{R}^{d_2} \to \mathbb{R}^{d_3}$ be an arbitrary function. Then, the function $g \circ M\colon \mathbb{R}^{d_1} \to \mathbb{R}^{d_3}$ is also $(\varepsilon, \delta)$-differentially private.*

The sum of multiple variables that follow the Laplace distribution satisfies the following properties.

**Lemma C.2** (Chan et al. (2011); Wainwright (2019)). *Let $\{X_i\}$ be a collection of independent random variables such that $X_i \sim \mathrm{Lap}(b_i)$ for all $1 \leq i \leq m$. Then, for $\nu \geq \sqrt{\sum_i b_i^2}$ and $0 < \lambda < \frac{2\sqrt{2}\nu^2}{b}$ for $b = \max_i\{b_i\}$, $\Pr\left[|\sum_i X_i| \geq \lambda\right] \leq 2 \cdot \exp(-\frac{\lambda^2}{8\nu^2})$. Furthermore, if $b = b_i$ for any $i \in [m]$ and $m \geq \log \beta$, we have $\Pr\left[|\sum_i X_i| \geq 2\sqrt{2} \cdot b\sqrt{m \log \beta}\right] \leq \frac{2}{\beta}$.*

### C.1. Definitions and properties of range tree

A range tree is a binary tree structure designed for interval queries and summation. For clarity, we define the tree construction and query process to streamline the algorithm's description. The construction of the range tree is based primarily on Bentley & Saxe (1978), with minor modifications.

We begin by introducing the basic 1D range tree.

**Definition C.3** (1D Range Tree). Given a set of points $P = \{(x_t, w_t)\}$, where each point has an $x$-coordinate $x_t$ and a weight $w_t$, the 1D range tree is constructed as follows:

1. Sort the points by $x$-coordinates, denoted as $x_1, \ldots, x_n$.
2. Begin building the tree recursively from the root node, where the interval spans from $x_1$ to $x_n$.
3. For a given interval $x_l, \ldots, x_r$ corresponding to a tree node $p$, set $mid = \lfloor \frac{l+r}{2} \rfloor$. Recursively construct the left child using points $x_l, \ldots, x_{mid}$ and the right child using points $x_{mid+1}, \ldots, x_r$. If the interval contains only one kind of coordinate, terminate the recursion and set the weight of the current tree node to the sum of the weights of the points at the coordinate.
4. During backtracking, sum the weights of the left child and the right child as the weight of the current tree node:

$$node.\text{weight} = left.\text{weight} + right.\text{weight}.$$

**Definition C.4** (1D Range Tree Query). Given a query range $[low, high]$, start at the root node of the 1D range tree $T$.

1. Start the recursive query from the root node of $T$.
2. For the current $node$, if $node$ falls within the range $[low, high]$, return $p$.weight. If $low$ lies within the left child of $p$, recursively query the left subtree; if $high_1$ lies within the right child, recursively query the right subtree.
3. When backtracking, sum the results of the left child and the right child and return them.

Next, we introduce the $k$D Range Tree ($k \geq 2$).

**Definition C.5** ($k$D Range Tree Construction). For a set of points $P = \{((x_t^i)_{i=1}^k, w_t)\}$ where each point has coordinates $(x_t^i)_{i=1}^k$ and a weight $w_t$, the $k$D range tree is constructed as follow:

1. Group the points by their first dimension, and sort each group by the first dimension, denoted as $p_1, \ldots, p_n$.
2. Construct the $k$D range tree $T_1$ using $p_1, \ldots, p_n$ in a similar approach to the 1D range tree, partitioning the first dimension. Note that each node of $T_1$ contains an associated $(k-1)$D range tree $T_2$ for the second dimension, recursively.
3. For each node in $T_1$, take the points covered by that node, group them by their second dimension, sort them, and construct a corresponding $(k-1)$D range tree $T_2$.

**Definition C.6** ($k$D Range Tree Query). Given a $k$-dimensional query range $\prod_{i=1}^k [low_i, high_i]$ and $k$D range tree $T_1$:

1. Start the recursive query from the root node of $T_1$.
2. For the current $node$, if $node$ falls within the range $[low_1, high_1]$, perform a query $\prod_{i=2}^k [low_i, high_i]$ on $T_2$ (call $(k-1)$D tree query recursively). If $low_1$ lies within the left child of $node$, recursively query the left subtree; if $high_1$ lies within the right child, recursively query the right subtree.
3. When backtracking, sum the results of the left child and the right child and return them.

We present construction and query time complexity of $k$D range tree for any $k \geq 1$.

**Lemma C.7** (Construction Time). *For any $k \geq 1$, the runtime of $k$D range tree construction is $O(n \log^k n)$.*

**Lemma C.8** (Query Time). *For any $k \geq 1$, the runtime of $k$D range tree query is $O(\log^k n)$.*

Based on the properties of the range tree, we can derive the following lemma.

**Lemma C.9.** *Let $T$ denote a $k$D range tree constructed from a set of points $P = \{((x_t^i)_{i=1}^k, w_t)\}$. If a $k$-dimensional query range $\prod_{i=1}^k [low_i, high_i]$ satisfies $low_i \leq \min_{x \in P} x^i$ or $high_i \geq \max_{x \in P} x^i$ for all $i \in [k]$, the query result is the sum of the weights of at most $\lceil \log n \rceil^k$ tree nodes.*

*Proof.* Note that the depth of the tree (i.e., the maximum number of nodes along the path from the root node to any node in the tree) in each dimension of the $k$D range tree is at most $(\lceil \log n \rceil + 1)$ by Definition C.5. For each $i \in [k]$, consider the process of querying the range $[low_i, high_i]$ on the $(k - i + 1)$D range trees.

If $low_i \leq \min_{x \in P} x^i$ and $high_i \geq \max_{x \in P} x^i$, only the root node of each $(k-i+1)$D range tree can be queried. Otherwise, the weight of the root node is not included in the sum. Moreover, when querying the left and right subtrees simultaneously on each $(k-i+1)$D range tree, at least one of the subtrees will return after performing a query at its root node. Therefore, for each 1D range tree, at most $\lceil \log n \rceil$ tree nodes' weights are summed; For each $(k-i+1)$D range tree satisfying $2 \leq i \leq k$, at most $\lceil \log n \rceil$ tree nodes will be queried.

After aggregating the results over $k$ dimensions, the query result is the sum of the weights of at most $\lceil \log n \rceil^k$ tree nodes. $\quad\square$

## C.2. Tight bound on the global sensitivity of subgraph counting

In Section 3, we used $\text{GS}_{f_H}$ to denote the global sensitivity of subgraph counting. We present an exact method for computing the global sensitivity $\text{GS}_{f_H}$, from which we derive a simple and asymptotically tight bound. Table 3 summarizes the global sensitivity of common pattern graphs.

*Table 3.* Global sensitivity $\text{GS}_{f_H}$ of some common pattern graphs $H$

| Pattern Graph | $\text{GS}_{f_H}$ |
|:---:|:---:|
| Edge | $1$ |
| Triangle | $n-2$ |
| $k$-Star | $2\binom{n-2}{k-1}$ |
| $k$-Clique | $\binom{n-2}{k-2}$ |

**Definition C.10** (Automorphism Group). An *automorphism* of a graph $G = (V, E)$ is a permutation $\sigma$ of $V$ such that $(u, v) \in E$ if and only if $(\sigma(u), \sigma(v)) \in E$. The set of all automorphisms of $G$ forms a group under composition, called the *automorphism group* of $G$, denoted by $\text{aut}(G)$.

**Lemma C.11.** *For $n$-vertex graphs, given a pattern graph $H$, the global sensitivity $\text{GS}_{f_H}$ is given by*

$$\text{GS}_{f_H} = \binom{n-2}{h-2} \cdot \frac{2m_H(h-2)!}{|\text{aut}(H)|} = \Theta\left(n^{h-2}\right),$$

*where the hidden constant depends only on $h$.*

*Proof.* Note that $\text{GS}_{f_H}$ is actually the number of occurrences of $H$ that contain a specific vertex pair $(i, j)$ in the complete graph $K_n$. This equals the number of ways to choose and permute the remaining $h - 2$ vertices, multiplied by the number of ways to map an edge of $H$ to $(i, j)$, divided by $|\text{aut}(H)|$ to account for symmetries. Since $h - 1 \leq m_H \leq \binom{h}{2}$ and $1 \leq |\text{aut}(H)| \leq h!$, the expression for $\text{GS}_{f_H}$ yields a tight bound up to constants depending only on $h$. $\quad\square$

Following analogous reasoning, we can obtain an upper bound for $f_H(G)$.

**Lemma C.12.** *For any $n$-vertex graph $G$, given a pattern graph $H$, it holds that*

$$f_H(G) \leq f_H(K_n) = \binom{n}{h} \cdot \frac{h!}{|\text{aut}(H)|} = \Theta\left(n^h\right),$$

*where the hidden constant depends only on $h$.*

## C.3. Discussions on the approximated local sensitivity $\widetilde{\text{HS}}_{f_H}(G)$

There is no explicit upper bound on $\widetilde{\text{HS}}_{f_H}(G)$ for all $H$, and its value typically varies depending on $H$ and $G$. For some $H$, $\widetilde{\text{HS}}_{f_H}(G)$ can be relatively easy to estimate, while for others, it presents more significant challenges. Nevertheless, our results remain practically significant. For convenience, we use $\triangle$ and $*$ to represent triangle and 2-star in the following lemmas, respectively.

**Lemma C.13.** *For any constant $\beta \in (0, 1)$, it holds that $\widetilde{\text{HS}}_{f_\triangle}(G) \leq \text{LS}_{f_\triangle}(G) + \frac{\ln(2/(\beta\delta'))}{\varepsilon'}\left(\frac{\ln(2/(\beta\delta'))}{\varepsilon'} + 1\right) \leq d_{\max}(G) + \frac{\ln(2/(\beta\delta'))}{\varepsilon'}\left(\frac{\ln(2/(\beta\delta'))}{\varepsilon'} + 1\right)$ with probability at least $1 - \beta$.*

*Proof.* Karwa et al. (2011) provided a proof for the case of $k$-triangles. For clarity, we have rewritten the proof for triangles.

If $H$ is a triangle, then $f_\triangle^{(1)}(G) = \mathrm{LS}_{f_\triangle}(G) \leq d_{\max}(G)$, $f_\triangle^{(2)}(G) \leq 1$, $f_\triangle^{(3)}(G) \leq 1$. According to the algorithm Algorithm 4, it holds that

$$\widetilde{\mathrm{HS}}_{f_\triangle}^{(2)}(G) \leq f_\triangle^{(2)}(G) + \frac{\ln{(1/\delta')}}{\varepsilon'} + |\mathrm{Lap}(1/\varepsilon')| \leq 1 + \frac{\ln{(1/\delta')}}{\varepsilon'} + |\mathrm{Lap}(1/\varepsilon')|$$

$$\widetilde{\mathrm{HS}}_{f_\triangle}(G) = \widetilde{\mathrm{HS}}_{f_\triangle}^{(1)}(G) \leq f_\triangle^{(1)}(G) + \widetilde{\mathrm{HS}}_{f_\triangle}^{(2)}(G) \frac{\ln{(1/\delta')}}{\varepsilon'} + \left|\mathrm{Lap}\left(\widetilde{\mathrm{HS}}_{f_\triangle}^{(2)}(G)/\varepsilon'\right)\right|$$

$$\leq \mathrm{LS}_{f_\triangle}(G) + \widetilde{\mathrm{HS}}_{f_\triangle}^{(2)}(G) \frac{\ln{(1/\delta')}}{\varepsilon'} + \left|\mathrm{Lap}\left(\widetilde{\mathrm{HS}}_{f_\triangle}^{(2)}(G)/\varepsilon'\right)\right|.$$

We have $\widetilde{\mathrm{HS}}_{f_\triangle}^{(2)}(G) \leq \frac{\ln{(2/(\beta\delta'))}}{\varepsilon'} + 1$ and $\widetilde{\mathrm{HS}}_{f_\triangle}(G) \leq \mathrm{LS}_{f_\triangle}(G) + \frac{\ln{(2/(\beta\delta'))}}{\varepsilon'}(\frac{\ln{(2/(\beta\delta'))}}{\varepsilon'} + 1) \leq d_{\max}(G) + \frac{\ln{(2/(\beta\delta'))}}{\varepsilon'}(\frac{\ln{(2/(\beta\delta'))}}{\varepsilon'} + 1)$ with probability at least $1 - \beta$ by Fact 2.6 and the union bound. $\square$

**Lemma C.14.** *For any constant $\beta \in (0, 1)$, it holds that $\widetilde{\mathrm{HS}}_{f_*}(G) \leq \mathrm{LS}_{f_*}(G) + \frac{\ln{(1/(\beta\delta'))}}{\varepsilon'} \leq 2d_{\max}(G) + \frac{\ln{(1/(\beta\delta'))}}{\varepsilon'}$ with probability at least $1 - \beta$.*

*Proof.* If $H$ is a 2-star, then $f_*^{(1)}(G) = \mathrm{LS}_{f_*}(G) \leq 2d_{\max}(G)$, $f_*^{(2)}(G) \leq 1$. According to Algorithm 4, $\widetilde{\mathrm{HS}}_{f_*}(G) \leq \mathrm{LS}_{f_*}(G) + \frac{\ln{(1/\delta')}}{\varepsilon'} + |\mathrm{Lap}(1/\varepsilon')| \leq 2d_{\max}(G) + \frac{\ln{(1/\delta')}}{\varepsilon'} + |\mathrm{Lap}(1/\varepsilon')|$. We have $\widetilde{\mathrm{HS}}_{f_\triangle}(G) \leq \mathrm{LS}_{f_*}(G) + \frac{\ln{(1/(\beta\delta'))}}{\varepsilon'} \leq 2d_{\max}(G) + \frac{\ln{(1/(\beta\delta'))}}{\varepsilon'}$ with probability at least $1 - \beta$ by Fact 2.6. $\square$

## D. Deferred proofs from Section 2

### D.1. Proof of Lemma 2.10

For a matrix $\mathbf{A}$ with $n$ columns, and a set $S \subseteq [n]$ we use $\mathbf{A}|_S$ to denote the submatrix of $\mathbf{A}$ consisting of the columns corresponding to elements of $S$. We adopt the following definition and result from Muthukrishnan & Nikolov (2012), with notation slightly modified. Note that the set $C_\alpha$ for $\alpha = 1$ is equivalent to the set $\{\pm 1\}^n$.

**Definition D.1.** For any $\mathbf{A} \in \mathbb{R}^{m \times n}$, we define $\mathrm{herdisc}_{C_\alpha}(\mathbf{A}) = \max_{S \subseteq [n]} \mathrm{disc}_{C_\alpha}(\mathbf{A}|_S)$.

**Lemma D.2** (Lemma 5 in Muthukrishnan & Nikolov (2012)). *Let $f(s) = \max_{S \subseteq [n]:|S| \leq s} \mathrm{disc}_{C_\alpha}(\mathbf{A}|_S)$. Then $\mathrm{disc}_{C_1}(\mathbf{A}) \leq \sum_{i=0}^\infty f((1-\alpha)^i n)$.*

**The case for $d = O(1)$** Let an *anchored axis-parallel box* denote an element of the form $[0, b_0] \times \cdots \times [0, b_d]$. By the main results in Matoušek & Nikolov (2015), we obtain the following lemma.

**Lemma D.3** (Matoušek & Nikolov (2015)). *Let $d = O(1)$. For any $n$ large enough, there exists a set of $n$ points in $[n]^d$ and $n$ anchored axis-parallel boxes $B_1, B_2, \ldots, B_n$ such that the following holds. Let $\mathbf{A}$ denote the incidence matrix of the collection of set $\{B_j \cap P, j \in [n]\}$. Then $\mathrm{herdisc}_{C_1}(\mathbf{A}) = \Omega(\log^{d-1} n)$.*

By an approach similar to that in Muthukrishnan & Nikolov (2012), we derive the following lemma.

**Lemma D.4.** *Let $d = O(1)$. For any $n$ large enough, there exists a set of $n$ points $P$ and $n$ axis-parallel boxes $B_1, B_2, \ldots, B_n$ with a non-empty common intersection such that the following holds. Let $\mathbf{A}$ denote the incidence matrix of the collection of set $\{B_j \cap P, j \in [n]\}$. Then for $\alpha = \Omega(1)$, $\mathrm{disc}_{C_\alpha}(\mathbf{A}) = \Omega(\log^{d-1} n)$.*

*Proof.* By Lemma D.3, for any $n$ large enough, there exists a set of $n$ points $P'$ in $[n]^d$ and $n$ anchored axis-parallel boxes $B_1, B_2, \ldots, B_n$ such that the incidence matrix $\mathbf{B}'$ of $\{B_j \cap P', j \in [n]\}$ has $\mathrm{herdisc}_{C_1}(\mathbf{B}') = \Omega(\log^{d-1} n)$. Note that all boxes $B_j$ contain the common point $(0, 0, \ldots, 0)$.

Let $S^* = \arg\max_{S \in [n]} \mathrm{disc}_{C_1}(\mathbf{B}'|_S)$. Let $P$ be the set of points corresponding to $S^*$, and augment $P$ with additional points not contained in any of the boxes $B_1, B_2, \ldots, B_n$, so that $|P| = n$. Let $\mathbf{B}$ denote the incidence matrix of $\{B_j \cap P, j \in [n]\}$. Then we have $\mathrm{disc}_{C_1}(\mathbf{B}) = \mathrm{disc}_{C_1}(\mathbf{B}'|_{S^*}) = \mathrm{herdisc}_{C_1}(\mathbf{B}') = \Omega(\log^{d-1} n)$.

Now we can prove the lemma. Assume for contradiction that all but finitely many $n \times n$ incidence matrices $\mathbf{A}$ of points and axis-parallel boxes with a non-empty common intersection have discrepancy $\mathrm{disc}_{C_\alpha}(\mathbf{A}) = o(\log^{d-1} n)$. We take any set

of points $P$, axis-parallel boxes, and the corresponding incidence matrix $\mathbf{B}$ satisfying $\mathrm{disc}_{C_1}(\mathbf{B}) = \Omega(\log^{d-1} n)$ from the previous construction. By augmenting additional points, any restriction $\mathbf{B}|_S$ for $S \subseteq P$ is also the incidence matrix of sets induced by points and axis-parallel boxes in the assumption, and thus we have $\mathrm{disc}_{C_\alpha}(\mathbf{B}|_S) = o(\log^{d-1} n)$. Plugging this bound into Lemma D.2, we get $\mathrm{disc}_{C_1}(\mathbf{B}) = o(\log^{d-1} n)$, a contradiction. $\square$

**The case for** $d = O(\log n)/\Omega(\log n)$ A *boolean point* is a point in $\{0,1\}^d$. A *boolean axis-parallel box* is an element of $\{\{0\},\{1\},[0,1]\}^d$. By a probabilistic construction in Chazelle & Lvov (2001), we obtain the following lemma.

**Lemma D.5** (Chazelle & Lvov (2001)). *Let* $d = \Theta(\log n)$. *For any* $m$ *large enough, there exists a set of* $n = \Theta(m)$ *boolean points and* $m$ *boolean axis-parallel boxes* $B_1, B_2, \ldots, B_m$ *such that the following holds. Let* $\mathbf{A}$ *denote the incidence matrix of the collection of set* $\{B_j \cap P, j \in [m]\}$. *Then* $\mathrm{herdisc}_{C_1}(\mathbf{A}) = n^{\Omega(1)}$.

By a simple transformation and an approach similar to the proof of Lemma D.4, we derive the following lemma.

**Lemma D.6.** *For any* $m$ *large enough, there exists a set of* $n = \Theta(m)$ *points* $P$ *and* $m$ *axis-parallel boxes* $B_1, B_2, \ldots, B_m$ *with a non-empty common intersection such that the following holds. Let* $\mathbf{A}$ *denote the incidence matrix of the collection of set* $\{B_j \cap P, j \in [m]\}$. *Then for* $\alpha = \Omega(1)$,

- *if* $d = O(\log n)$, *then* $\mathrm{disc}_{C_\alpha}(\mathbf{A}) = 2^{\Omega(d)}$;
- *if* $d = \Omega(\log n)$, *then* $\mathrm{disc}_{C_\alpha}(\mathbf{A}) = n^{\Omega(1)}$.

*Proof.* Let $d = \Theta(\log n)$. By Lemma D.5, for any $m$ large enough, there exists a set of $n = \Theta(m)$ boolean points $P'$ and $m$ boolean axis-parallel boxes $B_1', B_2', \ldots, B_m'$ such that the incidence matrix $\mathbf{B}'$ of $\{B_j' \cap P', j \in [m]\}$ has $\mathrm{herdisc}_{C_1}(\mathbf{B}') = n^{\Omega(1)}$. Replacing $\{0\}$ and $\{1\}$ in $B_1', B_2', \ldots, B_m'$ with the intervals $[0, 0.5]$ and $[0.5, 1]$, respectively, yields axis-parallel boxes $B_1, B_2, \ldots, B_m$. This transformation preserves the containment of points within boxes, and thus leaves the incidence matrix $\mathbf{B}'$ unchanged. Furthermore, all resulting boxes $B_i$ contain the common point $(0.5, 0.5, \ldots, 0.5)$.

Let $S^* = \arg\max_{S \in [n]} \mathrm{disc}_{C_1}(\mathbf{B}'|_S)$. Let $P$ be the set of points corresponding to $S^*$, and augment $P$ with additional points not contained in any of the boxes $B_1, B_2, \ldots, B_m$, so that $|P| = \Theta(m)$. Let $\mathbf{B}$ denote the incidence matrix of $\{B_j \cap P, j \in [m]\}$. Then we have $\mathrm{disc}_{C_1}(\mathbf{B}) = \mathrm{disc}_{C_1}(\mathbf{B}'|_{S^*}) = \mathrm{herdisc}_{C_1}(\mathbf{B}') = n^{\Omega(1)}$. Then by an approach similar to the proof of Lemma D.4, we can prove that there are infinitely many $m \times n$ incidence matrices $\mathbf{A}$ of points and axis-parallel boxes with a non-empty common intersection have discrepancy $\mathrm{disc}_{C_\alpha}(\mathbf{A}) = n^{\Omega(1)}$.

The case for $d = \Omega(\log n)$ follows immediately from that for $d = \Theta(\log n)$. We thus focus on the case for $d = O(\log n)$. Set $n' = \Theta(2^d)$ so that the above conclusion can be applied to the cases for $d = \Theta(\log n')$. Let $m'$ be the number of boxes corresponding to $n'$. Then we can add $n - n'$ points that are not contained in any boxes and $m - m'$ boxes that do not contain any points. Thus, we finish the proof of the case for $d = O(\log n)$. $\square$

Combining Lemmas D.4 and D.6, we complete the proof of Lemma 2.10.

# E. Deferred Proofs and Analysis from Section 3

## E.1. Proof of Theorem 3.3: The analysis of pure DP algorithm

**Privacy** We now prove that Algorithm 3 is an $\varepsilon$-DP algorithm.

**Lemma E.1.** *Let* $\mathbf{w}$ *be the weight vector generated by Algorithm 1. For any discretized query* $q = \prod_{i=1}^d [\ell_i, r_i]$, *let* $G_q$ *be the corresponding induced subgraph. The number of occurrences of* $H$ *in* $G_q$ *is equal to the sum of the weights of all rank tuples falling within the range* $\prod_{i=1}^d ([\ell_i, n] \times [1, r_i])$, *i.e.,* $f_H(G_q) = \sum_{\mathbf{v} \in \prod_{i=1}^d ([\ell_i, n] \times [1, r_i])} w_{\mathbf{v}}$.

*Proof.* Let $H' = (V(H'), E(H'))$ be an occurrence of $H$ in $G$ with rank tuple $(s_i(u_i), s_i(v_i))_{i=1}^d$. Then $H'$ is in $G_q$ if and only if the ranks of vertices in $H'$ satisfy $\min_{v \in V(H')} s_i(v) \geq \ell_i$ and $\max_{v \in V(H')} s_i(v) \leq r_i$ for all $i \in [d]$. The latter is equal to the condition that $s_i(u_i) \geq \ell_i$ and $s_i(v_i) \leq r_i$ for all $i \in [d]$ by Definition 3.1. By Algorithm 1, each such occurrence $H'$ exactly increases the weight vector $\mathbf{w}$ by 1 at the position corresponding to its rank tuple. Therefore, the number of occurrences of $H$ in $G_q$ is equal to the sum of the weights of all rank tuples falling within the range $\prod_{i=1}^d ([\ell_i, n] \times [1, r_i])$. $\square$

**Lemma E.2.** *Algorithm 3 is* $\varepsilon$-*DP.*

*Proof.* We use $\mathbf{w}$ and $\mathbf{w}'$ to denote the different weight vectors formed by two neighboring graphs $G$ and $G'$, respectively, where $G$ and $G'$ differ by a single edge. By fixing $H$ and treating $\mathbf{w}$ as a function of $G$, we can define the global sensitivity of $\mathbf{w}$, denoted as $\mathrm{GS}_{\mathbf{w}}$, by Definition 2.2. Note that for any $\mathbf{w}, \mathbf{w}'$, we have $\|\mathbf{w} - \mathbf{w}'\|_1 = |\,\|\mathbf{w}\|_1 - \|\mathbf{w}'\|_1\,|$. This follows from the fact that the function $\mathbf{w}$ are monotonic, meaning that the addition of any edge does not reduce the number of occurrences of $H$ at each component of $\mathbf{w}$. Thus, we have

$$\mathrm{GS}_{\mathbf{w}} = \max_{\mathbf{w}, \mathbf{w}'} \|\mathbf{w} - \mathbf{w}'\|_1 = \max_{\mathbf{w}, \mathbf{w}'} |\,\|\mathbf{w}\|_1 - \|\mathbf{w}'\|_1\,|$$

$$= \max \left| \sum_{\mathbf{v} \in [n]^{2d}} w_{\mathbf{v}} - \sum_{\mathbf{v} \in [n]^{2d}} w'_{\mathbf{v}} \right| \tag{1}$$

$$= \max_{G \sim G'} |f_H(G) - f_H(G')| = \mathrm{GS}_{f_H}, \tag{2}$$

where the penultimate equation follows from Lemma E.1 by setting the discretized query $q = [1, n]^{2d}$.

Let us revisit the individual dimensions of the $2d$D range tree in the algorithm. For any $(2d - i + 1)$D range tree $T_i$ in the $2d$D range tree, where $1 \le i \le 2d$, define the depth of each node in $T_i$ as the number of nodes along the path from the root node of $T_i$ to that node. Note that the depth of a node is at most $(\lceil \log n \rceil + 1)$ by Definition C.5.

We now partition the nodes of the $2d$D range tree $T_1$ according to their depths. This partition thus consists of at most $(\lceil \log n \rceil + 1)$ sets of nodes. For each node in each set, we recursively partition the nodes in its associated $(2d - 1)$D range tree. That is, in each recursive step, a node in the $2d$D range tree is replaced by the nodes of its associated $(2d - 1)$D range tree. We continue this process until each set consists solely of nodes from 1D range trees. After aggregating the results over $2d$ dimensions, we obtain a partition consisting of $\theta$ sets of nodes with $\theta \le (\lceil \log n \rceil + 1)^{2d}$.

For each node set in the partition, we concatenate the weights of the nodes in the set into a weight vector $\mathbf{w}_i$, resulting in a collection of weight vectors $\{\mathbf{w}_i\}_{i=1}^{\theta}$. Note that for each $\mathbf{w}_i$, there exists a set $S_i \subseteq [n]^{2d}$ associated with the node set, which is independent of the edges in $G$ and satisfies $\|\mathbf{w}_i\|_1 = \sum_{\mathbf{v} \in S_i} w_{\mathbf{v}}$. We can similarly define the global sensitivity of $\mathbf{w}_i$ as $\mathrm{GS}_{\mathbf{w}_i}$. Let $\mathbf{w}_i$ and $\mathbf{w}'_i$ be the weight vectors for neighboring graphs $G$ and $G'$, respectively. Then we have $\|\mathbf{w}_i - \mathbf{w}'_i\|_1 = |\,\|\mathbf{w}_i\|_1 - \|\mathbf{w}'_i\|_1\,|$ and

$$\mathrm{GS}_{\mathbf{w}_i} = \max_{\mathbf{w}_i, \mathbf{w}'_i} \|\mathbf{w}_i - \mathbf{w}'_i\|_1 = \max_{\mathbf{w}_i, \mathbf{w}'_i} |\,\|\mathbf{w}_i\|_1 - \|\mathbf{w}'_i\|_1\,|$$

$$= \max \left| \sum_{\mathbf{v} \in S_i} w_{\mathbf{v}} - \sum_{\mathbf{v} \in S_i} w'_{\mathbf{v}} \right| = \max \left| \sum_{\mathbf{v} \in S_i} (w_{\mathbf{v}} - w'_{\mathbf{v}}) \right|$$

$$\le \max \left| \sum_{\mathbf{v} \in [n]^{2d}} (w_{\mathbf{v}} - w'_{\mathbf{v}}) \right| = \mathrm{GS}_{\mathbf{w}}, \tag{3}$$

where the final inequality follows from the monotonicity of $\mathbf{w}$ and the final equation follows from Equation (1).

Let $\mathbf{w}_T = (\mathbf{w}_i)_{i=1}^{\theta}$ denote the weight vector obtained by concatenating all $\mathbf{w}_i$. We can similarly define the global sensitivity of $\mathbf{w}_T$ as $\mathrm{GS}_{\mathbf{w}_T}$. Then, we have $\mathrm{GS}_{\mathbf{w}_T} \le \sum_{i=1}^{\theta} \mathrm{GS}_{\mathbf{w}_i} \le (\lceil \log n \rceil + 1)^{2d} \mathrm{GS}_{\mathbf{w}} = (\lceil \log n \rceil + 1)^{2d} \mathrm{GS}_{f_H}$, where the first two inequalities and the final equation follow from the monotonicity of $\mathbf{w}_i$, Equation (3), and Equation (2), respectively.

Therefore, according to the Laplace mechanism (Definition 2.7), adding Laplace noise with scale $\frac{\mathrm{GS}_{f_H} \cdot (\lceil \log n \rceil + 1)^{2d}}{\varepsilon}$ to the weights of nodes corresponding to each component of $\mathbf{w}_T$ ensures that $\widetilde{T}_1$ achieves differential privacy. For each query, the range tree $\widetilde{T}_1$ are reused. Therefore, Algorithm 3 maintains $\varepsilon$-DP based on the post-processing property (Proposition C.1). $\qquad \square$

**Utility** we have the following lemma.

**Lemma E.3.** *For any $\varepsilon > 0$, the output of Algorithm 3 satisfies*

$$\max_{q \in Q} \left| f_H(G_q) - \widetilde{f}_H(G_q) \right| = O\left( \frac{\mathrm{GS}_{f_H} \cdot \sqrt{\log(n|Q|)} \cdot \log^{3d} n}{\varepsilon} \right)$$

*with probability at least $1 - \frac{1}{n}$. Here, the hidden constants are of the form $c^{O(d)}$ for some universal constant $c > 1$.*

*Proof.* Let $P_q = \{p_i\}_{i=1}^m$ with $|P_q| = m$ denote the set of nodes in the $2d$D range tree selected by a query $q \in Q$. Then we have $m \le \lceil \log n \rceil^{2d}$ by Lemma C.9. For any $p_i \in P_q$, Let $Y_{p_i}$ denote an independent random variable with $Y_{p_i} \sim \text{Lap}(\text{GS}_{f_H} \cdot (\lceil \log n \rceil + 1)^{2d}/\varepsilon)$. Let $w(p_i)$ denote its weight and $\widetilde{w}(p_i)$ denote the weight $w(p_i)$ plus Laplace noise. Then the additive error $|f_H(G_q) - \widetilde{f}_H(G_q)|$ satisfies

$$\left| f_H(G_q) - \widetilde{f}_H(G_q) \right| = \left| \sum_{i=1}^m w(p_i) - \sum_{i=1}^m \widetilde{w}(p_i) \right| \le \left| \sum_{i=1}^m Y_{p_i} \right| = O\left( \frac{\text{GS}_{f_H} \cdot \sqrt{\log(n|Q|)} \cdot \log^{3d} n}{\varepsilon} \right)$$

with a probability of at least $1 - \frac{1}{n|Q|}$ by Lemma C.2 which $b = \text{GS}_{f_H} \cdot (\lceil \log n \rceil + 1)^{2d}/\varepsilon$, $m = \lceil \log n \rceil^{2d}$ and $\beta = 2n|Q|$. By the union bound, we finish the proof. $\square$

Combining Lemma E.2 and Lemma E.3, we finish the proof of Theorem 3.3. The complexity analysis of the algorithm in Theorem 3.3 is presented in Appendix E.3.

### E.2. Proof of Theorem 1.3: The analysis of approximate DP algorithm

Let $\widetilde{\text{HS}}_{f_H}^{(k)}(G)$ be the quantity computed at Step 4 of Algorithm 4. Then, the noisy estimate of $\text{LS}_{f_H}(G)$ is $\widetilde{\text{HS}}_{f_H}(G) = \widetilde{\text{HS}}_{f_H}^{(1)}(G)$. This follows from the fact that $f_H^{(1)}(G) = \text{LS}_{f_H}(G)$, and the subsequent bias term at Step 4 of Algorithm 4 ensures privacy. We have the following lemma from Nguyen et al. (2023).

**Lemma E.4** (Nguyen et al. (2023)). *It holds that (1) $\widetilde{\text{HS}}_{f_H}(G)$ is a $(m_H\varepsilon', \delta' + m_H e^{\varepsilon'}\delta')$-DP estimate of $\text{LS}_{f_H}(G)$; (2)* $\Pr[\widetilde{\text{HS}}_{f_H}^{(k)}(G) \ge f_H^{(k)}(G)] \ge 1 - \delta'$, *for any $k \in [m_H - 1]$.*

We also need the following lemma, the proof of which is based on the proof of Lemma 4.4 in Karwa et al. (2011). We extend the proof there to the case of a family of multidimensional functions so that it can be adapted to the problem we are addressing.

**Lemma E.5.** *Let $\mathcal{F} = \{f_i\}_{i=1}^k$ be a family of $k$ functions, where $f_i : \mathcal{X} \to \mathbb{R}^{d_i}$ and $\text{LS}_{\mathcal{F}}(x) = \max_{f \in \mathcal{F}} \text{LS}_f(x)$. Let $\mathcal{B}$ be an $(\varepsilon_1, \delta_1)$-DP algorithm such that $\Pr[\mathcal{B}(x) \ge \text{LS}_{\mathcal{F}}(x)] > 1 - \delta_2$ for all $x$. Consider the algorithm $\mathcal{A}$ that runs $\mathcal{B}(x)$ to obtain an estimate $\widetilde{\text{LS}}_x$ of $\text{LS}_{\mathcal{F}}(x)$, and releases both $\widetilde{\text{LS}}_x$ and a noisy estimate of $\mathcal{F}$, i.e.,*

$$\mathcal{A}(x) = \left( \widetilde{\text{LS}}_x, \left( f_i(x) + \text{Lap}^{d_i}\left( \widetilde{\text{LS}}_x \cdot 2\sqrt{2k\ln(1/\delta')}/\varepsilon_2 \right) \right)_{i=1}^k \right),$$

*where $\widetilde{\text{LS}}_x = \mathcal{B}(x)$, $\delta' = \min(e^{-\varepsilon_2/8}, \delta_2)$, and $\text{Lap}^{d_i}(b)$ is a $d_i$-dimensional vector such that each element is independently sampled from a Laplace distribution with mean $0$ and scale parameter $b$. Then $\mathcal{A}$ is $(\varepsilon_1 + \varepsilon_2, \max(\delta_1 + 2e^{\varepsilon_1}\delta_2, 2\delta_2 + e^{\varepsilon_2}\delta_1))$-DP.*

*Proof.* Given neighboring datasets $x$ and $x'$, consider the following algorithms:

$$\mathcal{A}(x) = \left( \widetilde{\text{LS}}_x, \left( f_i(x) + \text{Lap}^{d_i}\left( \widetilde{\text{LS}}_x \cdot 2\sqrt{2k\ln(1/\delta')}/\varepsilon_2 \right) \right)_{i=1}^k \right),$$

$$\mathcal{A}(x') = \left( \widetilde{\text{LS}}_{x'}, \left( f_i(x') + \text{Lap}^{d_i}\left( \widetilde{\text{LS}}_{x'} \cdot 2\sqrt{2k\ln(1/\delta')}/\varepsilon_2 \right) \right)_{i=1}^k \right),$$

where $\widetilde{\text{LS}}_x = \mathcal{B}(x), \widetilde{\text{LS}}_{x'} = \mathcal{B}(x')$ and $\delta' = \min(e^{-\varepsilon_2/8}, \delta_2)$. Without loss of generality, assume that $\text{LS}_{\mathcal{F}}(x') \le \text{LS}_{\mathcal{F}}(x)$. Now, define the random variable

$$\mathcal{A}_{\text{mix}} = \left( \widetilde{\text{LS}}_x, \left( f_i(x') + \text{Lap}^{d_i}\left( \widetilde{\text{LS}}_x \cdot 2\sqrt{2k\ln(1/\delta')}/\varepsilon_2 \right) \right)_{i=1}^k \right).$$

Let $p_x, p_{x'}$ and $p_{\text{mix}}$ be the probability distributions of $\mathcal{A}(x), \mathcal{A}(x')$ and $\mathcal{A}_{\text{mix}}$, respectively. First, consider the difference between $\mathcal{A}(x')$ and $\mathcal{A}_{\text{mix}}$. They differ only in the initial estimate $\widetilde{\text{LS}}$ (either $\mathcal{B}(x')$ or $\mathcal{B}(x)$). Since $\mathcal{B}$ is $(\varepsilon_1, \delta_1)$-DP and post-processing does not affect differential privacy, it follows that for every event $E$,

$$p_{x'}(E) \le e^{\varepsilon_1} p_{\text{mix}}(E) + \delta_1 \text{ and } p_{\text{mix}}(E) \le e^{\varepsilon_1} p_{x'}(E) + \delta_1. \tag{4}$$

Then consider the difference between $\mathcal{A}(x)$ and $\mathcal{A}_{\text{mix}}$. Let $F$ denote the events $\widetilde{\text{LS}}_x \geq \text{LS}_{\mathcal{F}}(x)$. By the precondition of the lemma, $\Pr[\mathcal{B}(x) \geq \text{LS}_{\mathcal{F}}(x)] > 1 - \delta_2$, $\Pr[F] > 1 - \delta_2$. Let $\widetilde{\mathcal{X}} = \{\widetilde{x} \mid \widetilde{x} \in \mathcal{X}, \text{LS}_{\mathcal{F}}(\widetilde{x}) \leq \text{LS}_{\mathcal{F}}(x)\}$. Note that $x, x' \in \widetilde{\mathcal{X}}$. For any $\widetilde{x} \in \widetilde{\mathcal{X}}$ and $i \in [k]$, conditioned on $F$, the algorithm

$$\mathcal{A}_i(\widetilde{x}) = f_i(\widetilde{x}) + \text{Lap}^{d_i}\left(\widetilde{\text{LS}}_x \cdot 2\sqrt{2k\ln(1/\delta')}/\varepsilon_2\right)$$

over $\widetilde{\mathcal{X}}$ is $(\varepsilon_2/(2\sqrt{2k\ln(1/\delta')}))$-DP. By applying the advanced composition theorem (Proposition 2.4), the algorithm $(A_i(\widetilde{x}))_{i=1}^k$ is $(\widetilde{\varepsilon}, \delta')$-DP, where $\delta' \leq \delta_2$ and

$$\widetilde{\varepsilon} = k \cdot \frac{\varepsilon_2}{2} \cdot \frac{\varepsilon_2}{8k\ln(1/\delta')} + \frac{\varepsilon_2}{2} \leq \varepsilon_2.$$

It implies

$$p_{\text{mix}}(E|F) \leq e^{\varepsilon_2} p_x(E|F) + \delta_2 \text{ and } p_x(E|F) \leq e^{\varepsilon_2} p_{\text{mix}}(E|F) + \delta_2.$$

Since the probability of $F$ is the same under both $p_{\text{mix}}$ and $p_x$, we can strengthen this to

$$p_{\text{mix}}(E \cap F) \leq e^{\varepsilon_2} p_x(E \cap F) + \delta_2 \text{ and } p_x(E \cap F) \leq e^{\varepsilon_2} p_{\text{mix}}(E \cap F) + \delta_2.$$

Note that $\Pr(\overline{F}) \leq \delta_2$ and thus

$$\begin{aligned}
p_{\text{mix}}(E) &\leq p_{\text{mix}}(E \cap F) + p_{\text{mix}}(E \cap \overline{F}) \\
&\leq e^{\varepsilon_2} p_x(E \cap F) + \delta_2 + p_{\text{mix}}(E \cap \overline{F}) \\
&\leq e^{\varepsilon_2} p_x(E) + 2\delta_2.
\end{aligned} \tag{5}$$

By symmetry, we can obtain

$$p_x(E) \leq e^{\varepsilon_2} p_{\text{mix}}(E) + 2\delta_2. \tag{6}$$

Plugging the inequalities (4) into (5) and (6), we get

$$p_{x'}(E) \leq e^{\varepsilon_1 + \varepsilon_2} p_x(E) + 2e^{\varepsilon_1}\delta_2 + \delta_1 \text{ and } p_x(E) \leq e^{\varepsilon_1 + \varepsilon_2} p_{x'}(E) + e^{\varepsilon_2}\delta_1 + 2\delta_2.$$

Hence we prove $\mathcal{A}$ is $(\varepsilon_1 + \varepsilon_2, \max(\delta_1 + 2e^{\varepsilon_1}\delta_2, 2\delta_2 + e^{\varepsilon_2}\delta_1))$-DP. $\qquad\square$

**Privacy** We have the following lemma.

**Lemma E.6.** *Algorithm 5 is $(\varepsilon, \delta)$-DP.*

*Proof.* We continue to use $\mathbf{w}$ as the vector output in Algorithm 1. We use $\mathbf{w}$ and $\mathbf{w}'$ to denote the different weight vectors formed by neighboring graphs $G$ and $G'$, respectively. By fixing $H$ and treating $\mathbf{w}$ as a function of $G$, we can define the local sensitivity of $\mathbf{w}$, denoted as $\text{LS}_{\mathbf{w}}(G)$, by Definition 2.1. Then we have

$$\text{LS}_{\mathbf{w}}(G) = \max_{\mathbf{w}'} \|\mathbf{w} - \mathbf{w}'\|_1 = \max_{G':G'\sim G} |f_H(G) - f_H(G')| = \text{LS}_{f_H}(G)$$

by a proof similar to Lemma E.2. Thus, if we get a noisy estimate of $\text{LS}_{f_H}(G)$, we get a noisy estimate of $\text{LS}_{\mathbf{w}}(G)$.

Similar to the proof of Lemma E.2, we can obtain a collection of weight vectors $\{\mathbf{w}_i\}_{i=1}^{\theta}$ with $\theta \leq (\lceil \log n \rceil + 1)^{2d}$. Let $d_i$ denote the dimension of $\mathbf{w}_i$ for all $i \in [\theta]$. By Lemma E.4, we can get $\widetilde{\text{HS}}_{f_H}(G)$ for $(m_H\varepsilon', \delta' + m_H e^{\varepsilon'}\delta')$-DP at Step 3 of Algorithm 5. Then, according to Lemma E.5, if we release

$$\mathcal{A}(G) = \left(\widetilde{\text{HS}}_{f_H}(G), \left(\mathbf{w}_i + \text{Lap}^{d_i}\left(\frac{\widetilde{\text{HS}}_{f_H}(G) \cdot (\lceil \log n \rceil + 1)^d \cdot 2\sqrt{2\ln(1/\delta'')}}{\varepsilon'}\right)\right)_{i=1}^{\theta}\right),$$

where $\delta'' = \min(e^{-\varepsilon'/8}, \delta')$, we can obtain a $((m_H+1)\varepsilon', \max(2e^{m_H\varepsilon'} + m_H e^{\varepsilon'} + 1, m_H e^{2\varepsilon'} + e^{\varepsilon'} + 2)\delta')$-DP estimate of $\mathbf{w}$. Here, we set $\varepsilon = (m_H+1)\varepsilon'$ and $\delta = \max(2e^{m_H\varepsilon'} + m_H e^{\varepsilon'} + 1, m_H e^{2\varepsilon'} + e^{\varepsilon'} + 2)\delta'$. Note that Algorithm 5 is actually a post-processing step of $\mathcal{A}(G)$. By the post-processing property (Proposition C.1), Algorithm 5 satisfies $(\varepsilon, \delta)$-DP. $\qquad\square$

**Utility** we have the following lemma.

**Lemma E.7.** *For any $\varepsilon > 0$ and $\delta \in (0, 1)$, the output of Algorithm 5 satisfies*

$$\max_{q \in Q} \left| f_H(G_q) - \widetilde{f}_H(G_q) \right| = O\left( \frac{\widetilde{\mathrm{HS}}_{f_H}(G) \cdot \sqrt{(\varepsilon + \log(1/\delta)) \log(n|Q|)} \cdot \log^{2d} n}{\varepsilon} \right)$$

*with probability at least $1 - \frac{1}{n}$. Here, the hidden constants are of the form $c^{O(d)}$ for some universal constant $c > 1$.*

*Proof.* Let $P_q = \{p_i\}_{i=1}^m$ with $|P_q| = m$ denote the set of nodes in the $2d$D range tree selected by a query $q \in Q$. Then we have $m \leq \lceil \log n \rceil^{2d}$ by Lemma C.9. For any $p_i \in P_q$, let $Y_{p_i}$ denote an independent random variable with $Y_{p_i} \sim \mathrm{Lap}(\widetilde{\mathrm{HS}}_{f_H}(G) \cdot (\lceil \log n \rceil + 1)^d \cdot 2\sqrt{2\ln(1/\delta'')}/\varepsilon')$, where $\varepsilon'$ and $\delta''$ are defined in Algorithm 5. Let $w(p_i)$ denote its weight and $\widetilde{w}(p_i)$ denote the weight $w(p_i)$ plus Laplace noise. Then the additive error $|f_H(G_q) - \widetilde{f}_H(G_q)|$ satisfies

$$\begin{aligned}
\left| f_H(G_q) - \widetilde{f}_H(G_q) \right| &= \left| \sum_{i=1}^m w(p_i) - \sum_{i=1}^m \widetilde{w}(p_i) \right| \leq \left| \sum_{i=1}^m Y_{p_i} \right| \\
&= O\left( \frac{\widetilde{\mathrm{HS}}_{f_H}(G) \cdot \sqrt{\log(1/\delta'') \log(n|Q|)} \cdot \log^{2d} n}{\varepsilon'} \right) \\
&= O\left( \frac{\widetilde{\mathrm{HS}}_{f_H}(G) \cdot \sqrt{(\varepsilon + \log(1/\delta)) \log(n|Q|)} \cdot \log^{2d} n}{\varepsilon} \right)
\end{aligned}$$

with probability at least $1 - \frac{1}{n|Q|}$. The penultimate equation follows from the setting $b = \widetilde{\mathrm{HS}}_{f_H}(G) \cdot (\lceil \log n \rceil + 1)^d \cdot 2\sqrt{2\ln(1/\delta'')}/\varepsilon'$, $m = \lceil \log n \rceil^{2d}$, and $\beta = 2n|Q|$, as supported by Lemma C.2. The final equation follows from $\varepsilon = (m_H + 1)\varepsilon'$, $\delta = \max(2e^{m_H \varepsilon'} + m_H e^{\varepsilon'} + 1, m_H e^{2\varepsilon'} + e^{\varepsilon'} + 2)\delta'$ and $\delta'' = \min(e^{-\varepsilon'/8}, \delta')$ defined in Algorithm 5. By the union bound, we finish the proof. $\qquad\square$

Combining Lemma E.6 and Lemma E.7, we finish the proof of Theorem 1.3. The complexity analysis of the algorithm in Theorem 1.3 is presented in Appendix E.3.

## E.3. The complexity analysis of the algorithms in Theorem 1.3 and Theorem 3.3

The algorithms in Theorem 1.3 and Theorem 3.3 correspond to Algorithm 3 and Algorithm 5, respectively. Before formally presenting the complexity analysis of these algorithms, we first provide the implementation details about the weight vector **w** and the range tree in these algorithms, which further reduce the time and space consumption both theoretically and in practice.

**Implementation details** In practical networks, $f_H(G)$ may be significantly lower than the size of the weight vector **w** in Algorithm 1 (i.e., $n^{2d}$). This implies that most components in the weight vector **w** are 0. Thus, we maintain the non-zero components of **w** by a hash map to avoid materializing all of $[n]^{2d}$ and employ a *dynamic update strategy* to implement the range tree more efficiently.

This strategy is based on the observation that the structure of the range tree in Algorithm 2 depends solely on $n$ and $d$, rather than on the specific structure of $G$. Moreover, the Laplace noise added to the weight of each tree node is determined before the construction of the range tree in Algorithm 2. Therefore, we can assume that the entire range tree has already been constructed initially, with the weight of each node initialized to 0 and the corresponding Laplace noise added, although this is not actually the case. In other words, each node in the range tree is initially *virtual*.

Then, each non-zero component in **w**, whose number is bounded by $f_H(G)$, is treated as a modification, incrementing the weight of the relevant tree nodes by 1. During this process, if a node is virtual and is about to be modified, it will be *instantiated* just before the modification. The process of instantiation includes generating the corresponding Laplace noise and setting the tree node's weight to the noise value.

Similarly, for any query, the weights of the relevant tree nodes need to be summed. If a queried tree node is virtual, it will be instantiated just before the query.

Now we present the complexity analysis of the above algorithms. Let $F_H(G)$ denote the time required to list all occurrences of $H$ in $G$. It is always polynomial since, for any $H$, all occurrences can be simply enumerated in $O(n^h)$ time. In fact, for some specific $H$ such as a triangle, $F_H(G)$ can be improved in sparse graphs (Chiba & Nishizeki, 1985).

**Theorem E.8.** *The algorithm in Theorem 3.3 (i.e., Algorithm 3) requires* $O(\min(n^{2d}, (f_H(G) + |Q|)\log^{2d} n))$ *space and* $O(\min(f_H(G), (dn)^{2d})\log^{2d} n + F_H(G))$ *preprocessing time, and each query can be answered in* $O(\log^{2d} n)$ *time. Here, the hidden constants are of the form* $c^{O(d)}$ *for some universal constant* $c > 1$.

*Proof.* We should decide whether to adopt a dynamic update strategy depending on the specific context.

If so, the space used by Algorithm 3 actually depends on the number of tree nodes accessed during each modification or query, and this number also bounds the query time (see Lemma C.8). Therefore, Algorithm 3 requires $O((f_H(G) + |Q|)\log^{2d} n)$ space and $O(f_H(G)\log^{2d} n + F_H(G))$ preprocessing time by Lemma C.8 since the number of non-zero components of the weight vector $\mathbf{w}$ in Algorithm 1 can be bounded by $f_H(G)$. Note that querying the same induced subgraph multiple times does not incur additional space; therefore, we retain the previous assumption on $Q$ in Section 1.1.

Otherwise, since the number of nodes in a 1D range tree is at most $(2n - 1)$, the maximum number of nodes in the $2d$D range tree in Algorithm 1 is $(2n - 1)^{2d}$, obtained by aggregating the results over the $2d$ dimensions. Therefore, Algorithm 3 requires $O(n^{2d})$ space and $O((dn)^{2d}\log^{2d} n + F_H(G))$ preprocessing time by Lemma C.7.

Both cases require $O(\log^{2d} n)$ query time Lemma C.8. By combining the two cases, the proof is complete. $\square$

**Theorem E.9.** *The algorithm in Theorem 1.3 (i.e., Algorithm 5) requires* $O(\min(n^{2d}, (f_H(G) + |Q|)\log^{2d} n))$ *space and* $O(n^{2m_H} + \min(f_H(G), (dn)^{2d})\log^{2d} n + F_H(G))$ *preprocessing time, and each query can be answered in* $O(\log^{2d} n)$ *time. Here, the hidden constants are of the form* $c^{O(d)}$ *for some universal constant* $c > 1$.

*Proof.* The proof is similar to that of Theorem E.8. The only difference is that Algorithm 5 requires preprocessing $\widetilde{\mathrm{HS}}_{f_H}(G)$ (see Algorithm 4). By Lemma 6 in Nguyen et al. (2023), Algorithm 4 requires $O(n^{2m_H})$ time to preprocess $\widetilde{\mathrm{HS}}_{f_H}(G)$. $\square$

It is worth noting that the dynamic update strategy ensures that the above algorithms run in polynomial time for $d = O(\log n / \log\log n)$. Although the time and space complexity scale exponentially with the dimension and the size of the pattern graph, this limitation reflects the inherent difficulty of range counting and subgraph counting, rather than being specific to our algorithms.

## F. Deferred Proofs and Analysis from Section 4

### F.1. Proof of Lemma 4.3

Note for each query $q \in Q$, the quantity $(\mathbf{A}_H\mathbf{x}_H)_q$ denotes the number of occurrences of $H$ in the induced subgraph $(G(\mathbf{x}))[V_q]$, i.e.,

$$(\mathbf{A}_H\mathbf{x}_H)_q = f_H((G(\mathbf{x}))[V_q]). \tag{7}$$

For each $\chi = \mathbf{x}_H - \mathbf{x}'_H \in C_{\alpha,H}$, let $q_\chi = \arg\max_{q \in Q} |(\mathbf{A}(\mathbf{x} - \mathbf{x}'))_q|$. Then we have

$$\mathrm{disc}_{C_{\alpha,H}}(\mathbf{A}_H) = \min_{\chi \in C_{\alpha,H}} \|\mathbf{A}_H\chi\|_\infty$$

$$= \min_{\chi \in C_{\alpha,H}} \max_{q \in Q} |(\mathbf{A}_H\chi)_q| \geq \min_{\chi \in C_{\alpha,H}} |(\mathbf{A}_H\chi)_{q_\chi}|$$

$$= \min_{\substack{\chi \in C_{\alpha,H} \\ \chi = \mathbf{x}_H - \mathbf{x}'_H}} |(\mathbf{A}_H\mathbf{x}_H)_{q_\chi} - (\mathbf{A}_H\mathbf{x}'_H)_{q_\chi}|$$

$$= \min_{\substack{\chi \in C_{\alpha,H} \\ \chi = \mathbf{x}_H - \mathbf{x}'_H}} \left| f_H\left((G(\mathbf{x}))[V_{q_\chi}]\right) - f_H\left((G(\mathbf{x}'))[V_{q_\chi}]\right) \right|,$$

where the last equation follows from Equation (7) and the others follow from the definitions. For any $\chi \in C_{\alpha,H}$ and $\mathbf{x} \in \{0,1\}^n$, we use $G_{\chi,\mathbf{x}}$ to denote $(G(\mathbf{x}))[V_{q_\chi}]$ and let $\widetilde{E}_{\chi,\mathbf{x}}$ denote the set of private edges in $G_{\chi,\mathbf{x}}$. For any $G_{\chi,\mathbf{x}}$, we

partition all occurrences of $H$ in $G_{\chi,\mathbf{x}}$ by the number of private edges that they contain. Specifically, let $g_H^{(i)}(G_{\chi,\mathbf{x}})$ denote the number of occurrences of $H$ in $G_{\chi,\mathbf{x}}$ containing exactly $i$ private edges. Note that any occurrence of $H$ in $G_{\chi,\mathbf{x}}$ can contain at most $\frac{h}{2}$ private edges. Then we have $f_H(G_{\chi,\mathbf{x}}) = \sum_{i=0}^{\lfloor \frac{h}{2} \rfloor} g_H^{(i)}(G_{\chi,\mathbf{x}})$ for any $G_{\chi,\mathbf{x}}$. Therefore, it follows that

$$\mathrm{disc}_{C_{\alpha,H}}(\mathbf{A}_H) \geq \min_{\substack{\chi \in C_{\alpha,H} \\ \chi = \mathbf{x}_H - \mathbf{x}'_H}} |f_H(G_{\chi,\mathbf{x}}) - f_H(G_{\chi,\mathbf{x}'})| = \min_{\substack{\chi \in C_{\alpha,H} \\ \chi = \mathbf{x}_H - \mathbf{x}'_H}} \left| \sum_{i=0}^{\lfloor \frac{h}{2} \rfloor} g_H^{(i)}(G_{\chi,\mathbf{x}}) - \sum_{i=0}^{\lfloor \frac{h}{2} \rfloor} g_H^{(i)}(G_{\chi,\mathbf{x}'}) \right|. \tag{8}$$

We further partition $\left| \sum_{i=0}^{\lfloor \frac{h}{2} \rfloor} g_H^{(i)}(G_{\chi,\mathbf{x}}) - \sum_{i=0}^{\lfloor \frac{h}{2} \rfloor} g_H^{(i)}(G_{\chi,\mathbf{x}'}) \right|$ in Equation (8) into three parts, .i.e.,

$$\left| \sum_{i=0}^{\lfloor \frac{h}{2} \rfloor} g_H^{(i)}(G_{\chi,\mathbf{x}}) - \sum_{i=0}^{\lfloor \frac{h}{2} \rfloor} g_H^{(i)}(G_{\chi,\mathbf{x}'}) \right|$$

$$= \left| g_H^{(1)}(G_{\chi,\mathbf{x}}) - g_H^{(1)}(G_{\chi,\mathbf{x}'}) + g_H^{(0)}(G_{\chi,\mathbf{x}}) - g_H^{(0)}(G_{\chi,\mathbf{x}'}) + \sum_{i=2}^{\lfloor \frac{h}{2} \rfloor} g_H^{(i)}(G_{\chi,\mathbf{x}}) - \sum_{i=2}^{\lfloor \frac{h}{2} \rfloor} g_H^{(i)}(G_{\chi,\mathbf{x}'}) \right|$$

$$\geq \left| g_H^{(1)}(G_{\chi,\mathbf{x}}) - g_H^{(1)}(G_{\chi,\mathbf{x}'}) \right| - \left| g_H^{(0)}(G_{\chi,\mathbf{x}}) - g_H^{(0)}(G_{\chi,\mathbf{x}'}) \right| - \left( \left| \sum_{i=2}^{\lfloor \frac{h}{2} \rfloor} g_H^{(i)}(G_{\chi,\mathbf{x}}) \right| + \left| \sum_{i=2}^{\lfloor \frac{h}{2} \rfloor} g_H^{(i)}(G_{\chi,\mathbf{x}'}) \right| \right)$$

$$:= (\mathrm{I}) - (\mathrm{II}) - (\mathrm{III}),$$

where the last inequality follows from the inequality $|a + b + c + d| \geq |a| - |b| - |c| - |d|$. In the following, we denote the above three terms by (I) (II) (III), and we bound them separately.

**Term (I)** We denote by $g_H^{(e)}(G_{\chi,\mathbf{x}})$ the number of occurrences of $H$ in $G_{\chi,\mathbf{x}}$ containing exactly one private edge $e \in \widetilde{E}_{\chi,\mathbf{x}}$. Then we have $g_H^{(1)}(G_{\chi,\mathbf{x}}) = \sum_{e \in E_{\chi,\mathbf{x}}} g_H^{(e)}(G_{\chi,\mathbf{x}})$ by definition. Note that for any $e_1, e_2 \in \widetilde{E}_{\chi,\mathbf{x}}$, it holds that $g_H^{(e_1)}(G_{\chi,\mathbf{x}}) = g_H^{(e_2)}(G_{\chi,\mathbf{x}})$ since all private edges are structurally symmetric in $G_{\chi,\mathbf{x}}$. Furthermore, once $e$ is fixed, $g_H^{(e)}(G_{\chi,\mathbf{x}})$ is determined solely by $n$ and $|V_{q_\chi}|$, and is independent of any private information in $G_{\chi,\mathbf{x}}$. Without loss of generality, let $g_H^{(e)}(G_{\chi,\mathbf{x}}) = \bar{g}_H(n, |V_{q_\chi}|)$ for any $e \in \widetilde{E}_{\chi,\mathbf{x}}$. Observe that $\bar{g}_H(n, |V_{q_\chi}|) \geq \mathrm{GS}_{f_H}$ as each private edge, together with the vertices in $W$, forms a $n$-vertex clique (see Figure 2). By combining the above properties, we can derive

$$\left| g_H^{(1)}(G_{\chi,\mathbf{x}}) - g_H^{(1)}(G_{\chi,\mathbf{x}'}) \right|$$

$$= \left| \sum_{e \in E_{\chi,\mathbf{x}}} g_H^{(e)}(G_{\chi,\mathbf{x}}) - \sum_{e \in E_{\chi,\mathbf{x}'}} g_H^{(e)}(G_{\chi,\mathbf{x}'}) \right|$$

$$= \left| \left| \widetilde{E}_{\chi,\mathbf{x}} \right| \cdot \bar{g}_H(n, |V_{q_\chi}|) - \left| \widetilde{E}_{\chi,\mathbf{x}'} \right| \cdot \bar{g}_H(n, |V_{q_\chi}|) \right|$$

$$= \left| (\mathbf{Ax})_{q_\chi} \cdot \bar{g}_H(n, |V_{q_\chi}|) - (\mathbf{Ax}')_{q_\chi} \cdot \bar{g}_H(n, |V_{q_\chi}|) \right|$$

$$= \bar{g}_H(n, |V_{q_\chi}|) \cdot \left| (\mathbf{A}(\mathbf{x} - \mathbf{x}'))_{q_\chi} \right|$$

$$\geq \mathrm{GS}_{f_H} \cdot \left| (\mathbf{A}(\mathbf{x} - \mathbf{x}'))_{q_\chi} \right|, \tag{9}$$

where the third equation follows from $|\widetilde{E}_{\chi,\mathbf{x}}| = (\mathbf{Ax})_{q_\chi}$ by definition.

**Term (II)** Observe that for any $G_{\chi,\mathbf{x}}$, the function $g_H^{(0)}(G_{\chi,\mathbf{x}})$ is independent of any private information in $G_{\chi,\mathbf{x}}$. Then it holds that

$$\left| g_H^{(0)}(G_{\chi,\mathbf{x}}) - g_H^{(0)}(G_{\chi,\mathbf{x}'}) \right| = 0. \tag{10}$$

**Term (III)** We first prove the following claim.

**Claim F.1.** *For any $G_{\chi,\mathbf{x}}$ and $2 \leq i \leq \lfloor \frac{h}{2} \rfloor$, we have $g_H^{(i)}(G_{\chi,\mathbf{x}}) = O(n^{h-i})$.*

*Proof.* The number of occurrences of $H$ in $G_{\chi,\mathbf{x}}$ that contain exactly $i$ private edges is equivalent to the following procedure: first, select $i$ private edges and fix the $2i$ endpoints of these edges; then, determine the positions of the remaining vertices in the occurrences of $H$. The number of ways to select $i$ private edges is at most $\binom{n}{i} \cdot \binom{m_H}{i} \cdot i! \cdot 2^i = O(n^i)$, while the number of ways to assign the remaining vertices is at most $\binom{3n-2-2i}{h-2i} \cdot (h-2i)! = O(n^{h-2i})$. Therefore, $g_H^{(i)}(G_{\chi,\mathbf{x}}) \leq O(n^i) \cdot O(n^{h-2i}) = O(n^{h-i})$. $\qquad\square$

Then by Claim F.1, we have

$$\left( \left| \sum_{i=2}^{\lfloor \frac{h}{2} \rfloor} g_H^{(i)}(G_{\chi,\mathbf{x}}) \right| + \left| \sum_{i=2}^{\lfloor \frac{h}{2} \rfloor} g_H^{(i)}(G_{\chi,\mathbf{x}'}) \right| \right) = \sum_{i=2}^{\lfloor \frac{h}{2} \rfloor} O(n^{h-i}) = O(n^{h-2}). \tag{11}$$

Substituting Inequality (9) and Equations (10) and (11) into Equation (8), we obtain

$$\begin{aligned}
&\mathrm{disc}_{C_{\alpha,H}}(\mathbf{A}_H) \\
&\geq \min_{\substack{\chi \in C_{\alpha,H} \\ \chi = \mathbf{x}_H - \mathbf{x}'_H}} \left( \mathrm{GS}_{f_H} \cdot \left| (\mathbf{A}(\mathbf{x} - \mathbf{x}'))_{q_\chi} \right| - O\left(n^{h-2}\right) \right) \\
&= \min_{\substack{\chi \in C_{\alpha,H} \\ \chi = \mathbf{x}_H - \mathbf{x}'_H}} \left( \mathrm{GS}_{f_H} \cdot \left| (\mathbf{A}(\mathbf{x} - \mathbf{x}'))_{q_\chi} \right| \right) - O\left(n^{h-2}\right) \\
&= \mathrm{GS}_{f_H} \cdot \min_{\chi \in C_\alpha} \left| (\mathbf{A}\chi)_{q_\chi} \right| - O\left(n^{h-2}\right) \\
&= \mathrm{GS}_{f_H} \cdot \mathrm{disc}_{C_\alpha}(\mathbf{A}) - O\left(n^{h-2}\right) \\
&= \Omega\left( \mathrm{GS}_{f_H} \cdot \mathrm{disc}_{C_\alpha}(\mathbf{A}) \right).
\end{aligned}$$

The antepenultimate equation follows from the definitions of $C_\alpha$ and $C_{\alpha,H}$; the penultimate equation follows from the definition of $\mathrm{disc}_{C_\alpha}(\mathbf{A})$; while the last equation follows from $\mathrm{disc}_{C_\alpha}(\mathbf{A}) = \omega(1)$ and $\mathrm{GS}_{f_H} = \Theta(n^{h-2})$ (see Lemma C.11).

### F.2. Proof of Theorem 4.1

The following lemma shows a connection between discrepancy and the reconstruction algorithm, which is similar to Lemma 10 in Muthukrishnan & Nikolov (2012).

**Lemma F.2.** *For any $\mathbf{x} \in \{0,1\}^n$, there is a deterministic (not necessarily efficient) algorithm $\mathcal{A}$ given an output $\mathbf{y} = \mathcal{M}(\mathbf{x})$ of some mechanism $\mathcal{M}$ such that $\|\mathbf{y} - \mathbf{A}_H \mathbf{x}_H\|_\infty < \frac{1}{2}\mathrm{disc}_{C_{\alpha,H}}(\mathbf{A}_H)$ satisfies $\|\mathcal{A}(\mathbf{y}) - \mathbf{x}\|_1 \leq \alpha n$.*

*Proof.* Given $\mathbf{y} = \mathcal{M}(\mathbf{x})$, the algorithm outputs an vector $\mathbf{x}' \in \{0,1\}^n$ such that $\|\mathbf{y} - \mathbf{A}_H \mathbf{x}'_H\| < \frac{1}{2}\mathrm{disc}_{C_{\alpha,H}}(\mathbf{A}_H)$. Note that such $\mathbf{x}'$ exists, since $\mathbf{x}$ already satisfies the required properties. For the sake of contradiction, we assume that $\|\mathbf{x}' - \mathbf{x}\|_1 > \alpha n$. Then $\chi = \mathbf{x}'_H - \mathbf{x}_H$ belongs to $C_{\alpha,H}$ and therefore $\|\mathbf{A}_H \chi\|_\infty \geq \mathrm{disc}_{C_{\alpha,H}}(\mathbf{A}_H)$. However, by the triangle inequality, we have $\|\mathbf{A}_H \chi\|_\infty \leq \|\mathbf{A}_H \mathbf{x}'_H - \mathbf{y}\|_\infty + \|\mathbf{A}_H \mathbf{x}_H - \mathbf{y}\|_\infty < \mathrm{disc}_{C_{\alpha,H}}(\mathbf{A}_H)$ — a contradiction. $\qquad\square$

We utilize the following lemma, a simplified version of the result in De (2012), as a lower bound for RA. This lemma shows that decoding most of the input from the output violates differential privacy.

**Lemma F.3** (Lemma 3.9 in Eden et al. (2023)). *If $\mathcal{B}$ is an $(\varepsilon, \delta)$-DP reconstruction algorithm for RA and $\mathbf{x}$ is uniformly distributed in $\{0,1\}^N$, then we have*

$$\mathbb{E}[\|\mathcal{B}(\mathbf{x}) - \mathbf{x}\|_1] \geq e^{-\varepsilon}\left(\frac{1}{2} - \delta\right)N,$$

*where the expectation is taken over the randomness of both $\mathbf{x}$ and $\mathcal{B}$.*

Finally, we finish the proof of Theorem 4.1.

*Proof of Theorem 4.1.* We first establish that $\eta = \Omega(\log^{d-1} \bar{n} \cdot \mathrm{GS}_{f_H})$ for $d = O(1)$.

Let $\mathbf{A}, \mathbf{A}_H, G(\mathbf{x}), Q$ denote the corresponding matrices, graph, and query set, respectively, as defined earlier. Then it holds that $\mathrm{disc}_{C_\alpha}(\mathbf{A}) = \Omega(\log^{d-1} \bar{n})$ for any constant $\alpha$. By Lemma 4.3, it follows that $\mathrm{disc}_{C_{\alpha,H}}(\mathbf{A}_H) = \Omega(\log^{d-1} \bar{n} \cdot \mathrm{GS}_{f_H})$.

For the sake of contradiction, we assume that there exists an $(\varepsilon, \delta)$-DP mechanism $\mathcal{M}(\mathbf{x})$ with constants $\varepsilon > 0$ and $\delta \in [0, \frac{1}{2})$ for RSC of $(3n-2)$-vertex graph relaxed to protect $n$ private edges, whose additive error is less than $\frac{1}{2}\mathrm{disc}_{C_{\alpha,H}}(\mathbf{A}_H)$ with a sufficiently large constant probability $\beta < 1$.

We apply $\mathcal{M}(\mathbf{x})$ to the instance $(G(\mathbf{x}), Q)$. By Lemma F.2, there is a deterministic (not necessarily efficient) algorithm $\mathcal{A}$ satisfying $\|\mathcal{A}(\mathcal{M}(\mathbf{x})) - \mathbf{x}\|_1 \leq \alpha n$ for any $\mathbf{x} \in \{0,1\}^n$. Let $\mathcal{B}(\mathbf{x}) = \mathcal{A}(\mathcal{M}(\mathbf{x}))$ denote the reconstruction algorithm for RA. By the post-processing property (Proposition C.1), the algorithm $\mathcal{B}(\mathbf{x})$ preserves $(\varepsilon, \delta)$-DP. If $\mathbf{x}$ is uniformly distributed in $\{0,1\}^n$, we have $\mathbb{E}[\|\mathcal{B}(\mathbf{x}) - \mathbf{x}\|_1] \leq (\beta\alpha + (1-\beta))n$.

We can always choose a sufficiently small constant $\alpha > 0$ and a sufficiently large constant $\beta < 1$ such that $\mathbb{E}[\|\mathcal{B}(\mathbf{x}) - \mathbf{x}\|_1] < e^{-\varepsilon}(\frac{1}{2} - \delta)n$, which yields a contradiction with Lemma F.3.

By appropriately rescaling and $\mathrm{GS}_{f_H} = \Theta(n^{h-2})$ (see Lemma C.11), we can obtain the same lower bound for DPRSC of $n$-vertex graphs.

Now consider the case for $d = \omega(1)$. The proofs for $d = O(\log \bar{n})$ and $d = \Omega(\log \bar{n})$ follow by applying the same techniques as that for $d = O(1)$. $\qquad\square$

## F.3. Proof of Theorem 1.4

The hard instances in Theorem 1.4 corresponds to the graph $G(\mathbf{x})$. We first prove $\widetilde{\mathrm{HS}}_{f_H}(G(\mathbf{x})) = \Theta(\mathrm{GS}_{f_H})$ with a sufficiently large constant probability $(1 - \beta') < 1$. It holds that $\widetilde{\mathrm{HS}}_{f_H}(G(\mathbf{x})) = \mathrm{GS}_{f_H} = 1$ if $m_H = 1$. Now we assume that $m_H \geq 2$.

Note that the graph $G(\mathbf{x})$ contains a $n$-vertex clique. Thus we have $f_H^{(1)}(G(\mathbf{x})) = \mathrm{LS}_{f_H}(G(\mathbf{x})) = \Theta(n^{h-2}) = \Theta(\mathrm{GS}_{f_H})$ by an analysis similar to the proof of Lemma C.11. Once we determine at least 2 edges of $G(\mathbf{x})$, we have determined at least 3 vertices in $G(\mathbf{x})$. Therefore, based on an analysis similar to the proof of Lemma C.11, we can conclude that $f_H^{(k)}(G(\mathbf{x})) = O(n^{h-3})$ for any $2 \leq k \leq m_H$.

Then we can prove that $\widetilde{\mathrm{HS}}_{f_H}^{(k)}(G(\mathbf{x})) = O(n^{h-3})$ with probability $(1 - \frac{(m_H - k)\beta'}{m_H - 1})$ for any $2 \leq k \leq m_H$ by induction. When $\widetilde{\mathrm{HS}}_{f_H}^{(k)}(G(\mathbf{x}))$ satisfies the given condition, we can infer that $\widetilde{\mathrm{HS}}_{f_H}^{(k+1)}(G(\mathbf{x}))$ also satisfies the condition from Algorithm 4, Fact 2.6, and the union bound. Finally, using the same approach, we derive the bound for $\widetilde{\mathrm{HS}}_{f_H}(G(\mathbf{x}))$: since $f^{(1)}(G(\mathbf{x}))$ dominates the quantity, we have $\widetilde{\mathrm{HS}}_{f_H}(G(\mathbf{x})) = \Theta(\mathrm{GS}_{f_H})$ with probability $(1 - \beta')$.

Now for $d = O(1)$, we aim to establish that $\eta = \Omega(\log^{d-1} \bar{n} \cdot \widetilde{\mathrm{HS}}_{f_H}(G(\mathbf{x})))$ with $\bar{n} = \min(n, |Q|)$, and then substitute $|Q| \geq n^c$ for any sufficiently small constant $c > 0$. For the sake of contradiction, we assume that there exists an $(\varepsilon, \delta)$-DP mechanism $\mathcal{M}(\mathbf{x})$ with constants $\varepsilon > 0$ and $\delta \in [0, \frac{1}{2})$ for RSC of $(3n-2)$-vertex graph relaxed to protect $n$ private edges, whose additive error is $o(\log^{d-1} \bar{n} \cdot \widetilde{\mathrm{HS}}_{f_H}(G(\mathbf{x})))$ with a sufficiently large constant probability $\gamma < 1$.

By choosing a sufficiently small constant $\beta' > 0$ and applying the union bound, it holds that $\mathcal{M}(\mathbf{x})$ has an additive error less than $\frac{1}{2}\mathrm{disc}_{C_{\alpha,H}}(\mathbf{A}_H) = \Omega(\log^{d-1} \bar{n} \cdot \mathrm{GS}_{f_H})$ with a sufficiently large constant probability $\beta = \gamma - \beta' < 1$. Then a contradiction can be derived through the same proof as Theorem 4.1. By appropriately rescaling, we can obtain the same lower bound for DPRSC of $n$-vertex graphs.

Now consider the case for $d = \omega(1)$. The proofs for $d = O(\log n)$ and $d = \Omega(\log n)$ follow by applying the same techniques as that for $d = O(1)$.

## F.4. A lower bound for the case $d = 1$

By combining our approach with the method of Dwork et al. (2010), we obtain a lower bound for DPRSC with $d = 1$. When $|Q| \geq n^c$ for any sufficiently small constant $c > 0$, a similar lower bound $\Omega(\log n \cdot \widetilde{\mathrm{HS}}_{f_H}(G))$ follows from the following theorem and the proof of Theorem 1.4.

**Theorem F.4.** *Let $\varepsilon > 0$ be a constant and $\delta = n^{-\Omega(1)}$. For any pattern graph $H$, let $\mathcal{A}$ be an $(\varepsilon, \delta)$-DP algorithm for the*

*range subgraph counting problem with additive error $\eta = \max_{q \in Q} |f_H(G_q) - \widetilde{f}_H(G_q)|$ with probability at least $\frac{2}{3}$.*

*Then there exist infinitely many $n$-vertex graphs $G = (V, E, \mathbf{a})$ with $1$-dimensional vertex attributes and a corresponding query set $Q$, such that $\eta = \Omega(\log \bar{n} \cdot \mathrm{GS}_{f_H})$, where $\bar{n} = \min(n, |Q|)$ can be arbitrarily large.*

*Proof.* Let $P$ denote a set of $\bar{n}$ vertices, where the attribute of the $i$-th vertex is $i$ for all $i \in [\bar{n}]$. Let $Q$ denote a set of $\bar{n}$ intervals, where the $i$-th interval is $[0, i]$ for all $i \in [\bar{n}]$. We can always adjust the size of $P$ by adding $n - \bar{n}$ dummy points lying outside all boxes in $Q$, or adjust the size of $Q$ by adding $|Q| - \bar{n}$ dummy intervals not containing any points. Therefore, it suffices to analyze the case for $|P| = n$ and $|Q| = \bar{n}$.

Note that all intervals in $Q$ share a common intersection at $0$. We construct an $(3n - 2)$-vertex graph $G(\mathbf{x})$ based on $P$ and $Q$ as Section 4.1. Assume that $\delta \in [0, n^{-c})$ for some small constant $c > 0$. Let $k = c_1 \ln \bar{n}$ for a constant $c_1 \in (0, \frac{\min(1,c)}{2\varepsilon})$.

Let $\mathrm{GS}_{f_H}$ denote the global sensitivity of $f_H$ on $n$-vertex graphs. For the sake of contradiction, assume that there exists an $(\varepsilon, \delta)$-DP algorithm $\mathcal{A}$ for RSC of $(3n - 2)$-vertex graph relaxed to protect $n$ private edges. The algorithm achieves additive error at most $c_2 \ln \bar{n} \cdot \mathrm{GS}_{f_H}$ with probability at least $\frac{2}{3}$, where $c_2$ is a constant with $c_2 \in (0, \frac{c_1}{6})$.

Without loss of generality, assume that $\frac{\bar{n}}{k}$ is an integer. We sort the $\bar{n}$ queries in $Q$ by their right endpoints and partition them into $\frac{\bar{n}}{k}$ phases, each consisting of $k$ consecutive queries. For the $s$-th phase, we construct a vector $\mathbf{x}^{(s)} \in \{0, 1\}^n$ whose coordinates are indexed by $r \in [n]$ such that

$$
x_r^{(s)} = \begin{cases} 1, & \text{if } r \in [n] \cap [ks - k + 1, ks]; \\ 0, & \text{otherwise.} \end{cases}
$$

Let $Q_{\mathrm{L}}^{(s)} = \{[0, r] \mid r \in [n] \cap [1, ks - k]\}$ and $Q_{\mathrm{R}}^{(s)} = \{[0, r] \mid r \in [n] \cap [ks, \bar{n}]\}$ denote the sets of all queries before and after the $s$-th phase, respectively. In particular, let $q_{\mathrm{L}}^{(s)}$ and $q_{\mathrm{R}}^{(s)}$ denote the interval $[0, ks - k]$ and $[0, ks]$, respectively.

For any $\mathbf{x} \in \{0, 1\}^n$ and $q \in Q$, we use $G_{\mathbf{x}, q}$ to denote $(G(\mathbf{x}))[V_q]$ for simplicity. Similar to the proof of Lemma 4.3, for any subgraph $G'$ of $G(\mathbf{x})$, let $g_H^{(i)}(G')$ denote the number of occurrences of $H$ in $G'$ containing exactly $i$ private edges. For any $s \in [\frac{\bar{n}}{k}]$ and $\mathbf{x} \in \{0, 1\}^n$, the output of $\mathcal{A}(\mathbf{x})$ *matches* the $s$-th phase if $\widetilde{f}_H(G_{\mathbf{x}, q}) - g_H^{(0)}(G_{\mathbf{x}, q}) < \frac{k}{2}\mathrm{GS}_{f_H}$ for all $q \in Q_{\mathrm{L}}^{(s)}$ and $\widetilde{f}_H(G_{\mathbf{x}, q}) - g_H^{(0)}(G_{\mathbf{x}, q}) \geq \frac{k}{2}\mathrm{GS}_{f_H}$ for all $q \in Q_{\mathrm{R}}^{(s)}$. Then we have the following claim.

**Claim F.5.** *For any $s \in [\frac{\bar{n}}{k}]$ and sufficiently large $\bar{n}$, the output of $\mathcal{A}(\mathbf{x}^{(s)})$ matches the $s$-th phase with probability at least $\frac{2}{3}$.*

*Proof.* By the definition of the algorithm $\mathcal{A}$, we have

$$
f_H(G_q) - c_2 \ln \bar{n} \cdot \mathrm{GS}_{f_H} \leq \widetilde{f}_H(G_{\mathbf{x}, q}) \leq f_H(G_q) + c_2 \ln \bar{n} \cdot \mathrm{GS}_{f_H}
$$

for all $q \in Q$ with probability at least $\frac{2}{3}$.

Then for any $s \in [\frac{\bar{n}}{k}]$ and all $q_1 \in Q_{\mathrm{L}}^{(s)}, q_2 \in Q_{\mathrm{R}}^{(s)}$, we have

$$
\left( \widetilde{f}_H\left(G_{\mathbf{x}^{(s)}, q_2}\right) - g_H^{(0)}\left(G_{\mathbf{x}^{(s)}, q_2}\right) \right) - \left( \widetilde{f}_H\left(G_{\mathbf{x}^{(s)}, q_1}\right) - g_H^{(0)}\left(G_{\mathbf{x}^{(s)}, q_1}\right) \right)
$$

$$
\geq \left( f_H\left(G_{\mathbf{x}^{(s)}, q_2}\right) - g_H^{(0)}\left(G_{\mathbf{x}^{(s)}, q_2}\right) - c_2 \ln \bar{n} \cdot \mathrm{GS}_{f_H} \right) - \left( f_H\left(G_{\mathbf{x}^{(s)}, q_1}\right) - g_H^{(0)}\left(G_{\mathbf{x}^{(s)}, q_1}\right) + c_2 \ln \bar{n} \cdot \mathrm{GS}_{f_H} \right)
$$

$$
\geq \left( f_H\left(G_{\mathbf{x}^{(s)}, q_{\mathrm{R}}^{(s)}}\right) - g_H^{(0)}\left(G_{\mathbf{x}^{(s)}, q_{\mathrm{R}}^{(s)}}\right) - c_2 \ln \bar{n} \cdot \mathrm{GS}_{f_H} \right) - \left( f_H\left(G_{\mathbf{x}^{(s)}, q_{\mathrm{L}}^{(s)}}\right) - g_H^{(0)}\left(G_{\mathbf{x}^{(s)}, q_{\mathrm{L}}^{(s)}}\right) + c_2 \ln \bar{n} \cdot \mathrm{GS}_{f_H} \right)
$$

$$
= \sum_{i=1}^{\lfloor \frac{h}{2} \rfloor} g_H^{(i)}\left(G_{\mathbf{x}^{(s)}, q_{\mathrm{R}}^{(s)}}\right) - \sum_{i=1}^{\lfloor \frac{h}{2} \rfloor} g_H^{(i)}\left(G_{\mathbf{x}^{(s)}, q_{\mathrm{L}}^{(s)}}\right) - 2c_2 \ln \bar{n} \cdot \mathrm{GS}_{f_H}
$$

$$
= \left( g_H^{(1)}\left(G_{\mathbf{x}^{(s)}, q_{\mathrm{R}}^{(s)}}\right) - g_H^{(1)}\left(G_{\mathbf{x}^{(s)}, q_{\mathrm{L}}^{(s)}}\right) \right) + \sum_{i=2}^{\lfloor \frac{h}{2} \rfloor} \left( g_H^{(i)}\left(G_{\mathbf{x}^{(s)}, q_{\mathrm{R}}^{(s)}}\right) - g_H^{(i)}\left(G_{\mathbf{x}^{(s)}, q_{\mathrm{L}}^{(s)}}\right) \right) - 2c_2 \ln \bar{n} \cdot \mathrm{GS}_{f_H}
$$

$$
\geq k\mathrm{GS}_{f_H} - 2c_2 \ln \bar{n} \cdot \mathrm{GS}_{f_H} > \frac{2}{3}k\mathrm{GS}_{f_H},
$$

with probability at least $\frac{2}{3}$, where the second inequality follows from the monotonicity of $f_H(G_{\mathbf{x}^{(s)},q}) - g_H^{(0)}(G_{\mathbf{x}^{(s)},q})$ for $q \in Q$ by definition; the first equation follows from the fact that $f_H(G') = \sum_{i=0}^{\lfloor \frac{h}{2} \rfloor} g_H^{(i)}(G')$, and the penultimate inequality follows from an analysis similar to the proof of Lemma 4.3. This completes the proof since

$$\widetilde{f}_H(G_{\mathbf{x}^{(s)},q}) - g_H^{(0)}(G_{\mathbf{x}^{(s)},q}) \leq f_H(G_{\mathbf{x}^{(s)},q}) - g_H^{(0)}(G_{\mathbf{x}^{(s)},q}) + c_2 \ln \bar{n} \cdot \mathrm{GS}_{f_H} < \frac{k}{6}\mathrm{GS}_{f_H},$$

$$\widetilde{f}_H(G_{\mathbf{x}^{(s)},q}) - g_H^{(0)}(G_{\mathbf{x}^{(s)},q}) \geq f_H(G_{\mathbf{x}^{(s)},q}) - g_H^{(0)}(G_{\mathbf{x}^{(s)},q}) - c_2 \ln \bar{n} \cdot \mathrm{GS}_{f_H} > -\frac{k}{6}\mathrm{GS}_{f_H}$$

for any $q \in Q_{\mathrm{L}}^{(s)}$. $\qquad\square$

For any $s, t \in [\frac{\bar{n}}{k}]$, let $E_{s,t}$ denote the event that the output of $\mathcal{A}(\mathbf{x}^{(s)})$ matches the $t$-th phase. By Claim F.5, we have $\Pr[E_{s,s}] \geq \frac{2}{3}$ for any $s \in [\frac{\bar{n}}{k}]$. By the definition of $(\varepsilon, \delta)$-DP, it holds that

$$\begin{aligned} \Pr[E_{s,t}] &\geq e^{-2\varepsilon k}\Pr[E_{t,t}] - \frac{e^{-\varepsilon}(1 - e^{-2\varepsilon k})}{1 - e^{-\varepsilon}}\delta \\ &> \frac{2}{3}\bar{n}^{-2c_1\varepsilon} - \frac{1 - \bar{n}^{-2c_1\varepsilon}}{e^{\varepsilon} - 1}\bar{n}^{-c} \\ &> \frac{2}{3}\bar{n}^{-2c_1\varepsilon} - \frac{\bar{n}^{-c}}{e^{\varepsilon} - 1}. \end{aligned}$$

Fix an $s \in [\frac{\bar{n}}{k}]$. This yields a contradiction: the events $\{E_{s,2t-1}\}_{t=1}^{\lceil \frac{\bar{n}}{2k} \rceil}$ for different $t$ are disjoint, yet their total probability

$$\sum_{t=1}^{\lceil \frac{\bar{n}}{2k} \rceil} \Pr[E_{s,2t-1}] \geq \frac{\frac{2}{3}\bar{n}^{1-2c_1\varepsilon} - \frac{\bar{n}^{1-c}}{e^{\varepsilon}-1}}{2c_1 \ln \bar{n}}$$

is more than 1 for sufficiently large $\bar{n}$. By appropriately rescaling, we can obtain the same lower bound for pure DPRSC of $n$-vertex graphs. $\qquad\square$

# G. Deferred Experiments from Section 5

## G.1. Deferred experiments when $H$ is an edge

In this subsection, we present the experimental results when $H$ is an edge. Figures 7 to 10 report the relationship between the relative error and $\varepsilon$, the relationship between the relative error and $|Q|$, the query time and preprocessing time, and the total time, respectively.

For edge counting, Algorithm 4 used in ADP_comp and ADP_RSC is meaningless since the global sensitivity is 1. In practice, we directly use $\mathrm{GS}_{f_H} = 1$ and apply the advanced composition theorem (Proposition 2.4). The baselines require no preprocessing time for similar reasons and compute exact counts of edges using a straightforward implementation in $O(m)$ time. The experimental results obtained are similar to those presented in Section 5.

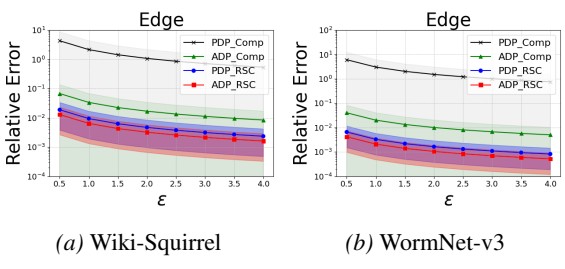

*(a)* Wiki-Squirrel

*(b)* WormNet-v3

*Figure 7.* Relative error vs. $\varepsilon$

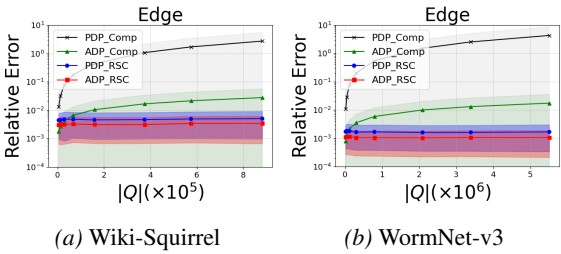

*(a)* Wiki-Squirrel

*(b)* WormNet-v3

*Figure 8.* Relative error vs. $|Q|$

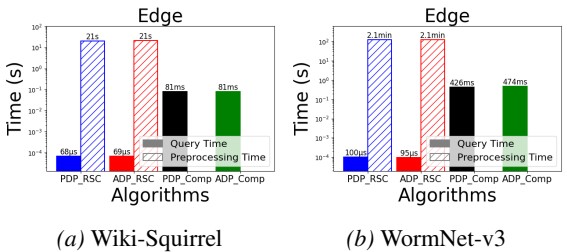

*(a)* Wiki-Squirrel      *(b)* WormNet-v3

*Figure 9.* Query time and preprocessing time

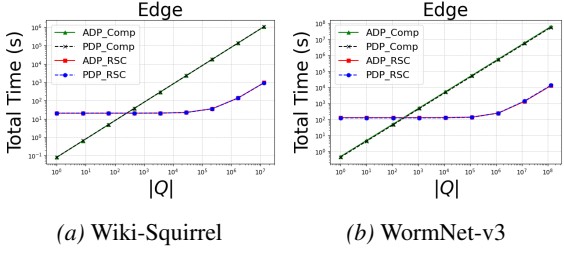

*(a)* Wiki-Squirrel      *(b)* WormNet-v3

*Figure 10.* Total time

### G.2. Deferred experiments when $d > 1$

We report results for the case $d > 1$ on the relatively small CA-Netscience dataset in this subsection. Due to the increased computational cost, experiments on larger datasets become prohibitively expensive. We set $|Q| = n^2$ and compare the cases for $d = 1, 2$ as shown in Figures 11 to 14.

These figures show that both the additive error and the runtime of our algorithms increase significantly as $d$ grows, with the observed scaling largely matching our theoretical analysis. While our algorithms no longer outperform the baselines on this dataset when $d = 2$, the behavior is consistent with our theoretical expectations: although our method improves the asymptotic dependence, its advantage only becomes apparent in regimes where $n$ and $|Q|$ are large. Additionally, we observe that our approximate DP algorithm exhibits a more pronounced advantage over the pure DP algorithm when $d = 2$, consistent with their different constant factors in the exponent with respect to $d$ in Theorems 1.3 and 3.3.

### G.3. Deferred experiments for small-range queries

In this subsection, we report results for small-range queries with $|V_q| = O(\sqrt{|Q|})$ on WormNet-v3, and show the additive error with respect to $\varepsilon$ in Figure 15. We observe that our algorithms maintain a significant advantage over the baselines on small-range queries. Meanwhile, both the mean and variance of the relative error increase substantially for all methods, due to the reduced magnitude and higher variability of the true answers in this regime.

### G.4. Deferred implementation details and experiments for real attributes

We note that in many real-world applications, attributes often exhibit significant redundancy, with many duplicate values along each dimension. In such cases, the effective domain size per attribute is small, which can, in fact, improve the performance of our algorithms. In contrast, the setting with independently sampled continuous attributes represents a more challenging regime, where the values in each attribute dimension are almost surely all distinct. Thus, the setting used in the main text can be viewed as a conservative (worst-case) evaluation of our algorithms.

From a theoretical perspective, this observation can be made precise. The analysis of both the additive error and the time and space complexity can be refined to depend on the effective domain size of each attribute. In particular, the factor $\log^{O(d)} n$ can be replaced by $(\prod_{i=1}^{d}(\lceil \log |A_i(V, \mathbf{a})| \rceil + 1))^{O(1)}$, where $A_i(V, \mathbf{a})$ denotes the set of distinct values of the $i$-th attribute (as defined before Definition 3.1). This refinement follows from the fact that the rank $s_i(v)$ in Section 3 naturally takes values in $[|A_i(V, \mathbf{a})|]$ rather than $[n]$. Consequently, the dependence on $\log n$ in the noise parameters of our algorithms can be replaced by $(\lceil \log |A_i(V, \mathbf{a})| \rceil + 1)$, yielding improved bounds for both the additive error and the time and space complexity. In particular, by a minor adaptation of the proofs of Theorems 1.3 and E.9, we obtain the following corollary. A similar corollary can also be derived from Theorems 3.3 and E.8.

**Corollary G.1.** *Let $L(V, \mathbf{a}) = \prod_{i=1}^{d}(\lceil \log |A_i(V, \mathbf{a})| \rceil + 1)$ and $M(V, \mathbf{a}) = \prod_{i=1}^{d}(2|A_i(V, \mathbf{a})| - 1)$. For any $\varepsilon > 0$ and $\delta \in (0, 1)$, there exists a $(\varepsilon, \delta)$-DP algorithm that, given a graph $G = (V, E, \mathbf{a})$ with $d$-dimensional vertex attributes, a fixed pattern graph $H$, and a query set $Q$, outputs noisy answers $\widetilde{f}_H(G_q)$ which satisfy*

$$\max_{q \in Q} \left| f_H(G_q) - \widetilde{f}_H(G_q) \right| = O\left( \frac{\widetilde{\mathrm{HS}}_{f_H}(G) \cdot \sqrt{(\varepsilon + \log(1/\delta)) \log(n|Q|)} \cdot L^2(V, \mathbf{a})}{\varepsilon} \right)$$

*with probability at least $1 - \frac{1}{n}$. The algorithm requires $O(\min(M^2(V, \mathbf{a}), (f_H(G) + |Q|)L^2(V, \mathbf{a})))$ space and $O(\min(f_H(G)L^2(V, \mathbf{a}), M^2(V, \mathbf{a}) \log^{2d} M^2(V, \mathbf{a})) + F_H(G))$ preprocessing time, and each query can be answered in*

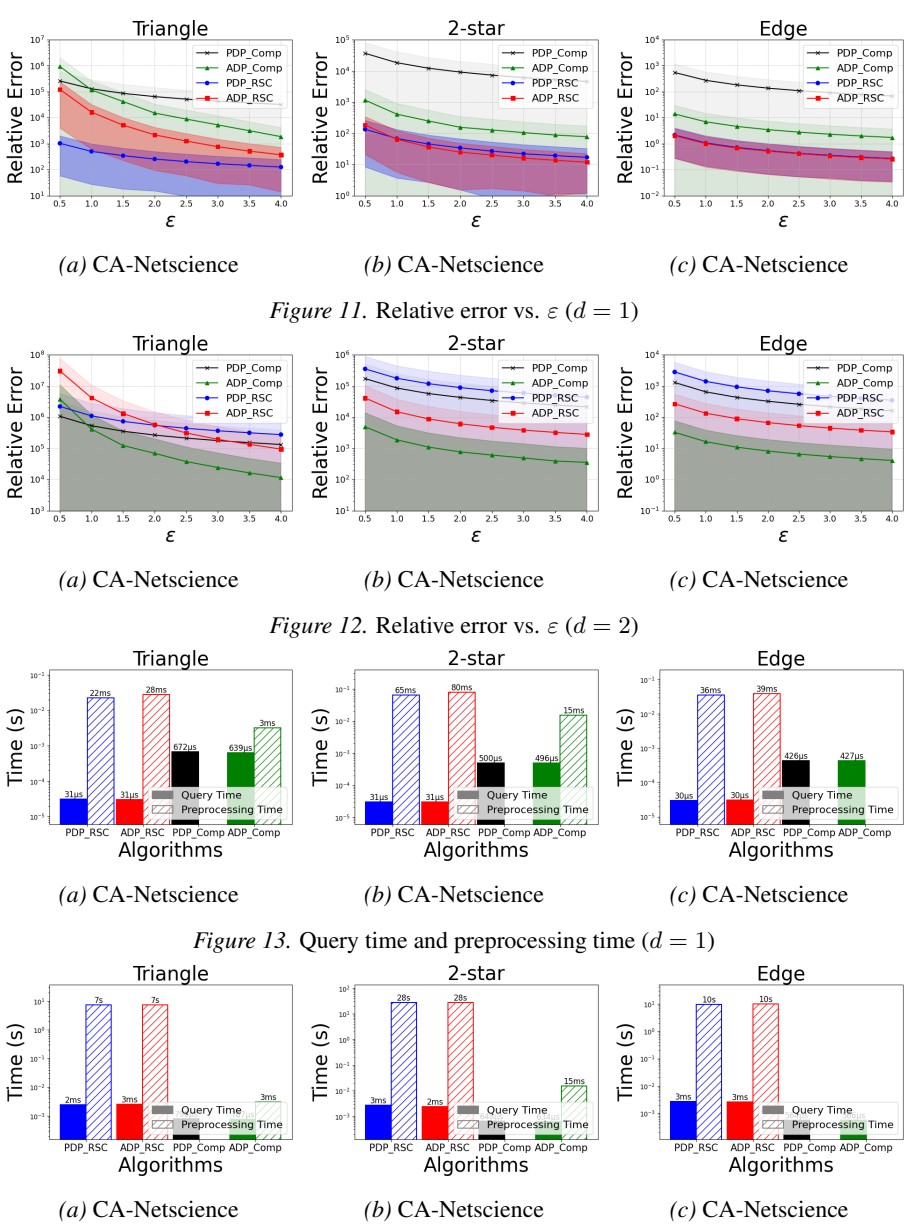

*Figure 11.* Relative error vs. $\varepsilon$ ($d = 1$)

*Figure 12.* Relative error vs. $\varepsilon$ ($d = 2$)

*Figure 13.* Query time and preprocessing time ($d = 1$)

*Figure 14.* Query time and preprocessing time ($d = 2$)

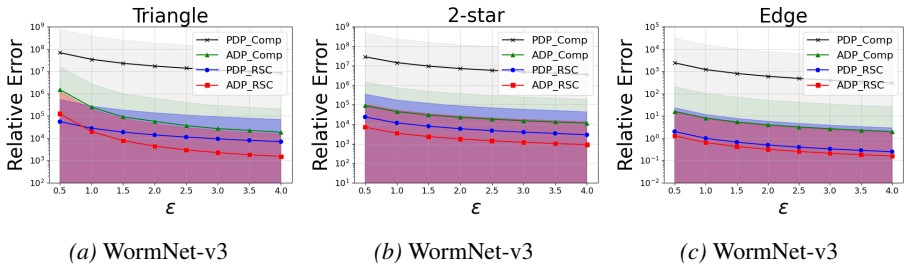

*Figure 15.* Relative error vs. $\varepsilon$ (small-range queries with $|V_q| = O(\sqrt{|Q|})$)

$O(L^2(V, \mathbf{a}))$ *time. Here, the hidden constants are of the form* $c^{O(d)}$ *for some universal constant* $c > 1$.

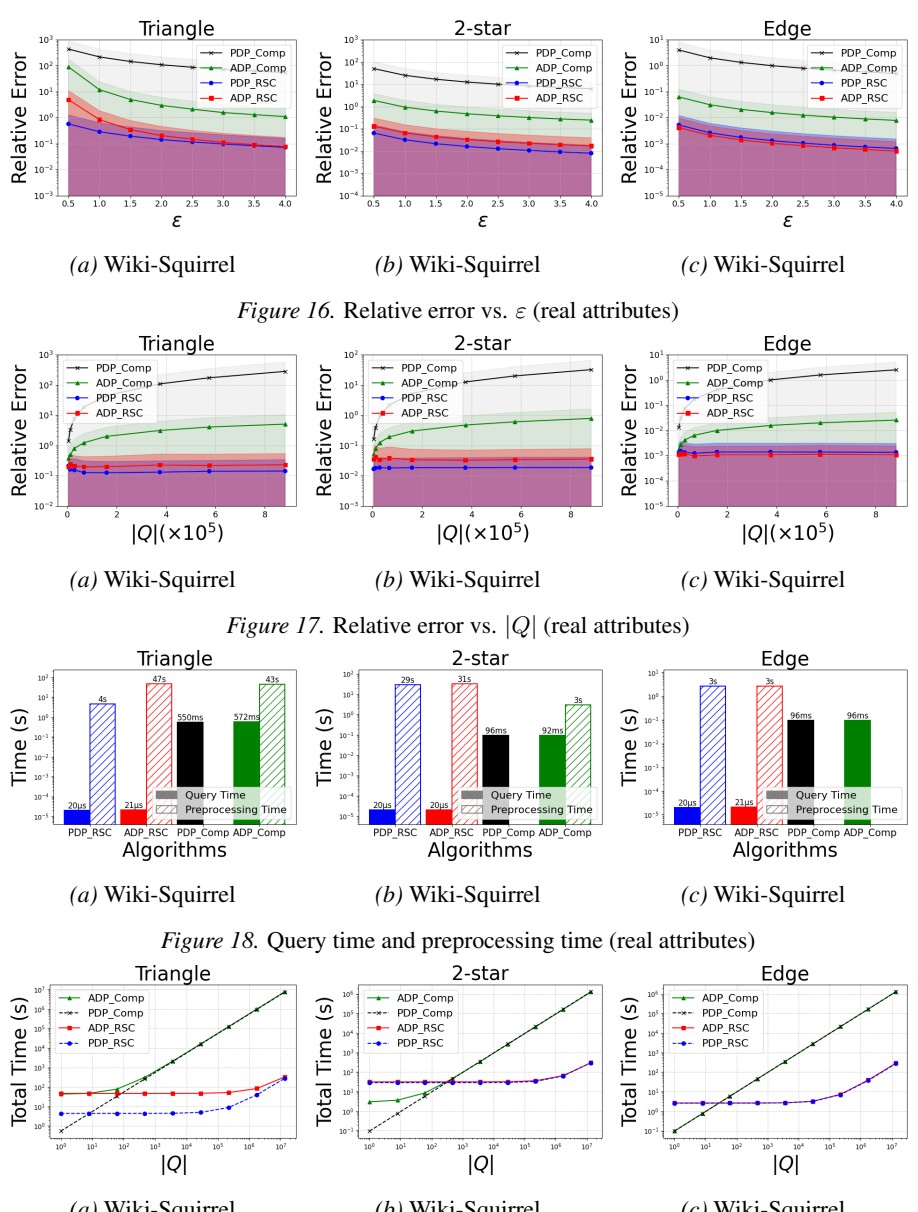

*Figure 16.* Relative error vs. $\varepsilon$ (real attributes)

*Figure 17.* Relative error vs. $|Q|$ (real attributes)

*Figure 18.* Query time and preprocessing time (real attributes)

*Figure 19.* Total time (real attributes)

Due to the scarcity of suitable real-world graph datasets with attributes aligned with our setting, we conduct experiments on Wiki-Squirrel with real attributes satisfying $|A_1(V, \mathbf{a})| = 2002$. We present the results in Figures 16 to 19. These figures show that our algorithms remain significantly better than the baselines for real attributes. Moreover, due to the smaller effective domain size, our algorithms achieve substantially improved performance compared to the setting where attributes are independently sampled from a standard normal distribution, which is consistent with our theoretical analysis.

## H. Extensions

### H.1. Randomized response method

In this subsection, we show that by using the randomized response, we obtain polynomial-time DP algorithms for range subgraph counting. Let $\mathbf{1}_k$ denote the $k$-bit all-ones vector.

**Definition H.1** (Randomized Response, Kasiviswanathan et al. (2011); Warner (1965)). Given a privacy parameter $\varepsilon > 0$

---

**Algorithm 6** UB_RR $(\mathbf{x}, \varepsilon)$      ▷ Unbiased Randomized Response

---

1: **Input:** A $k$-bit vector $\mathbf{x}$ and privacy parameter $\varepsilon$.
2: **return** $\frac{(e^\varepsilon+1)\mathrm{RR}(\mathbf{x},\varepsilon)-\mathbf{1}_k}{e^\varepsilon-1}$.

---

and a $k$-bit vector $\mathbf{v}$, the algorithm $\mathrm{RR}_\varepsilon(\mathbf{v})$ outputs a $k$-bit vector, where for each $i \in [k]$, bit $i$ is $v_i$ with probability $\frac{e^\varepsilon}{e^\varepsilon+1}$ and $1 - v_i$ otherwise.

**Lemma H.2.** *Randomized Response is an $\varepsilon$-DP.*

We apply a linear transformation to the output of randomized response and obtain an unbiased estimation of the input vector, which is described in Algorithm 6. Our subsequent algorithms mainly rely on the result of Algorithm 6.

**Lemma H.3.** *Algorithm 6 is a $\varepsilon$-DP and returns an unbiased estimate of the input vector $\mathbf{x}$.*

*Proof.* By Lemma H.2 and the post-processing property (Proposition C.1), Algorithm 6 preserves $\varepsilon$-DP. Let $\mathbf{y}$ be the output of Algorithm 6. For each $e \in E$, note that

$$\mathbb{E}[y_e] = \frac{(e^\varepsilon+1)\mathbb{E}[\widetilde{x}_e]-1}{e^\varepsilon-1} = \frac{(e^\varepsilon+1)\left(\frac{e^\varepsilon x_e}{e^\varepsilon+1}+\frac{1-x_e}{e^\varepsilon+1}\right)-1}{e^\varepsilon-1} = x_e.$$

That is, $\mathbf{y}$ is an unbiased estimate of $\mathbf{x}$. $\qquad\square$

### H.1.1. The first bound

Let $\mathbf{x} \in \{0,1\}^{\binom{n}{2}}$ be an indicator vector of $E$. Let $\mathcal{H}_V$ be the set of all occurrences of $H$ in the complete graph with vertex set $V$. For simplicity, we denote by $V(H')$ and $E(H')$ the vertex and edge sets of $H'$, respectively. Note that for each query $q \in Q$ and the corresponding induced vertex set $V_q$, we have

$$f_H(G_q) = \sum_{H' \in \mathcal{H}_{V_q}} \prod_{e \in E(H')} x_e. \tag{12}$$

In Algorithm 7, we construct the noisy vector $\mathbf{y} = \mathrm{UB\_RR}(\mathbf{x}, \varepsilon)$. And we replace $x_e$ in Equation (12) with $y_e$ as the output of Algorithm 7. We will make use of the following lemma to analyze Algorithm 7.

**Definition H.4** (Martingale). A sequence of real-valued random variables $Z_0, Z_1, \ldots, Z_t$ is a *martingale* with respect to a sequence of random variables $\xi_1, \ldots, \xi_t$ if for all $i \in [t]$, it holds that $\mathbb{E}[Z_i \mid \xi_1, \ldots, \xi_{i-1}] = Z_{i-1}$.

**Lemma H.5** (Azuma's Inequality, Azuma (1967)). *Let $Z_0, \ldots, Z_t$ be a martingale satisfying $|Z_i - Z_{i-1}| \leq c_i$ for any $i \in [t]$. Then for any $\lambda > 0$,*

$$\Pr[|Z_t - Z_0| \geq \lambda] \leq 2\exp\left(\frac{-\lambda^2}{2\left(\sum_{i=1}^t c_i^2\right)}\right)$$

**Theorem H.6.** *For any $\varepsilon > 0$, there exists a polynomial-time $\varepsilon$-DP algorithm that, given a graph $G = (V, E, \mathbf{a})$, where the attribute of each vertex is a $d$-dimensional vector, a pattern graph $H$, a query set $Q$, outputs all range subgraph counting queries which satisfy*

$$\max_{q \in Q}\left|f_H(G_q) - \widetilde{f}_H(G_q)\right| = O\left(\frac{n\sqrt{\log(n|Q|)}\mathrm{GS}_{f_H}}{\varepsilon^{m_H}}\right)$$

*with probability at least $1 - \frac{1}{n}$.*

*Proof.* By Lemma H.3 and the post-processing property (Proposition C.1), the entire algorithm is also $\varepsilon$-DP.

Then we analyze the additive error of Algorithm 7. We use $\binom{A}{k}$ to denote the set of all subsets of $A$ with size $k$. By Lemma H.3, let $\mathcal{Y}_e = y_e - x_e$ with $\mathbb{E}[\mathcal{Y}_e] = 0$ and $|\mathcal{Y}_e| \leq \frac{e^\varepsilon}{e^\varepsilon-1} \leq O(\frac{1}{\varepsilon})$ for each $e \in E$. Note that for each induced

---

**Algorithm 7** RR_RSC $(G, H, Q, \varepsilon)$      ▷ Randomized Response for Range Subgraph Counting

1: **Input:** An $n$-vertex graph $G = (V, E, \mathbf{a})$, a pattern graph $H$, a set of range queries $Q$, and privacy parameter $\varepsilon$.
2: Let $\mathbf{x}$ be an indicator vector of $E$ and $\mathbf{y} = \mathrm{UB\_RR}(\mathbf{x}, \varepsilon)$.
3: **for** each query $q \in Q$ **do**
4:     output $\widetilde{f}_H(G_q) = \sum_{H' \in \mathcal{H}_{V_q}} \prod_{e \in E(H')} y_e$.

---

subgraph $G_q$, we obtain

$$
\left| \widetilde{f}_H(G_q) - f_H(G_q) \right| = \left| \sum_{H' \in \mathcal{H}_{V_q}} \left( \prod_{e \in E(H')} (x_e + \mathcal{Y}_e) - \prod_{e \in E(H')} x_e \right) \right|
$$

$$
= \left| \sum_{H' \in \mathcal{H}_{V_q}} \sum_{i=1}^{m_H} \sum_{S \in \binom{E(H')}{i}} \prod_{e \in S} \mathcal{Y}_e \prod_{e' \in E(H') \setminus S} x_{e'} \right|
$$

$$
= \left| \sum_{i=1}^{m_H} \sum_{H' \in \mathcal{H}_{V_q}} \sum_{S \in \binom{E(H')}{i}} \prod_{e \in S} \mathcal{Y}_e \prod_{e' \in E(H') \setminus S} x_{e'} \right|,
$$

where the second and third equations follow from the binomial expansion and rearranging the summation order, respectively. For each $i \in [m_H]$, define

$$
w_{q,i} = \sum_{H' \in \mathcal{H}_{V_q}} \sum_{S \in \binom{E(H')}{i}} \prod_{e \in S} \mathcal{Y}_e \prod_{e' \in E(H') \setminus S} x_{e'}.
$$

Although $w_{q,i}$ is not a sum of independent random variables, we can still bound it by constructing an edge-exposure martingale applying Lemma H.5. Fix an arbitrary order of $\binom{n}{2}$ edges in the complete graph with vertex set $V$, i.e., $e_1, e_2, \ldots, e_{\binom{n}{2}}$. For any $j \in \left[ \binom{n}{2} \right]$, let $\mathcal{E}_j$ denote the set consisting of the first $j$ edges in this ordering, i.e., $\mathcal{E}_j = \{ e_1, e_2, \ldots, e_j \}$. Assume that $e_j = (u_j, v_j)$ and denote by $\mathcal{H}_{A,B}$ the set of all occurrences $H'$ of $H$ in $K_n$ whose vertex set satisfies $A \subseteq V(H') \subseteq B$. Let $X_{i,0} = 0$, and define

$$
X_{i,j} = X_{i,j-1} + \mathcal{Y}_{e_j} \sum_{H' \in \mathcal{H}_{\{u_j, v_j\}, V_q}} \sum_{S \in \binom{E(H') \setminus \{e_j\}}{i-1}} \prod_{e \in S} \mathcal{Y}_e \prod_{e' \in E(H') \setminus \{e_j\} \setminus S} x_{e'}.
$$

Note that $X_{i,0}, X_{i,1}, \ldots, X_{i,\binom{n}{2}}$ is a martingale satisfying $|X_{i,j} - X_{i,j-1}| \leq O(\mathrm{GS}_{f_H}/\varepsilon^i)$ for any $j \in \left[ \binom{n}{2} \right]$ since

$$
\mathbb{E}\left[ X_{i,j} \mid \mathcal{Y}_{e_1}, \mathcal{Y}_{e_2}, \ldots, \mathcal{Y}_{e_{j-1}} \right]
$$
$$
= X_{i,j-1} + \mathbb{E}\left[ \mathcal{Y}_{e_j} \right] \sum_{H' \in \mathcal{H}_{\{u_j, v_j\}, V_q}} \sum_{S \in \binom{E(H') \setminus \{e_j\}}{i-1}} \prod_{e \in S} \mathcal{Y}_e \prod_{e' \in E(H') \setminus S} x_{e'}
$$
$$
= X_{i,j-1},
$$

and we have $X_{i,\binom{n}{2}} = w_{q,i}$. By applying Lemma H.5, with probability at least $1 - \frac{1}{n m_H |Q|}$, we have

$$
\left| X_{i,\binom{n}{2}} - X_{i,0} \right| = |w_{q,i}| \leq O\left( \frac{n \sqrt{\log(n|Q|)} \mathrm{GS}_{f_H}}{\varepsilon^i} \right).
$$

Then, by the union bound, it holds that

$$
\max_{q \in Q} \left| f_H(G_q) - \widetilde{f}_H(G_q) \right| \leq \max_{q \in Q} \left| \sum_{i=1}^{m_H} w_{q,i} \right| \leq \max_{q \in Q} \sum_{i=1}^{m_H} |w_{q,i}|
$$

$$
\leq \max_{q \in Q} \sum_{i=1}^{m_H} O\left( \frac{n\sqrt{\log(n|Q|)} \mathrm{GS}_{f_H}}{\varepsilon^i} \right)
$$

$$
= O\left( \frac{n\sqrt{\log(n|Q|)} \mathrm{GS}_{f_H}}{\varepsilon^{m_H}} \right)
$$

with probability at least $1 - \frac{1}{n}$. $\qquad\square$

### H.1.2. **The second bound**

For moderate $|Q|$ (i.e., $|Q| = 2^{n^{o(1)}}$), we extend the methods in Eden et al. (2023) and obtain an instance-dependent error for the private range subgraph counting problem. We sketch the analysis in this subsection and present the entire algorithm in Algorithm 8.

---

**Algorithm 8** RR_IE $(G, H, Q, \varepsilon)$ $\qquad \triangleright$ Randomized Response with Instance-dependent Error

---

1: **Input:** An $n$-vertex graph $G = (V, E, \mathbf{a})$, a pattern graph $H$, a set of range queries $Q$, and privacy parameter $\varepsilon$.
2: Let $t = \lceil 18 \ln(n|Q|) \rceil$.
3: Let $\mathbf{x}$ be an indicator vector of $E$ and $\mathbf{y}^{(1)}, \mathbf{y}^{(2)}, \ldots, \mathbf{y}^{(t)}$ be $t$ independent outputs of UB_RR$(\mathbf{x}, \frac{\varepsilon}{t})$.
4: **for** each query $q \in Q$ **do**
5: $\quad$ Initialize $\widetilde{z}^{(i)} = \sum_{H' \in \mathcal{H}_{V_q}} \prod_{e \in E(H')} y_e^{(i)}$ for any $i \in [t]$.
6: $\quad$ Output $\widetilde{f}_H(G_q) = \mathrm{median}_{i \in [t]} \widetilde{z}^{(i)}$.

---

For any two sets $A$ and $B$, define their symmetric difference as $A \triangle B = (A \setminus B) \cup (B \setminus A)$. Given two graphs $G_1 = (V_1, E_1)$ and $G_2 = (V_2, E_2)$, their symmetric difference is defined as $G_1 \triangle G_2 = (V_1 \cup V_2, E_1 \triangle E_2)$. We write $G_1 \cong G_2$ if graphs $G_1$ and $G_2$ are isomorphic. For a fixed graph $H$ and an integer $i \in [m_H]$, let $\mathcal{G}_H^{(i)}$ denote the class of graphs that can be obtained as the symmetric difference of two isomorphic copies $H_1, H_2$ of $H$ whose edge sets intersect in exactly $i$ edges, i.e., $\mathcal{G}_H^{(i)} = \{ H_1 \triangle H_2 \mid H_1, H_2 \cong H, |E(H_1) \cap E(H_2)| = i \}$. Then we have the following theorem.

**Theorem H.7.** *For any $\varepsilon > 0$, there exists a polynomial-time $\varepsilon$-DP algorithm that, given a graph $G = (V, E, \mathbf{a})$, where the attribute of each vertex is a $d$-dimensional vector ($d = n^{o(1)}$), a pattern graph $H$, a query set $Q$, outputs all range subgraph counting queries which satisfy*

$$
\max_{q \in Q} \left| f_H(G_q) - \widetilde{f}_H(G_q) \right| = O\left( \left( \frac{\tau}{\varepsilon} \right)^{m_H} n^{\frac{h}{2}} + \sum_{i=1}^{m_H - 1} \left( \frac{\tau}{\varepsilon} \right)^i \sum_{H' \in \mathcal{G}_H^{(i)}} \sqrt{f_{H'}(G)} \right)
$$

*with probability at least $1 - \frac{1}{n}$, where $\tau = \log(n|Q|) \leq O(d \log n)$.*

*Proof.* We begin by analyzing the privacy guarantees of Algorithm 8. Let $\varepsilon' = \frac{\varepsilon}{t}$. By Lemma H.3, releasing $y^{(i)}$ preserves $\varepsilon'$-DP for each $i \in [t]$. By the basic composition theorem (Proposition 2.3), the entire algorithm preserves $\varepsilon$-DP.

Now we analyze the additive error of Algorithm 8. We have the following claim.

**Claim H.8.** *For any $q \in Q$ and $i \in [t]$, we have*

$$
\left| \widetilde{z}^{(i)} - f_H(G_q) \right| = O\left( \frac{n^{\frac{h}{2}}}{\varepsilon'^{m_H}} + \sum_{i=1}^{m_H - 1} \frac{1}{\varepsilon'^i} \sum_{H' \in \mathcal{G}_H^{(i)}} \sqrt{f_{H'}(G)} \right)
$$

*with probability at least $\frac{2}{3}$.*

*Proof.* Fix any $i \in [t]$ and $q \in Q$. Let $\widetilde{\mathbf{x}}^{(i)} = \mathrm{RR}(\mathbf{x}, \varepsilon')$ and $\mathbf{y}^{(i)} = ((e^{\varepsilon'} + 1)\widetilde{\mathbf{x}}^{(i)} - \mathbf{1}_{\binom{n}{2}})/(e^{\varepsilon'} - 1)$. For any $e \in E$, we have

$$\mathrm{Var}\left[y_e^{(i)}\right] = \frac{(e^{\varepsilon'} + 1)^2}{(e^{\varepsilon'} - 1)^2}\mathrm{Var}\left[\widetilde{x}_e^{(i)}\right] = \frac{(e^{\varepsilon'} + 1)^2}{(e^{\varepsilon'} - 1)^2}\frac{e^{\varepsilon'}}{(e^{\varepsilon'} + 1)^2} = \frac{e^{\varepsilon'}}{(e^{\varepsilon'} - 1)^2} = O\left(\frac{1}{\varepsilon'^2}\right),$$

since $\widetilde{x}_e$ is a Bernoulli variable. By Lemma H.3, it follows that

$$\mathbb{E}\left[\left(y_e^{(i)}\right)^2\right] = \mathrm{Var}\left[y_e^{(i)}\right] + \left(\mathbb{E}\left[y_e^{(i)}\right]\right)^2 = O\left(\frac{1}{\varepsilon'^2}\right) + x_e^2 = O\left(\frac{1}{\varepsilon'^2}\right). \tag{13}$$

Let $z_{H'}^{(i)} = \prod_{e \in E(H')} x_e^{(i)}$ and $\widetilde{z}_{H'}^{(i)} = \prod_{e \in E(H')} y_e^{(i)}$ for any fixed occurrence $H'$. Thus,

$$\mathbb{E}\left[\widetilde{z}_{H'}^{(i)}\right] = \prod_{e \in E(H')} \mathbb{E}\left[y_e^{(i)}\right] = \prod_{e \in E(H')} x_e^{(i)} = z_{H'}^{(i)}.$$

Then we have

$$\mathrm{Var}\left[\widetilde{z}_{H'}^{(i)}\right] = \mathbb{E}\left[\left(\widetilde{z}_{H'}^{(i)}\right)^2\right] - \left(\mathbb{E}\left[\widetilde{z}_{H'}^{(i)}\right]\right)^2 = \prod_{e \in E(H')} \mathbb{E}\left[\left(y_e^{(i)}\right)^2\right] - z_{H'}^{(i)} = O\left(\frac{1}{\varepsilon'^{2m_H}}\right). \tag{14}$$

Combining the above results, we obtain

$$\mathrm{Var}\left[\widetilde{z}^{(i)}\right] = \sum_{H' \in \mathcal{H}_{V_q}} \mathrm{Var}\left[\widetilde{z}_{H'}^{(i)}\right] + \sum_{H_1, H_2 \in \mathcal{H}_{V_q}} \mathbb{E}\left[\left(\widetilde{z}_{H_1}^{(i)} - \mathbb{E}\left[\widetilde{z}_{H_1}^{(i)}\right]\right)\left(\widetilde{z}_{H_2}^{(i)} - \mathbb{E}\left[\widetilde{z}_{H_2}^{(i)}\right]\right)\right]$$

$$= \sum_{H' \in \mathcal{H}_{V_q}} \mathrm{Var}\left[\widetilde{z}_{H'}^{(i)}\right] + \sum_{H_1, H_2 \in \mathcal{H}_{V_q}} \left(\mathbb{E}\left[\widetilde{z}_{H_1}^{(i)} \widetilde{z}_{H_2}^{(i)}\right] - z_{H_1}^{(i)} z_{H_2}^{(i)}\right)$$

$$\leq \sum_{H' \in \mathcal{H}_{V_q}} \mathrm{Var}\left[\widetilde{z}_{H'}^{(i)}\right] + \sum_{H_1, H_2 \in \mathcal{H}_{V_q}} \mathbb{E}\left[\widetilde{z}_{H_1}^{(i)} \widetilde{z}_{H_2}^{(i)}\right]$$

$$= O\left(\frac{f_H(G_q)}{\varepsilon'^{2m_H}}\right) + \sum_{H_1, H_2 \in \mathcal{H}_{V_q}} \prod_{e \in E(H_1) \triangle E(H_2)} \mathbb{E}\left[y_e^{(i)}\right] \prod_{e' \in E(H_1) \cap E(H_2)} \mathbb{E}\left[\left(y_{e'}^{(i)}\right)^2\right]$$

$$= O\left(\frac{n^h}{\varepsilon'^{2m_H}}\right) + \sum_{i=1}^{m_H - 1} O\left(\frac{1}{\varepsilon'^{2i}}\right) \sum_{\substack{H_1, H_2 \in \mathcal{H}_{V_q} \\ |E(H_1) \cap E(H_2)| = i}} \prod_{e \in E(H_1) \triangle E(H_2)} x_e$$

$$= O\left(\frac{n^h}{\varepsilon'^{2m_H}} + \sum_{i=1}^{m_H - 1} \frac{1}{\varepsilon'^{2i}} \sum_{H' \in \mathcal{G}_H^{(i)}} f_{H'}(G)\right).$$

The antepenultimate equation follows from Equation (14); the penultimate equation follows from Lemma C.12 and Equation (13); while the last equation follows from the definition of $\mathcal{G}_H^{(i)}$. Note that $\mathbb{E}[\widetilde{z}^{(i)}] = \sum_{H' \in \mathcal{H}_{V_q}} \mathbb{E}[\widetilde{z}_{H'}^{(i)}] = f_H(G_q)$. By Chebyshev's Inequality, we finish the proof. $\square$

By applying the standard median trick and setting the number of copies $t = \Theta(\log(n|Q|))$, we can amplify the success probability to $\frac{1}{n|Q|}$ without increasing the additive error. This completes the proof by the union bound. $\square$

By Lemma C.11 and Lemma C.12, we have $\sum_{H' \in \mathcal{G}_H^{(i)}} \sqrt{f_{H'}(G)} = O(n^{h - \frac{i+1}{2}}) = O(n^{\frac{3-i}{2}} \mathrm{GS}_{f_H})$ since any $H' \in \mathcal{G}_H^{(i)}$ has at most $(2h - i - 1)$ vertices for all $i \in [m_H - 1]$. Thus, Theorem H.7 does not have a stronger dependence on $n$ than Theorem Theorem H.6 for moderate $d$ up to a factor of poly $\log n$. We denote $K_{1,k}$ by a $k$-star and $C_k$ by a $k$-cycle. Table 4 summarizes the additive error of Algorithm 8 for some common pattern graphs.

*Table 4.* The additive error of Algorithm 8 for some common pattern graphs $H$

| Pattern Graph | Additive Error |
| --- | --- |
| Triangle | $O\left((\frac{\tau}{\varepsilon})^3 n^{\frac{3}{2}} + \frac{\tau}{\varepsilon}\sqrt{f_{C_4}(G)}\right)$ |
| 3-Star | $O\left((\frac{\tau}{\varepsilon})^3 n^2 + \frac{\tau}{\varepsilon}\sqrt{n f_{K_{1,4}}(G)} + (\frac{\tau}{\varepsilon})^2 n\sqrt{f_{K_{1,2}}(G)}\right)$ |
| 4-Cycle | $O\left((\frac{\tau}{\varepsilon})^4 n^2 + \frac{\tau}{\varepsilon}\sqrt{f_{C_6}(G)} + (\frac{\tau}{\varepsilon})^2\sqrt{n f_{C_4}(G)}\right)$ |

## H.2. The lower bound for sufficiently large $d$

In this subsection, we extend the methods in Eliáš et al. (2020) to prove the lower bound for $d \geq n/2$. For convenience, we reuse the notations $\mathbf{x}, \mathbf{x}_H, \mathbf{A}_H$ to denote the analogous vectors and matrix defined on $n$-vertex graphs. Their meanings here are independent of those in Section 4.

We consider an $n$-vertex graph $G = (V, E)$ and a query set $Q$. Let $\mathbf{x} \in \{0,1\}^{\binom{n}{2}}$ be an indicator vector of $E$. Let $\mathbf{x}_H = f_H(\mathbf{x}) \in \{0,1\}^{f_H(K_n)}$ be an indicator vector, where each coordinate corresponds to an occurrence $H' = (V', E')$ of $H$ in the complete graph $K_n$ with vertex set $V$, and the entry is 1 if all edges of $H'$ are present in $G$, and 0 otherwise, i.e., $(f_H(\mathbf{x}))_{H'} = \prod_{e \in E'} x_e$. We construct a matrix $\mathbf{A}_H$ with $|Q|$ rows, each corresponding to an induced subgraph $K_n[V_q]$ for a query $q \in Q$, and $f_H(K_n)$ columns, each corresponding to an occurrence $H'$ of $H$ in $K_n$, such that

$$(A_H)_{q,H'} = \begin{cases} 1, & \text{if } E(H') \subseteq E(K_n[V_q]); \\ 0, & \text{otherwise.} \end{cases}$$

Note that $\mathbf{A}_H$ is fixed and does not depend on the edge set of $G$. For each query $q \in Q$, the quantity $(\mathbf{A}_H \mathbf{x}_H)_q$ denotes the number of occurrences of $H$ in the induced subgraph $G[V_q]$.

If the dimension $d$ of the vertex attributes is at least $n/2$, we can set the $i$-th coordinate of the attributes of the $(2i - 1)$-th and $(2i)$-th vertices to $-1$ and 1, respectively, and set all other coordinates to 0 for all $1 \leq i \leq \lceil n/2 \rceil$; then, by choosing the interval for the $i$-th dimension in a query to include $-1$ (resp. 1) or not, we can determine whether the $(2i - 1)$-th (resp. $(2i)$-th) vertex is included in the queried vertex set. Therefore, when $d \geq n/2$, we can always construct the attributes of each vertex so that it is possible to query the induced subgraph of any subset of vertices. Hence there are $2^n$ rows in the matrix $\mathbf{A}_H$. We use the following lemma to show that $\mathbf{A}_H$ satisfies the following discrepancy property.

**Lemma H.9** (Lemma 6 in Bollobás & Scott (2006)). *Let $\{a_i, 1 \leq i \leq n\}$ be a sequence of real numbers. Let $\rho_i \in \{0,1\}$ be i.i.d. Bernoulli$(\frac{1}{2})$, for $1 \leq i \leq n$. Then*

$$\mathbb{E}\left[\left|\sum_{i=1}^n \rho_i a_i\right|\right] \geq \frac{\sum_{i=1}^n |a_i|}{\sqrt{8n}}.$$

**Lemma H.10.** *Let $\gamma \in (0, \frac{1}{2}]$ and $\sigma \in [0, 1]$. Let $C_{\sigma,\gamma}$ be the set of all vectors $\chi = \mathbf{x}_H - \mathbf{x}'_H$, where $\mathbf{x}$ and $\mathbf{x}'$ are the indicator vectors of edges of graphs, denoted by $G = (V, E)$ and $G' = (V, E')$, respectively, such that*

*1. $\|\mathbf{x} - \mathbf{x}'\|_1 \geq \sigma \gamma n^2$;*

*2. for each vertex $v \in V$, its degree lies in the interval $[\frac{\gamma n}{2}, 2\gamma n]$;*

*3. for each edge $e \in E \cup E'$, the number of occurrences of $H$ containing $e$ lies in the interval $[\frac{\mathrm{LS}_{f_H}(G)}{2}, 2\mathrm{LS}_{f_H}(G)]$.*

*Then for the matrix $\mathbf{A}_H$ defined above, we have $\mathrm{disc}_{C_{\sigma,\gamma}}(\mathbf{A}_H) = \Omega\left(\sigma n\sqrt{\gamma} \cdot \mathrm{LS}_{f_H}(G)\right)$.*

*Proof.* In the complete graph $K_n$ with vertex set $V$, we denote by $\mathcal{H}_{A,B}$ the set of all occurrences $H'$ of $H$ whose vertex set satisfies $A \subseteq V(H') \subseteq B$.

Let $V' \subseteq V$ be a random subset such that for each $v \in V$, the indicator variable $\rho_v = \mathbb{1}_{v \in V'}$ is i.i.d. Bernoulli$(\frac{1}{2})$. For any given $v \in V$, denote $N(v)$ the set of vertices adjacent to $v$ by edges whose color is non-zero, i.e., $N(v) = \{w \mid (v, w) \in$

$E \triangle E'$}. Not that $|N(v)| \le 4\gamma n$. Let $d_{H'}(v)$ denote the degree of $v$ in an occurrence $H'$, and we have

$$
\begin{aligned}
\text{disc}_{C_{\sigma,\gamma}}(\mathbf{A}_H) = \min_{\chi \in C_{\sigma,\gamma}} \|\mathbf{A}_H \chi\|_\infty &\ge \min_{\chi \in C_{\sigma,\gamma}} \mathbb{E}\left[\left|\sum_{H' \in \mathcal{H}_{\emptyset,V'}} \chi_{H'}\right|\right] \\
&= \min_{\chi \in C_{\sigma,\gamma}} \frac{1}{h}\mathbb{E}\left[\left|\sum_{v \in V} \rho_v \sum_{H' \in \mathcal{H}_{\{v\},V'}} \chi_{H'}\right|\right] \\
&\ge \min_{\chi \in C_{\sigma,\gamma}} \frac{1}{h\sqrt{8n}} \sum_{v \in V} \mathbb{E}\left[\left|\sum_{H' \in \mathcal{H}_{\{v\},V'}} \chi_{H'}\right|\right] \\
&= \min_{\chi \in C_{\sigma,\gamma}} \frac{1}{h\sqrt{8n}} \sum_{v \in V} \mathbb{E}\left[\left|\sum_{w \in N(v) \cap V} \rho_w \sum_{H' \in \mathcal{H}_{\{v,w\},V'}} \frac{\chi_{H'}}{d_{H'}(v)}\right|\right] \\
&\ge \min_{\chi \in C_{\sigma,\gamma}} \frac{1}{16hn\sqrt{\gamma}} \sum_{(v,w) \in E\triangle E'} \mathbb{E}\left[\left|\sum_{H' \in \mathcal{H}_{\{v,w\},V'}} \frac{\chi_{H'}}{d_{H'}(v)}\right|\right],
\end{aligned}
\tag{15}
$$

where the factor $1/h$ (resp. $1/d_{H'}(v)$) compensates for the fact that each $\chi_{H'}$ is counted $h$ (resp. $d_{H'}(v)$) times in the sum. The last and penultimate inequalities follow from Lemma H.9.

Now, by our assumption on the properties of the graphs $G$ and $G'$, for each edge $(u,v) \in E\triangle E'$, it holds that the occurrences $H'$ containing $(u,v)$ are either in $G$ or in $G'$, but not both; and each edge belongs to at least $\frac{\text{LS}_{f_H}(G)}{2}$ occurrences of $H$. Therefore, we obtain

$$
\begin{aligned}
\mathbb{E}\left[\left|\sum_{H' \in \mathcal{H}_{\{v,w\},V'}} \frac{\chi_{H'}}{d_{H'}(v)}\right|\right] &= \sum_{H' \in \mathcal{H}_{\{v,w\},V'}} \mathbb{E}\left[\left|\frac{\chi_{H'}}{d_{H'}(v)}\right|\right] \\
&\ge \frac{1}{h} \sum_{H' \in \mathcal{H}_{\{v,w\},V}} \Pr[H' \in V'] \\
&= \frac{|\mathcal{H}_{\{v,w\},V}|}{h 2^{h-2}} \ge \frac{\text{LS}_{f_H}(G)}{h 2^{h-1}},
\end{aligned}
\tag{16}
$$

where the penultimate inequality follows from the fact $d_{H'}(v) \le h$. Substituting Equation (16) into Equation (15), we have

$$
\text{disc}_{C_{\sigma,\gamma}}(\mathbf{A}_H) \ge \frac{\text{LS}_{f_H}(G) \cdot |E\triangle E'|}{16h^2 2^{h-1} n\sqrt{\gamma}} \ge \frac{\text{LS}_{f_H}(G) \cdot \sigma\gamma n^2}{16h^2 2^{h-1} n\sqrt{\gamma}} = \Omega\left(\sigma n\sqrt{\gamma} \cdot \text{LS}_{f_H}(G)\right).
$$

$\square$

Let $G(n,p)$ denote the distribution of Erdős–Rényi random graphs. For any graph $G = (V,E) \sim G(n,p)$, we use the following lemma to bound the number of occurrences of $H$ in $G$ containing each edge $e \in E$.

**Lemma H.11.** *Let $G = (V,E)$ be a graph generated from $G(n,p)$. If $p \gg ((\ln n)/n)^{1/(2m_H-2)}$ or $H$ is an edge (i.e., $m_H = 1$), with probability at least $1 - \text{poly}(1/n)$, the following properties $(\star)$ hold:*

1. *for each vertex $v \in V$, its degree belongs to the interval $[\frac{\gamma n}{2}, 2\gamma n]$;*
2. *for each $e \in E$, the number of occurrences of $H$ containing $e$ belongs to the interval $[\frac{\text{LS}_{f_H}(G)}{2}, 2\text{LS}_{f_H}(G)]$ and $\text{LS}_{f_H}(G) = \Theta(\text{GS}_{f_H} \cdot p^{m_H-1})$.*

*Proof.* Item 1 in the properties $(\star)$ follows from the Chernoff bound. We now focus on Item 2. Note that when $h = 2$ and $m_H = 1$ (i.e., $H$ is an edge), it holds automatically since $\text{LS}_{f_H}(G) = 1$. Therefore, we assume $h \ge 3$ and $m_H \ge 2$ in the following. Fix an edge $e \in E$. Let $X_e$ denote the number of occurrences of $H$ containing $e$ in $G$. Fix an arbitrary

ordering of the edges in $\{(u,v) \mid u,v \in V\} \setminus \{e\}$, denoted by $e_1, e_2, \ldots, e_{\binom{n}{2}-1}$, and define the indicator random variable $Y_i = \mathbb{1}_{e_i \in E}$ for each $i \in [\binom{n}{2}] - 1]$. Let $Z_0 = \mathbb{E}[X_e] = \mathrm{GS}_{f_H} \cdot p^{m_H - 1}$. For each $i \in [\binom{n}{2}] - 1]$, let $Z_i = \mathbb{E}[X_e | Y_1, \ldots, Y_i]$. Then $Z_{\binom{n}{2}-1} = X_e$ and $Z_0, \ldots, Z_{\binom{n}{2}-1}$ is a martingale by the law of total expectation.

By an analysis similar to the proof of Lemma C.11, for each edge $e_i$, if it shares an endpoint with $e$, then $|Z_i - Z_{i-1}| \leq \Theta(n^{h-3})$; otherwise, $|Z_i - Z_{i-1}| \leq \Theta(n^{h-4})$. By Lemma C.11 and Lemma H.5, if $p \gg ((\ln n)/n)^{1/(2m_H - 2)}$, there exists a constant $C > 0$ such that

$$\exp\left(-\frac{(\varepsilon \cdot \mathrm{GS}_{f_H} \cdot p^{m_H - 1})^2}{2\left(\sum_{i=1}^{\binom{n}{2}-1} c_i^2\right)}\right) \leq \exp\left(-\frac{C \cdot n^{2h-4} \cdot p^{2m_H - 2}}{2n \cdot n^{2h-6} + n^2 \cdot n^{2h-8}}\right) \leq \mathrm{poly}(1/n).$$

Then with probability at least $1 - \mathrm{poly}(1/n)$, $X_e$ belongs to the interval $[(1-\varepsilon)\mathrm{GS}_{f_H} \cdot p^{m_H - 1}, (1+\varepsilon)\mathrm{GS}_{f_H} \cdot p^{m_H - 1}]$ for some $\varepsilon > 0$.

Note that $\mathrm{LS}_{f_H}(G) = \max_{e \in E} X_e$. By the union bound and choosing a suitable constant $\varepsilon > 0$, it holds that with probability at least $1 - \mathrm{poly}(1/n)$, all the random variables $X_e$ lie in the interval $[\frac{\mathrm{LS}_{f_H}(G)}{2}, 2\mathrm{LS}_{f_H}(G)]$ and $\mathrm{LS}_{f_H}(G) = \Theta\left(\mathrm{GS}_{f_H} \cdot p^{m_H - 1}\right)$. By the union bound, with probability at least $1 - \mathrm{poly}(1/n)$, the properties $(\star)$ are satisfied. $\qquad \square$

The remaining proof largely follows directly from Eliáš et al. (2020), and we only need to make slight adjustments to adapt their proofs for our case. We sketch the proofs here for the sake of completeness. By an analysis similar to the proof of Lemma F.2, we obtain the following lemma.

**Lemma H.12.** *Let $\mathbf{x}$ be the indicator vector of the edge set of some graph $G$ satisfying the properties $(\star)$ in Lemma H.11. There is a deterministic algorithm $\mathcal{A}$ given an output $\mathbf{y} = \mathcal{M}(\mathbf{x})$ of some mechanism $\mathcal{M}$ such that $\|\mathbf{y} - \mathbf{A}_H \mathbf{x}_H\|_\infty < \frac{1}{2}\mathrm{disc}_{C_{\sigma,\gamma}}(\mathbf{A}_H)$ satisfies $\|\mathcal{A}(\mathbf{y}) - \mathbf{x}\|_1 \leq \sigma \gamma n^2$.*

The following lemmas on differential privacy will be useful in the subsequent proof.

**Lemma H.13** (Lemma 5.4 in Eliáš et al. (2020)). *Let $\mathcal{M}$ be an $(\varepsilon, \delta)$-differentially private mechanism and let $Y$ be the probability distribution over the transcripts of $\mathcal{M}(\mathbf{x})$, where $\mathbf{x}$ is drawn from distribution $X$. Then for any $\gamma > 0$ and $\mathbf{y} \sim Y$, it holds that with probability $1 - \delta'$ over $i \in [n]$ and $\mathbf{x} \leftarrow X_{|Y=\mathbf{y}}$, we have*

$$2^{-\varepsilon - \gamma}\frac{1-p}{p} \leq \frac{\Pr_{\mathbf{x} \leftarrow X_{|Y=\mathbf{y}}}[\mathbf{x}_i = 0 \mid \mathbf{x}_{-i}]}{\Pr_{\mathbf{x} \leftarrow X_{|Y=\mathbf{y}}}[\mathbf{x}_i = 1 \mid \mathbf{x}_{-i}]} \leq 2^{\varepsilon + \gamma}\frac{1-p}{p},$$

*where $x_{-i}$ denotes the vector of all coordinates of $x$ excluding $x_i$ and $\delta' = 2\delta \cdot \frac{1+e^{-\varepsilon - \gamma}}{1 - e^{-\gamma}}$.*

**Lemma H.14** (Lemma 2.1.2 in Bun (2016)). *Let $\mathcal{M} : \mathcal{X} \to \mathcal{R}$ be an $(\varepsilon, \delta)$-differentially private mechanism, $c \in \mathbb{N}$, and $\mathbf{x}, \mathbf{x}' \in \mathcal{X}$ such that $\|\mathbf{x} - \mathbf{x}'\|_1 \leq c$. Then, for every $S \subseteq \mathcal{R}$, we have*

$$\Pr[\mathcal{M}(\mathbf{x}) \in S] \leq e^{c\varepsilon} \Pr[\mathcal{M}(\mathbf{x}') \in S] + \frac{e^{c\varepsilon} - 1}{e^\varepsilon - 1}\delta.$$

Now we prove the lower bound for $\varepsilon = 1$.

**Lemma H.15.** *Let $G \sim G(n,p)$, where $p \in (0, \frac{1}{2}]$, be a random graph. And we assume $p \gg ((\ln n)/n)^{1/(2m_H - 2)}$ unless $H$ is an edge. Let $\mathcal{M}$ be a $(1, \delta)$-DP mechanism that answers all range subgraph counting queries up to additive error $\alpha$ with probability $\beta$. Then $\alpha \geq \Omega(\mathrm{disc}_{C_{\sigma,\gamma}}(\mathbf{A}_H))$, where $\gamma = p, \sigma = \Omega(1 - \frac{9\delta}{\beta})$.*

*Proof Sketch.* We choose $\varepsilon = 1$ and $\varepsilon' = \varepsilon + 10$. By Lemma H.13, this implies $\delta' = 2\delta \cdot \frac{1+e^{-\varepsilon - 10}}{1-e^{-10}} \leq 3\delta$, then with probability $1 - \delta'$ over $i \in [n]$ and $\mathbf{x} \leftarrow X_{|Y=\mathbf{y}}$, we have

$$2^{-\varepsilon'}\frac{1-p}{p} \leq \frac{\Pr_{\mathbf{x} \leftarrow X_{|Y=\mathbf{y}}}[\mathbf{x}_i = 0 \mid \mathbf{x}_{-i}]}{\Pr_{\mathbf{x} \leftarrow X_{|Y=\mathbf{y}}}[\mathbf{x}_i = 1 \mid \mathbf{x}_{-i}]} \leq 2^{\varepsilon'}\frac{1-p}{p}. \tag{17}$$

Then we can prove the lemma by contradiction. That is, we assume that $\mathcal{M}$ has an additive error smaller than $\frac{1}{2}\mathrm{disc}_{C_{\sigma,\gamma}}(\mathbf{A}_H)$ with probability at least $\beta$. Then we can show that for each possible output $\mathbf{y}$ of the mechanism $\mathcal{M}$, with probability greater than $\delta'$, Appendix H.2 is violated. To do so, we only need to show that 1) with high probability, $\mathbf{x}$ is *good* in the sense that it satisfies the desired properties; and 2) conditioned on the event that $\mathbf{x}$ is good, the inequality (17) is violated with probability greater than $\delta'$, if we set $\gamma = p$ and $\sigma = 2^{-\varepsilon'-3} \cdot (1 - \frac{3\delta'}{\beta})$, which leads to a contradiction.

Part 2) follows from the same argument as those in the proof of Lemma 5.3 in (Eliáš et al., 2020). For part 1), we describe our changes. Formally, we define that $\mathbf{x} \sim X_{|Y=\mathbf{y}}$ is *good* if $\|\mathbf{A}_H \mathbf{x}_H - \mathbf{y}\|_\infty < \frac{1}{2}\mathrm{disc}_{C_{\sigma,\gamma}}(\mathbf{A}_H)$ and the properties $(\star)$ given in the statement of Lemma H.11 are satisfied. Note that the probability that $\mathbf{x} \sim X_{|Y=\mathbf{y}}$ is good is at least $(1 - \mathrm{poly}(1/n)) \cdot \beta$ by Lemma H.11. This then finishes the proof of the lemma. $\qquad\square$

Finally, we finish the proof of the lower bound for $d \geq n/2$.

**Theorem H.16.** *Let $\mathcal{M}$ be an $(\varepsilon, \delta)$-DP mechanism and $G \sim G(n, p)$, where $p \in (0, \frac{1}{2}]$. And we assume $p \gg ((\ln n)/n)^{1/(2m_H-2)}$ unless $H$ is an edge. If $\mathcal{M}$ answers all $d$-dimensional $(d \geq n/2)$ range subgraph counting queries about $G$ up to an additive error $\alpha$ with probability at least $\beta$, then $\alpha = \Omega(\sqrt{m}(1-c) \cdot \mathrm{LS}_{f_H}(G))$, where $c = \frac{e-1}{e^\varepsilon-1}\frac{9\delta}{\beta}$.*

*Proof.* Note that $m = \Theta(\gamma n^2)$. Then $\Theta(n\sqrt{\gamma}) = \Theta(\sqrt{m})$. Lemma H.15 together with Lemma H.10 imply that there is no $(1, \delta)$-DP mechanism $\mathcal{M}$ whose error with probability at least $\beta$ is below $o(\sqrt{m}(1 - \frac{9\delta}{\beta}) \cdot \mathrm{LS}_{f_H}(G))$.

Let us assume for contradiction, that there is an $(\varepsilon, \delta)$-DP mechanism $\mathcal{M}(\mathbf{x})$ whose error is smaller than $o(\sqrt{m}(1-c) \cdot \mathrm{LS}_{f_H}(G))$ with probability $\beta$, where $c = \frac{e-1}{e^\varepsilon-1} \cdot \frac{9\delta}{\beta}$. Note that $f_H(\frac{\mathbf{x}}{\varepsilon}) = \varepsilon^{-m_H} f_H(\mathbf{x})$. Let us consider a mechanism $\varepsilon^{m_H} \mathcal{M}(\frac{\mathbf{x}}{\varepsilon})$. By Lemma H.14, it is $(1, \frac{e-1}{e^\varepsilon-1}\delta)$-DP. Moreover, it has error

$$\varepsilon^{m_H} \cdot o\left(\frac{\sqrt{m}}{\varepsilon^{m_H}}(1-c) \cdot \mathrm{LS}_{f_H}(G)\right) \leq o\left(\sqrt{m}(1-c) \cdot \mathrm{LS}_{f_H}(G)\right)$$

with probability at least $\beta$ — a contradiction. $\qquad\square$

When the parameters $p, \varepsilon, \delta$ in Theorem H.16 are constants, we show that any $(\varepsilon, \delta)$-DP algorithm incurs an additive error of $\Omega(n\mathrm{GS}_{f_H})$ by Lemma H.11.

