# OpenReview forum: "Differentially Private Range Subgraph Counting"
_ICML.cc/2026/Conference — ICML 2026 regular_

### Official Review · Reviewer_T7Lu · 2026-02-23

**Soundness:** 3
**Presentation:** 3
**Significance:** 3
**Originality:** 3
**Overall Recommendation:** 5
**Confidence:** 4

**Summary:**

The paper studied the differentially private algorithms for subgraph counting on vertices in range queries. The paper focused on the version of subgraph counting with structures of $O(1)$ vertices, and the “range query” aspect is defined by $d$-dimensional attribute vectors: here, a query $q$ encodes the upper and lower bounds of the attributes of each dimension, and the subgraph counting is only over the “eligible” vertices in the range. The query set $Q$ contains multiple queries, and the error is defined as the maximum error over the queries $q\in Q$. The paper asks about the *differentially private* release of the queries when the query set $Q$ is known in advance.

The paper derived upper and lower bounds for the problem, including:
- *pure DP Algorithm*: an algorithm that answers a set of $Q$ of range queries with additive error $O(\frac{GS_{F_H}\cdot \sqrt{\log(n |Q|)} \log^{O(d)}(n)}{\varepsilon})$.
- *approximate DP Algorithm*: an algorithm that answers a set of $Q$ of range queries with additive error $O(\frac{HS_{F_H}\cdot \sqrt{\log(1/\delta)\log(n |Q|)} \log^{O(d)}(n)}{\varepsilon})$ (ignoring lower-order terms).
These algorithms run in polynomial pre-processing and query time.
The paper further proved several lower bounds; one of them indicates that when $d=O(\log{n})$, an additive error of $2^{\Omega(d)}$ is inevitable.


*Main techniques.** The algorithms of the paper utilized the range query tree to reduce the dependency on $|Q|$. To avoid composition error (which will be $\sqrt{|Q|}$), the paper utilized a product of binary tree structures such that the search depth will be at most $O(\log(|Q|))$, which intuitively reduced the dependency on the size of the query set.
To recover the number of isomorphic subgraphs using the range tree, the paper defined a “projection” that maps the number of occurrences to a point in the nested range trees. The algorithm then adds noise to points to ensure DP. The lower bound is based on some known discrepancy method and a novel reduction to solve the reconstruction attack with subgraph counting queries.

**Compliance With Llm Reviewing Policy:**

Affirmed.

**Final Justification:**

========== Post-rebuttal ============
The authors addressed my concerns in the rebuttal, especially for questions related to the technical and practical significance. I now see that the paper provided a way to compute the sensitivity measure. Furthermore, the practical scaling concern for the subgraph numeration/counting problem can be resolved by some existing algorithm engineering results. In light of this, I improved my rating of significance to 3 and increase the total score to 5.

**Key Questions For Authors:**

My main questions are encoded in the weakness comments for the paper. Two MISC comments:
- Line 81, right column: $d_{max}()$ (missing left “(”)
- The legends on your figures are too small; I have to zoom in to read them.

**Limitations:**

yes (N/A, no foreseeable negative societal impact)

**Strengths And Weaknesses:**

**Strength**
- I believe the problem is well-motivated: modern graphs can and often carry edge attributes as weights, and querying some specific patterns within a given range is well-motivated.
- The technical depth is good, and the paper provides enough explanations of the results. Although I do not know some of the techniques (e.g., it seems from the description that the projection between weights in the range trees and subgraph counting is standard, although I don’t know about it), I think the main ideas in the algorithms are pretty intuitive and easy to follow.
- The paper provided some experiments, showing that the algorithm has low additive errors and fast running time (although the pre-processing time is high, which is as expected).

**Weakness**
- The sensitivity measures in the bounds: although I understand why the bounds are written in the forms of $GS_{f_H}$ and $HS_{f_H}$, these bounds still appear to be quite not natural and involved to me. Also, the paper provided very limited insights on the *values of these sensitivity measures* – what should be $GS_{f_H}$ in a graph with $H$ as, e.g., a 4-clique?
- The role of $|Q|$: in the upper bound, I think the paper spent some effort trying to beat the native $\sqrt{|Q|}$ bound induced by composition; however, the lower bound is entirely independent of such a value. Therefore, although the lower bound techniques are quite nice, I found them less informative.
- Given the fact that the algorithm needs to enumerate all possible H in the graph, the “polynomial time” is not likely to scale with $V(H)\geq 30$ vertices unless our graph is really really small. I’m OK with the main contribution of the paper to be primarily theoretical, but I want to flag this.

Overall, my opinion of the paper is mostly positive, although I decided not give higher scores due to the weaknesses.

---

> ### Author Rebuttal · Authors · 2026-03-31
>
> We thank the reviewer for the supportive feedback and positive evaluation of our work. We will address your concerns as follows.
>
> >W1: The sensitivity measures in the bounds: although I understand why the bounds are written in the forms of $GS_{f_H}$ and $HS_{f_H}$, these bounds still appear to be quite not natural and involved to me. Also, the paper provided very limited insights on the values of these sensitivity measures – what should be $GS_{f_H}$ in a graph with $H$ as, e.g., a 4-clique?
>
> We note that Appendix C.2 already provides a concrete method for computing $GS_{f_H}$ (see Lemma C.11) for any pattern graph $H$, along with a table of values for several common patterns (see Table 3), including $k$-cliques. For example, when $H$ is a 4-clique, we have $GS_{f_H} = \binom{n - 2}{2}$.
>
> In addition, Appendix C.3 discusses the challenges of computing $HS_{f_H} $ and provides explicit upper bounds for $HS_{f_H}$ in the cases where $H$ is a 2-star or a triangle. These results illustrate both how these sensitivity measures can be evaluated and why their exact computation can be nontrivial for general patterns.
>
> Thank you for this valuable feedback. In the final version of the paper, we will revise the main text to better highlight these results and improve their accessibility, so that the meaning and magnitude of $GS_{f_H}$ and $HS_{f_H}$ are clearer to readers.
>
> >W2: The role of $|Q|$: in the upper bound, I think the paper spent some effort trying to beat the native bound $\sqrt{|Q|}$ induced by composition; however, the lower bound is entirely independent of such a value. Therefore, although the lower bound techniques are quite nice, I found them less informative.
>
> In fact, we do provide a lower bound that depends on $|Q|$ in Theorem 4.1. For example, when $d = O(1)$, the lower bound takes the form
> $\Omega\big(\log^{d-1} \min(n, |Q|) \cdot GS_{f_H}\big)$,
> which explicitly captures the dependence on the query set size $|Q|$.
>
> In the statement of the main results (Theorem 1.4), the assumption $|Q| \ge \mathrm{poly}(n)$ and the reformulation of $GS_{f_H}$ in terms of $HS_{f_H}$ are introduced primarily for clarity and cleaner presentation. These two aspects are independent and are not required in the underlying proofs; the analysis applies more generally without these assumptions.
>
> Thank you for this valuable feedback. We will revise the presentation of the main results in the final version to better clarify the dependence on $|Q|$ and make the role of these parameters more explicit.
>
> >W3: Given the fact that the algorithm needs to enumerate all possible $H$ in the graph, the “polynomial time” is not likely to scale with $|V(H)|\ge 30$ vertices unless our graph is really really small. I’m OK with the main contribution of the paper to be primarily theoretical, but I want to flag this.
>
> We agree that the time complexity of our algorithms depends heavily on $|V(H)|$, since subgraph enumeration is inherently exponential in the size of the pattern graph.
>
> At the same time, the subgraph enumeration step is a well-studied problem, and in practice there exist optimized algorithms and heuristics that can handle moderately sized patterns efficiently on sparse graphs. Moreover, the enumeration of all occurrences of $H$ is only one component of our overall procedure; the additional overhead of computing approximate local sensitivity in our framework is relatively substantial compared to the cost of enumeration itself.
>
> Therefore, while the overall runtime indeed scales rapidly with $|V(H)|$ and may become impractical for very large patterns (e.g., $|V(H)| \ge 30$) on large graphs, this limitation reflects the inherent difficulty of the subgraph counting problem rather than being specific to our approach. We will clarify this point in the revision and explicitly discuss the practical scalability with respect to $|V(H)|$.
>
> >Q1: My main questions are encoded in the weakness comments for the paper. Two MISC comments:
> Line 81, right column: $d_{\text{max}}()$ (missing left “(”)
> The legends on your figures are too small; I have to zoom in to read them.
>
> Thank you for your helpful suggestions. We will correct the typographical error on Line 81 (right column) by fixing the missing parenthesis in $d_{\text{max}}(\cdot)$, and we will also increase the font size of the figure legends to improve readability in the final version of the paper.

---

> > ### Author Rebuttal · Reviewer_T7Lu · 2026-04-04
> >
> > The authors addressed my concerns in the rebuttal, especially for questions related to the technical and practical significance. I now see that the paper provided a way to compute the sensitivity measure. Furthermore, the practical scaling concern for the subgraph numeration/counting problem can be resolved by some existing algorithm engineering results. In light of this, I improved my rating of significance to 3 and increased the total score to 5. Good luck :)

---

### Official Review · Reviewer_uTBn · 2026-03-07

**Soundness:** 3
**Presentation:** 3
**Significance:** 2
**Originality:** 2
**Overall Recommendation:** 4
**Confidence:** 4

**Summary:**

The paper introduces the problem of range subgraph counting (RSC) for graphs. Compared to subgraph counting on the entire subgraph, here the subgraph queries are made on an induced subgraph that is defined through range queries on $d$-dimensional attribute vectors associated with the vertices. They propose DP algorithms for both pure and approximate differential privacy. The error for the pure DP algorithm depends on the global sensitivity, while the error for the approximate DP algorithm depends on a hybrid sensitivity term (denoted HS). They also establish lower bounds for approximate DP algorithms across various ranges of the dimension of the attribute vectors $d$. These results show that the error must contain a term exponential in the dimension $d$ for certain range of values $d$. They also perform experiments on two real-world datasets.

**Compliance With Llm Reviewing Policy:**

Affirmed.

**Ethical Review Concerns:**

None.

**Final Justification:**

I think the problem is interesting and timely. However, the upper and lower bounds are relatively straightforward. Even with the examples provided by the authors, the issue of attributes remaining non-private makes the DP motivation weak. Overall, I maintain my original scores.

**Key Questions For Authors:**

1) Can your algorithm be modified to ensure the privacy of individual attribute vectors?
2) Can you discuss the limitations and future directions regarding the RSC problem?
3) Can you describe some concrete examples to motivate the RSC problem?

**Limitations:**

As noted earlier, the article would benefit from an explicit discussion on limitations and future directions.

**Strengths And Weaknesses:**

Soundness: Yes. The claims in the submission are well supported by theoretical analysis.


Presentation:  It would be better to introduce the $HS$ term in more detail in the main paper, preferably before stating the main results that contain this term.

Significance: The problem of subgraph counting itself is an important problem, and the extension to RSC seems natural. However, the formulation would benefit from more well-substantiated real-world examples of scenarios where the problem arises.
The DP setting in particular is somewhat harder to motivate given the attribute vectors themselves are not kept private. Protecting the attribute vectors would be of at least the same, if not more importance as protecting relationship information for many scenarios, e.g. financial networks.

Originality: The originality lies more in the problem formulation than the techniques used in the work. Both the upper and lower bound follows from relatively straightforward reduction to known results. The article would benefit from a discussion on the limitations, and possible future directions regarding the RSC problem.

---

> ### Author Rebuttal · Authors · 2026-03-31
>
> We thank the reviewer for the supportive feedback and positive evaluation of our work. We will address your concerns as follows.
>
> >W1: It would be better to introduce the term $\text{HS}$ in more detail in the main paper, preferably before stating the main results that contain this term.
>
> Thank you for this helpful suggestion. In the final version of the paper, we will provide a more detailed introduction and discussion of the term $\mathrm{HS}$ prior to stating the main results that involve it, to improve clarity and readability.
>
> >W2/Q1: The DP setting in particular is somewhat harder to motivate given the attribute vectors themselves are not kept private. Protecting the attribute vectors would be of at least the same, if not more importance as protecting relationship information for many scenarios, e.g. financial networks. Can your algorithm be modified to ensure the privacy of individual attribute vectors?
>
> We acknowledge the importance of protecting the privacy of individual attribute vectors and agree that there are important settings in which the graph structure is public while the attributes are sensitive (e.g., in financial or demographic applications). Our current work focuses on protecting edge-level information while treating attributes as public inputs. Extending the framework to also provide differential privacy guarantees for attribute vectors is an interesting and meaningful direction. It seems plausible that one could still leverage the graph structure together with range tree–based techniques to obtain some level of privacy protection in this setting.
>
> However, achieving strong (and especially near-optimal) utility guarantees under attribute-level privacy would likely require new ideas beyond our current approach, as the sensitivity structure and noise calibration would fundamentally change. We therefore leave this as an important direction for future work and will clarify this limitation and discussion in the revision.
>
> >W3/Q2: Can you discuss the limitations and future directions regarding the RSC problem?
>
> The limitations of our work include that our current algorithm does not address the setting in which the attribute information itself is sensitive and must be protected. In addition, there remains a gap between our upper and lower bounds for certain regimes of the parameters, leaving room for potential improvement in the analysis or algorithm design.
>
> As for future directions, it would be interesting to extend the RSC problem to more challenging settings, such as dynamic graphs where edges or attributes evolve over time, as well as alternative privacy models including local differential privacy or node-level differential privacy. These directions may require new techniques to handle the additional structural and privacy constraints.
>
> We thank the reviewer for this helpful suggestion, and we will include a dedicated discussion of limitations and future research directions in the final version of the paper.
>
> >Q3: Can you describe some concrete examples to motivate the RSC problem?
>
> We thank the reviewer for this suggestion. We provide two concrete motivating examples for the RSC problem.
>
> (1) Social networks. Consider a social network (e.g., Facebook) where each user (vertex) is associated with publicly available attributes such as gender and geographic location. An advertiser or recommendation system may wish to query the structural connectivity of a specific demographic group defined via these attributes (i.e., range subgraph counting over selected attribute ranges), while the underlying friendship relationships (edges) are considered sensitive and should be protected. In this setting, the goal is to release aggregate structural statistics about attribute-defined subpopulations without revealing individual relationships.
>
> (2) Financial or transaction networks. Consider a transaction network where vertices represent accounts and edges represent transactions. Each account is associated with attributes such as transaction volume, geographic region, or risk score. Analysts may issue queries restricted to accounts whose attributes fall within certain ranges (e.g., high-volume accounts in a given region) to detect structural patterns such as anomalous subgraphs indicative of fraud or money laundering. At the same time, it is important to preserve the privacy of individual transactions (edges), motivating the need for differentially private range subgraph counting.
>
> These examples illustrate how RSC naturally arises in applications where one needs to combine attribute-based filtering with privacy-preserving analysis of graph structure.

---

> > ### Author Rebuttal · Reviewer_uTBn · 2026-04-04
> >
> > Thanks to the authors for their detailed response. It clarifies all my concerns. However, it does not change my overall assessment of the paper. Hence, I maintain my score.

---

### Official Review · Reviewer_n6j4 · 2026-03-11

**Soundness:** 3
**Presentation:** 3
**Significance:** 2
**Originality:** 2
**Overall Recommendation:** 4
**Confidence:** 4

**Summary:**

## Problems
This paper introduces the problem of differentially private range subgraph counting (DPRSC).
In the range subgraph counting problem, each vertex of the graph $G$ is associated with an attribute vector $a(v) \in R^d$.
Fix a connected pattern graph $H$ with $O(1)$ vertices.
A query range $q = [\ell_1, r_1] \times \cdots \times [\ell_d, r_d]$ defines a vertex subset $V_q = \{v \in V : \ell_i \le a_i(v) \le r_i, \forall i \in [d]\}$.
Given a set $Q$ of query ranges, the goal is to return, for each query $q \in Q$, the number of occurrences of $H$ in $G_q := G[V_q]$, denoted by $f_H(G_q)$.
Regarding DP, this paper considers edge DP, i.e., two graphs are neighboring if they differ by an edge. (Vertex attributes are assumed to be public.)

## Results
This paper gives
- an eps-DP algorithm that outputs $\tilde{f}_H(G_q)$ for each query $q \in Q$ and satisfies $\max_{q \in Q} |f_H(G_q)| \lesssim GS \sqrt{\log n |Q|} \log^{3d}(n) / \epsilon$, where GS denotes the global sensitivity of $f_H$
- an (eps, delta)-DP algorithm whose utility guarantee replaces GS by some approximation of the local sensitivity of $f_H$ by (Nguyen et al., 2023)
- a lower bound of $\Omega(\log^{d-1}(n) GS)$ for $d=O(1)$ as well as lower bounds for $d = O(\log n)$ and $d = \Omega(\log n)$.

## Algorithmic Techniques
They first reduce RSC to orthogonal range counting. Specifically, each occurrence of H is mapped to a weighted point in $R^{2d}$ in such a way that the answer to a query q is equal to the sum of weights of points in a hypercube induced by $q$.
Then they use existing DP algorithms for orthogonal range counting by (Dwork et al. 2015).
They also show that one can use the local sensitivity estimation technique of (Nguyen et al. 2023) to replace the global sensitivity by some approximate local sensitivity, at the cost of moving from pure DP to approximate DP.

## Lower Bounds
The lower bounds are proved via a reduction from the reconstruction attacks (RA) problem, the well-known connection between discrepancy and DP lower bounds, and some new discrepancy bound (Lemma 4.3, main technical contribution).

**Compliance With Llm Reviewing Policy:**

Affirmed.

**Final Justification:**

The rebuttal addressed my main concerns.
The DPRSC problem, and both the algorithm and lower bound look interesting.
Overall, I think this paper is on the borderline, and I slightly lean towards acceptance.

**Key Questions For Authors:**

- When $a$ maps all vertices to the same point, we are back to the standard subgraph counting problem. In this case, how does your result compare with previous results?

**Strengths And Weaknesses:**

- The DPRSC problem looks interesting and well-motivated to me.
- The main technical novelties of this paper are the conversion from DPRSC to DP orthogonal range counting and the lower bound.
- The paper is well-written in general.

- Maybe it's due to my unfamiliarity with the literature, it is hard for me access the significance/meaning of the utility guarantees of the algorithms. It would be nice to add more discussions on this, e.g. how does the additive errors compare with the true answers, maybe through more concrete examples.

---

> ### Author Rebuttal · Authors · 2026-03-31
>
> We thank the reviewer for the supportive feedback and positive evaluation of our work. We will address your questions as follows:
>
> >W1: Maybe it's due to my unfamiliarity with the literature, it is hard for me access the significance/meaning of the utility guarantees of the algorithms. It would be nice to add more discussions on this, e.g. how does the additive errors compare with the true answers, maybe through more concrete examples.
>
> We thank the reviewer for this helpful suggestion. To better illustrate the significance of our utility guarantees, we provide concrete examples comparing the additive error to the magnitude of the true answers.
>
> For clarity, consider a simple setting where $d$, $\varepsilon$, and $\delta$ are treated as constants, the number of queries $|Q|$ is polynomial in $n$, each query $q \in Q$ satisfies $|V_q| = \Theta(n)$, and the pattern graph $H$ is a triangle.
>
> (1) Dense graph (complete graph): When $G$ is a complete graph, the true answer to any query in $Q$ is $\Theta(n^3)$, while the additive error of our algorithms is $\widetilde{O}(n)$. Thus, the relative error is $\widetilde{O}(1/n^2)$, which is negligible for large $n$.
>
> (2) Sparse graph (bounded-degree graph): When $G$ has maximum degree $d_{\max} \ll n$, the true answer to any query in $Q$ is $O(d_{\max}^2 n)$. In this case, the additive error of our pure DP algorithm is $\widetilde{O}(n)$, while that of our approximate DP algorithm is $O(d_{\max} \cdot \mathrm{polylog}(n))$. Hence, the approximate DP algorithm achieves substantially better accuracy in sparse graphs.
>
> These examples demonstrate that, in typical regimes of interest, the additive errors of our algorithms are significantly smaller than the true answers.
>
> We will incorporate additional discussion and examples in the final version to make the utility guarantees more accessible and intuitive.
>
> >Q1: When $\textbf{a}$ maps all vertices to the same point, we are back to the standard subgraph counting problem. In this case, how does your result compare with previous results?
>
> In this special case, where all vertices are mapped to the same point, the problem reduces to the standard subgraph counting problem. Under our current analysis, the bounds incur an additional $\log^{O(d)} n$ factor compared to prior results. This gap arises because our bounds are expressed in terms of $n$, rather than the effective domain size of the attributes.
>
> However, the analysis can be refined to depend on the domain size of each attribute. In particular, the factor $\log^{O(d)} n$ can be replaced by
> $\left(\prod_{i=1}^{d} \big(\lceil \log |A_i(V)| \rceil + 1\big)\right)^{O(1)}$,
> where $A_i(V)$ denotes the set of distinct values of the $i$-th attribute (as defined before Definition 3.1). This refinement follows from the fact that the rank $s_i(v)$ in our construction naturally takes values in $[|A_i(V)|]$ rather than $[n]$. Accordingly, the $\log^{O(d)} n$ term in the noise parameters can be replaced by the refined expression above, yielding improved additive error bounds without additional technical difficulty.
>
> In the special case where all vertices share the same attribute value, we have $|A_i(V)| = 1$ for all $i \in [d]$, and thus the extra factor disappears. Therefore, our bounds recover consistency with prior results for standard subgraph counting.
>
> We thank the reviewer for this helpful suggestion and will include this discussion in the final version of the paper.

---

> > ### Author Rebuttal · Reviewer_n6j4 · 2026-04-01
> >
> > Thanks for the responses. I will maintain my score.

---

### Official Review · Reviewer_N29U · 2026-03-11

**Soundness:** 3
**Presentation:** 2
**Significance:** 3
**Originality:** 3
**Overall Recommendation:** 4
**Confidence:** 3

**Summary:**

This paper studies differentially private range subgraph counting, where one must answer counts of a fixed pattern graph inside induced subgraphs selected by multidimensional attribute ranges. The main idea is to project each occurrence of the pattern to a rank-based tuple in $([n]^{2d})$, turning the problem into weighted orthogonal range counting, and then answer queries privately with a noisy range tree, using either global sensitivity or a privately estimated higher-order local sensitivity. The paper also gives lower bounds via a reduction from reconstruction attacks combined with discrepancy arguments, showing an exponential dependence on dimension in relevant regimes. Experiments on two real graph datasets compare the proposed methods against composition-based baselines and show better relative error and query time.

**Compliance With Llm Reviewing Policy:**

Affirmed.

**Final Justification:**

I raised my score and appreciated the authors effort in providing the asked for experimental results. I hope to see the new results and discussions as part of the main paper.

**Key Questions For Authors:**

- Complexity clarification: How exactly are Algorithm 1 and Algorithm 2 implemented in practice without materializing all of $([n]^{2d})$? Please provide the precise time and space complexity of the implementation used in experiments. If the actual implementation is sparse, how does that affect the theorem statement and privacy analysis?
- Dimension dependence: Why are all experiments run at (d=1) when the main theoretical contribution is about how error scales with (d)? Can you provide results for (d>1), even on smaller datasets or with fewer queries?

- Attribute realism: Why were attributes sampled iid from a standard normal distribution instead of using dataset-provided attributes or semantically structured synthetic attributes? How sensitive are the results to the attribute distribution and to query selectivity?

- Baseline strength: Did you consider any stronger workload-aware baseline after projection to weighted points, rather than only independent composition baselines? If not, why is that comparison unavailable or inappropriate here?

**Limitations:**

yes

**Strengths And Weaknesses:**

Strengths
- The problem formulation is strong and timely. Differentially private analysis on attributed graphs is important, and the paper identifies a nontrivial query class that is more realistic than whole-graph motif counting in many applications.

 - The move from induced-subgraph range constraints to weighted orthogonal range counting is the technical heart of the paper, and it is a good one. Equation-level formalism is relatively light, but the connection between Definition 3.1, Definition 3.2, and the query box used in Algorithm 3 is intuitive and coherent.

- The lower-bound component adds real value.This is not just an algorithm paper. The discrepancy-based reduction gives a principled reason why the dimension dependence is hard to improve, which makes the submission more substantial.

- The figures are useful rather than decorative. Figure 1 genuinely helps interpret the data structure, and Figure 2 makes the lower-bound construction much more transparent. That is good scientific communication.

 - The empirical trend is consistent with the intended advantage. In Figure 3, the proposed methods consistently dominate the composition baselines. In Figure 5, the query-time difference is indeed very large, which matches the intuition that preprocessed structures should help for many queries.

Weaknesses

  - The main-paper efficiency claim is not convincingly supported by the algorithms as written. This is my biggest concern. On Page 5, Algorithm 1 initializes $(w_{\mathbf v}=0)$ for every $(\mathbf v\in[n]^{2d})$, and Algorithm 2 constructs a range tree over all such points. Taken literally, this is a dense structure of size $(n^{2d})$. That is already expensive for constant (d>1), and for the regime mentioned in the introduction, $(d=O(\log n/\log\log n))$, it is not obviously polynomial in (n). The paper says the complexity analysis is in Appendix E.3, but the main text should not make an “efficient algorithm” claim when the pseudocode currently suggests something much larger. This matters because efficiency is one of the paper’s headline contributions, not a side detail.

  - The experiments avoid the central variable of the paper, namely dimension (d). The theory repeatedly emphasizes the curse of dimensionality, including in Theorem 1.4 and the discussion around it on Pages 2 to 3. Yet the experiments fix (d=1) throughout. That means the empirical section does not test the actual tradeoff the paper spends most of its theory on. This matters because a reader cannot tell whether the method remains viable beyond the easiest case.

 - The use of synthetic random attributes weakens the practical motivation.  On Page 7, attributes are sampled independently from a standard normal distribution for each vertex. But the introduction motivates applications with meaningful attributes such as age, geography, and risk scores. Those applications are about structured correlations between topology and attributes. Random Gaussian attributes do not reflect that. This matters because the workload and induced subgraphs can change substantially when attributes are meaningful.

-  The baseline story is too convenient. The paper compares mainly against two naive composition baselines. Table 2 makes the theoretical win look crisp, but it is essentially comparing structure-aware release to structure-ignorant release. That is not wrong, but it is also not the strongest test of the contribution. This matters because strong empirical claims require strong baselines. If no better direct baselines exist, the paper should make that case explicitly and perhaps include adapted alternatives from DP range counting.

- The experimental coverage of patterns and datasets is too narrow for the claimed generality. The method is presented for any fixed connected pattern graph (H), but the experiments use only edge, 2-star, and triangle, on two datasets only. There is no evidence for slightly larger motifs where counting and sensitivity behavior become more challenging. This matters because the paper’s generality is mostly theoretical at present.

 - The theory-heavy paper leaves too much outside the main body. The proofs of the main results, the complexity analysis, and several key technical details are deferred. For a paper whose main value is theoretical, that makes the main body feel thinner than it should. This matters because reviewers and readers should be able to audit at least the skeleton of the hardest arguments from the main text.

 - There are avoidable clarity issues in the algorithms and notation. In Algorithm 3 and Algorithm 5, the pseudocode says “for each query $(q\in Q)$” and then “return the query result” inside the loop, which literally returns after one query. This is presumably a notation bug, but it should not be there in a polished theory paper. There are also minor typographical issues, for example the malformed expression around triangle sensitivity on Page 2. These are not fatal, but they hurt confidence.

  - The runtime discussion is partial. Figure 5 is about query time only, and while it looks impressive, it does not settle the practical question because preprocessing and memory are central for this approach. The text says preprocessing is higher and refers to Figure 6, but the discussion still leans heavily on query latency. This matters because practitioners care about total cost, not just per-query time after preprocessing.

 - The query workload is too narrowly chosen and insufficiently justified. The experiments use random queries with $(|V_q|=\Theta(n))$. That may be one regime, but it is far from the only one that matters. The method could behave differently for small selective ranges or heavily skewed workloads. This matters because the evaluation currently reflects one relatively favorable slice of the workload space.

- The main-text literature discussion is incomplete. Because much of the related work is deferred, and because some directly relevant papers are not discussed, the paper does not position itself as sharply as it should. This matters because novelty is easier to appreciate when the comparison to prior private subgraph-analysis work is explicit.

---

> ### Author Rebuttal · Authors · 2026-03-31
>
> Thank you for your review; we summarize your concerns and respond below. See additional experiments at https://anonymous.4open.science/r/Anonymous-C7B5.
> >W1/Q1: How are Algorithms 1–2 implemented without materializing $[n]^{2d}$, what are the time/space complexities in experiments, and how does sparsity affect the theorems and privacy?
>
> We agree efficiency is important and provide details on implementation of the pseudocode in Appendix E.3. Specifically, we maintain the non-zero components of $\mathbf{w}$ by a hash map to avoid materializing all of $[n]^{2d}$, and employ a strategy to implement the range tree more efficiently. The precise time and space complexities are stated in Theorems E.8-E.9. The strategy only removes nodes in the range tree that are entirely irrelevant to the queries; therefore, it has no impact on the additive error guarantees or privacy analysis.
>
> We defer efficiency to the appendix due to space limits, focusing the main text on the accuracy-privacy trade-off; we will revise the paper to better highlight implementation details.
> >W2/Q2: Why are all experiments run at (d=1) when theory is about scales with d? Can you provide results for (d>1)?
>
> We agree that how the error depends on $d$ is central, while our first DP solution already provides meaningful empirical insights into its practical behavior even for $d=1$.
>
> When $d>1$, both the running time and space incur a multiplicative blowup of $\log^{2(d-1)}n$ compared to the case $d=1$. This factor is inherent and becomes substantial even for moderate parameters (e.g., $n=5000$ and $d=2$), making the time and memory costs prohibitive for our current computational capacity.
>
> Despite these limitations, we do include additional experiments on smaller-scale settings where $d > 1$. We evaluated our methods on the CA-Netscience dataset with $n=379$ from the Network Data Repository. We set $|Q|=n^2$ and compare the cases for $d=1,2$, and the results are consistent with our theory. These experiments confirm that our algorithms are primarily advantageous when $n$ and $|Q|$ are large.
> >W3/Q3/W9:Why use iid standard normal attributes instead of dataset-provided or structured ones, and how sensitive are results to the attribute distribution or query selectivity?
>
> The primary reason is the lack of suitable real-world graph datasets with attributes aligned with our setting. Using a standardized Gaussian distribution provides a clean and standard evaluation environment.
>
> We emphasize that our method is not sensitive to the attribute distribution; experiments on real attributes of the Wiki-Squirrel dataset show consistent trends, indicating robustness of our algorithms.
>
> We note that real-world attributes often exhibit redundancy, leading to small effective domain sizes per dimension, which can improve the performance of our algorithms. See our response to Reviewer `n6j4` (Q1) for theoretical justification. In contrast, the Gaussian setting with distinct values per attribute dimension can be viewed as a conservative (worst-case) evaluation.
>
> Empirically, we observe consistent behavior across different query selections. We extend experiments to include additional query workloads with varying selectivity.
> >W4/Q4:Did you consider stronger workload-aware baselines after projection, beyond independent composition; if not, why not?
>
> We appreciate the suggestion and considered standard DP mechanisms after projection, but these are not practical in our setting. Specifically, The dimension of the weight vector $\mathbf{w}$ is $n^{2d}$. Any direct DP mechanism operating on this representation would require at least iterating over the relevant coordinates of $\mathbf{w}$ for each query. This leads to a per-query time complexity of $\Theta(n^{2d})$, which is computationally prohibitive even for moderate values of $n$ and $d$. Similar reasons make our randomized-response baselines (Appendix H.1) infeasible.
> >W5: too narrow experimental coverage of patterns
>
> We agree with the significance of evaluating larger motifs. However, Theorems E.8-E.9 show that the time complexity of our algorithms grows rapidly with pattern size, including $O(n^{|V(H)|})$ for pure DP and $O(n^{2|E(H)|})$ for approximate DP.
>
> As a result, even moderately large $H$ makes these terms computationally prohibitive, rendering larger-motif experiments infeasible. This limitation is inherent to subgraph counting, which is costly even non-privately; we will clarify this in the final version.
> >W6/W7/W10: too much theory outside main text, incomplete literature review, and unclear notations
>
> We note that the main text presents key ideas and proof sketches, with full proofs deferred to the appendix for readability. We appreciate these suggestions and will revise the paper in the final version.
> >W8: partial runtime discussion
>
> We note that the main context already discusses that query time dominates when $|Q|$ is large, our primary regime. We will revise the paper to include total runtime and memory.

---

> > ### Author Rebuttal · Reviewer_N29U · 2026-03-31
> >
> > I appreciate the responses, and would like a discussion on the runtime and the new figures with d > 1 in the main paper.

---

> > > ### Author Response · Authors · 2026-04-04
> > >
> > > > (1) Discussion on the runtime:
> > >
> > > We first recall the timing metrics: the *query time* is the time for answering a query, the *total query time* is the time for answering all the queries, and the *total time* is the sum of preprocessing time and total query time.
> > >
> > > To further illustrate the total time advantage of our algorithms over the baselines, we include additional figures showing total time versus $|Q|$ on the datasets Wiki-Squirrel and WormNet-v3; please refer to  https://anonymous.4open.science/r/Anonymous-2F16/total_time.pdf for details. We assume that the queries in $Q$ are i.i.d. and uniformly sampled from all queries that produce distinct induced subgraphs. $|Q|$ ranges from 1 to $\Theta(n^2)$, which corresponds to the full range of $|Q|$ under our setting when $d = 1$ (see Subsection 1.1, first paragraph). Since exhaustively evaluating the full range of $|Q|$ is beyond our current computational power, we sample a sufficiently large number of queries from the distribution over queries defined above. We continue sampling until the average query time converges, as measured by a small relative standard error (below 5%), and use this average query time to estimate the total query time. Given the orders-of-magnitude difference in query time between our algorithms and the baselines,
> > > the resulting estimation error in these figures is practically negligible.
> > >
> > > These figures show that, on both datasets, our algorithms consistently outperform the baselines in total time when $|Q| = \Omega(n)$, and the advantage becomes increasingly pronounced as $|Q|$ grows. Moreover, the advantage is more significant on larger datasets. When $|Q| = \Theta(n^2)$, the total time is dominated by the total query time, yielding a speedup of 3 to 4 orders of magnitude, consistent with the statements in the main text.
> > >
> > > > (2) Discussion on the new figures with $d>1$:
> > >
> > > These figures show that both the additive error and the runtime of our algorithms increase significantly as $d$ grows, with the observed scaling largely matching our theoretical analysis. While our algorithms no longer outperform the baselines on this dataset when $d = 2$, this behavior is consistent with our theoretical expectations: although our method improves the asymptotic dependence, its advantage only becomes apparent in regimes where $n$ and $|Q|$ are large. Additionally, we observe that our approximate DP algorithm exhibits a more pronounced advantage over our pure DP algorithm when $d = 2$, consistent with their different constant factors in the exponent with respect to $d$ in the theoretical upper bounds.
> > >
> > > Thank you for your further feedback. We will include the discussion on the runtime and the new figures with $d > 1$ in the final version of the paper.

---

### Decision · Program_Chairs · 2026-04-30

**Decision:**

Accept (regular)

**Comment:**

### Metareview

This paper introduces the problem of Differentially Private Range Subgraph Counting (DPRSC), where each vertex contains attribute values and we would like to count the number of certain motifs (i.e. subgraphs) restricted only to vertices whose attributes lie in certain ranges. The authors propose an algorithmic framework that reduces this problem to (orthogonal) multi-dimensional range counting; this yields both pure and approximate DP algorithms by using tree-style mechanisms. They complement their algorithmic contributions with lower bounds demonstrating an exponential dependence on the dimension of the attribute space, nearly matching their algorithm's guarantee.

**Strengths**

* **Well-Motivated Problem Formulation:**
The problem of querying subgraphs restricted by specific vertex attribute ranges is natural and addresses realistic scenarios in privacy-preserving graph analytics (e.g., social or financial networks where attributes act as filters).

* **Solid Theoretical Contributions:**
The reduction to orthogonal range counting and the projection technique are technically sound and intuitive. Furthermore, the discrepancy-based lower bounds nearly match the algorithm's dimensional dependence.

* **Empirical Advantage over Baselines (Reviewers N29U, T7Lu):**
The experiments successfully demonstrate that the proposed range tree-based methods outperform naive composition baselines in both relative error and query time.

**Weaknesses**

* **Privacy Model Limitations (Reviewer uTBn):**
The DP guarantees apply only to the graph edges; vertex attributes are assumed to be public. In many real-world scenarios, the attributes themselves require privacy protection. The authors acknowledged this constraint and mentioned it as an area for future work.

* **Complexity and Scalability Concerns (Reviewers N29U, T7Lu):**
The time and space complexity scales exponentially with the dimension $d$ and the size of the pattern graph. This restricts the practical utility to small motifs and low-dimensional attributes. Indeed, the experiments in the paper were only for the $d = 1$ case. During rebuttal, the authors addressed this by providing additional experiments for $d > 1$ to clarify these limits.

**Recommendation**
The paper tackles a novel, well-motivated extension of privacy-preserving subgraph counting with strong initial theoretical results. Given the technical contributions and the foundational nature of the results for future work in DPRSC (or its extensions), we recommend acceptance.